# Mapping the global distribution of C$_4$ vegetation using observations and optimality theory

Xiangzhong Luo [1,2,9] ✉, Haoran Zhou [3,9] ✉, Tin W. Satriawan[1], Jiaqi Tian[1], Ruiying Zhao[1], Trevor F. Keenan [4,5], Daniel M. Griffith[6], Stephen Sitch [7], Nicholas G. Smith [8] & Christopher J. Still [6]

Plants with the C$_4$ photosynthesis pathway typically respond to climate change differently from more common C$_3$-type plants, due to their distinct anatomical and biochemical characteristics. These different responses are expected to drive changes in global C$_4$ and C$_3$ vegetation distributions. However, current C$_4$ vegetation distribution models may not predict this response as they do not capture multiple interacting factors and often lack observational constraints. Here, we used global observations of plant photosynthetic pathways, satellite remote sensing, and photosynthetic optimality theory to produce an observation-constrained global map of C$_4$ vegetation. We find that global C$_4$ vegetation coverage decreased from 17.7% to 17.1% of the land surface during 2001 to 2019. This was the net result of a reduction in C$_4$ natural grass cover due to elevated CO$_2$ favoring C$_3$-type photosynthesis, and an increase in C$_4$ crop cover, mainly from corn (maize) expansion. Using an emergent constraint approach, we estimated that C$_4$ vegetation contributed 19.5% of global photosynthetic carbon assimilation, a value within the range of previous estimates (18–23%) but higher than the ensemble mean of dynamic global vegetation models (14 ± 13%; mean ± one standard deviation). Our study sheds insight on the critical and underappreciated role of C$_4$ plants in the contemporary global carbon cycle.

C$_4$ is one of the three photosynthetic pathways for terrestrial plants[1] and is reported to account for 18–23%[2–4] of global photosynthesis. C$_4$ plants also drive wildfire dynamics in tropical and subtropical ecosystems[5]. C$_4$ plants first evolved in the low atmospheric CO$_2$ environment of the Oligocene Epoch, roughly 24–35 million years ago[6]. They developed distinct biochemical and anatomical characteristics to enrich CO$_2$ concentration at the site of Rubisco carboxylation in leaves, thereby reducing photorespiration and enhancing carbon-fixation rates[7]. These characteristics produce different climate sensitivities in C$_4$ plants compared to more prevalent C$_3$ plants[4,8], and thus are expected to cause a shift in C$_4$ plant distributions and their contribution to global photosynthesis under contemporary and future climate change[8–10].

[1]Department of Geography, National University of Singapore, Singapore, Singapore. [2]Center for Nature-based Climate Solutions, National University of Singapore, Singapore, Singapore. [3]School of Earth System Science, Institute of Surface-Earth System Science, Tianjin University, Tianjin, China. [4]Department of Ecosystem Sciences, Policy and Management, UC Berkeley, Berkeley, CA, USA. [5]Earth and Environmental Sciences Area, Lawrence Berkeley National Lab, Berkeley, CA, USA. [6]Department of Forest Ecosystems and Society, Oregon State University, Corvallis, OR, USA. [7]Faculty of Environment, Science and Economy, University of Exeter, Exeter, UK. [8]Department of Biological Sciences, Texas Tech University, Lubbock, TX, USA. [9]These authors contributed equally: Xiangzhong Luo, Haoran Zhou. ✉e-mail: xzluo.remi@nus.edu.sg; haoran.zhou@yale.edu

Many previous studies have examined $C_4$ plant responses to multiple environmental factors. A consensus is that since most $C_4$ species originated in lower atmospheric $CO_2$ concentrations[11,12], they are expected to benefit less from rising $CO_2$ concentrations compared to $C_3$ plants. Meanwhile, higher temperatures are expected[4] and reported[13–15] to favor $C_4$ over $C_3$ photosynthesis, because the affinity of $O_2$ to Rubisco relative to $CO_2$ becomes stronger with increasing temperature and also due to differing solubilities of $CO_2$ and $O_2$ with increasing temperature. This should produce an advantage for the carbon concentrating mechanism of $C_4$ species, especially under high temperatures[16]. Hence $C_4$ species are characteristic of tropical and subtropical ecosystems. Correspondingly, since $C_4$ photosynthesis is less limited by $CO_2$ than $C_3$ photosynthesis, it achieves a higher photosynthetic quantum yield and photosynthetic rates under high light, especially under high temperatures[17]. $C_4$ species also should have a carbon assimilation advantage in arid environments[18,19] due to their higher water use efficiency (i.e., less water loss through stomata for equivalent carbon gain) than $C_3$ species, though under humid conditions this advantage could be limited[9]. Contemporary climate change, such as elevated $CO_2$, rising temperatures, and changing rainfall patterns, can therefore lead to temporal and spatial shifts in the relative advantages of $C_4$ to $C_3$ photosynthesis. For instance, the differential response to a changing environment has been linked to observed woody plant encroachment in tropical Africa, where an increase in precipitation and elevated atmospheric $CO_2$ levels are hypothesized to have caused a net decrease in $C_4$ grassland distribution[20,21]. However, we currently lack a consensus on how the relative advantages of $C_4$ to $C_3$ photosynthesis change at the global scale, as regional studies have reported contrasting results and different driving factors—such as increased $C_4$ grass distribution due to increased temperature[22], decreased distribution due to elevated $CO_2$[14] or no overall trend[23]. Understanding of how climate change has impacted $C_4$ vegetation constitutes a major challenge due to its role in global photosynthesis and the terrestrial carbon cycle.

$C_4$ vegetation overwhelmingly consists of natural grasses and crops using the $C_4$ pathway. One prominent approach to estimate the distribution of $C_4$ natural grasses is based on the crossover-temperature model, which predicts that a particular month is determined to favor $C_4$ grasses over co-occurring $C_3$ grasses when the mean daytime air temperature is >22 °C and precipitation in that same month is ≥25 mm[2,24,25]. This approach is based on each pathway's relative carbon assimilation as a function of temperature, and thus the crossover temperature is dependent on atmospheric $CO_2$ concentration with higher crossovers at higher $CO_2$ levels. A few efforts to model $C_4$ vegetation distribution have further incorporated the seasonality of precipitation[26–29], or mean annual temperature and precipitation[8,22,26], but so far they are only validated and applied at the regional scale. Some dynamic global vegetation models (DGVMs) allow adjustment of $C_3$ and $C_4$ grass distribution based on the difference between simulated $C_3$ and $C_4$ photosynthesis or the difference between their net primary productivity[30], or based on the simulations from bioclimate distribution models in each time step, with the baseline $C_4$ map acquired from remote sensing land cover classifications[31,32]. Some cohort-based DGVMs further consider competition for resources[33] and disturbances[34] when simulating $C_4$ distributions. In general, current estimates of the distribution of $C_4$ vegetation adopt a wide range of assumptions and generate rather different results[10].

Uncertainty in global $C_4$ grass distribution is further exacerbated by the lack of ground observations for validation and then for model extrapolation, since previous models often relied on either local datasets[26,27] or literature reviews of $C_4$ grass presence and absence[2] for validation. This issue has become less prominent recently as some studies have used continental scale (i.e., North America) $C_4$ plots[25] and $^{13}C$ isotopic records[23,35] to validate $C_4$ grass distribution models. Meanwhile, the distribution of $C_4$ crops has been collated and estimated in some open datasets[36–39]. These datasets are based on Food and Agriculture Organization (FAO) census and national reporting of the harvested area for major $C_4$ crops (i.e., maize, sorghum, millet, sugarcane), which comprised 24% of the global harvested area[37], and are supplemented by total cropland area change from FAO and remote sensing[40]. In particular, the Land-Use Harmonization dataset version 2 (LUHv2) is the principal gridded land use dataset for the assessment of global carbon budgets[36] and future climate change in CMIP6[41], in which $C_4$ crop area over time is explicitly reported. Changes in global $C_4$ crop distribution and the related contribution to global photosynthesis have yet to be evaluated, except for a few studies that have examined the $C_4$ crop distribution for specific years[2,42,43].

Here we quantify the global $C_4$ vegetation distribution (including natural grasses and crops) and its contribution to global photosynthesis, as well as examine changes in $C_4$ vegetation distribution over the past two decades. To do so, we use photosynthetic optimality theory to estimate the relative advantage of $C_4$ to $C_3$ photosynthesis over the global land surface, and then use the estimated difference in combination with observations to infer global $C_4$ grass distribution. The optimality model includes a wide array of selective drivers for $C_4$ grass distribution - $CO_2$, temperature, light, aridity, nitrogen, and their interactions[44] (see Methods), which are advances over previous crossover-temperature approaches which include $CO_2$, temperature, and a precipitation threshold. The optimality model estimates the optimal leaf photosynthetic rate for $C_3$ and $C_4$ plants, along with optimal stomatal conductance and root/shoot carbon allocation based on growing season climate, with a target to maximize carbon gain with minimized water loss[44]. We further take advantage of multiple open-access databases of $C_4$ species richness and coverage (i.e., the global TRY database[45], a dataset for the contiguous United States (the DG dataset)[23] and a subset of the Nutrient Network (NutNet)[46]), global grassland fraction maps from remote sensing (i.e., as the majority of $C_4$ plant cover is non-woody[47]), in combination with the optimality model simulations to acquire data-constrained estimates of $C_4$ grass distribution for the past 20 years. Meanwhile, we obtain and examine $C_4$ crop distribution using multiple open datasets[36,39]. We further use an emergent constraint technique—a method to infer an unobservable variable from an observable variable based on the large spread of estimates of both variables from DGVMs (see Methods) –to estimate the contribution of $C_4$ plants to global photosynthesis. By quantifying how $C_4$ vegetation distribution and photosynthesis have changed over recent decades, our study improves understanding of historical changes in terrestrial photosynthesis and the global carbon cycle.

## Results

### $C_4$ photosynthetic advantage and $C_4$ grass coverage

We found a strong positive relationship between the observed $C_4$ grass coverage (i.e., the % of grassland area covered by $C_4$ grass species) and the relative advantage of $C_4$ photosynthesis ($A_{C4}$) over $C_3$ photosynthesis ($A_{C3}$) estimated by the optimality model (denoted as the $A_{C4}/A_{C3}$ - $C_4$ coverage relationship hereafter; see Methods; Fig. 1b). With the increase in modeled $A_{C4}/A_{C3}$, the observed $C_4$ coverage increased and then gradually plateaued. When $A_{C4}/A_{C3} = 1$, $C_4$ accounts for only 5.2% of grassland cover; when $A_{C4}/A_{C3} = 2.5$, $C_4$ coverage approaches 100% (Fig. 1b). Across global non-woody regions, $A_{C4}/A_{C3}$ ranged from 0.5 to 2.5, with a mean of 1.9 (Fig. 1a). Importantly, we obtained $C_4$ coverage observations from multiple sources (i.e., TRY and DG; see Methods), which have different geographic representations and spatial resolutions. However, the $A_{C4}/A_{C3}$ - $C_4$ coverage relationships are similar when using different observations (Fig. 1b), affirming the robustness of the relationship for $C_4$ coverage estimation. Using the relationship between $A_{C4}/A_{C3}$ and $C_4$ coverage (Fig. 1b), and the global $A_{C4}/A_{C3}$ estimated from the optimality model (Fig. 1a), we estimated the global $C_4$ grass coverage (% of grassland

covered by $C_4$; Fig. 1c). We found $C_4$ grass coverage followed a clear climatic gradient, and tended to be greater under warmer conditions (Fig. 1d).

## The global distribution of $C_4$ vegetation

After predicting the $C_4$ grass coverage (% of grassland covered by $C_4$ grasses; Fig. 1c), we overlaid a global grassland fraction map from remote sensing (see Methods and Fig. S1) to estimate the actual $C_4$ natural grass area abundance (% of the land surface covered by $C_4$ grasses; Fig. 2a). From 2001 to 2019, $C_4$ natural grass accounted for $14.8 \pm 1.3\%$ (mean $\pm$ one standard deviation) of the non-frozen land surface area (Fig. 2a), while $C_4$ crops accounted for $2.8 \pm 0.3\%$ (Fig. 2c). The total estimated $C_4$ area abundance was $17.5 \pm 1.4\%$ (Fig. 2e). There were several $C_4$ natural grass hotspots (i.e., >30% $C_4$ area abundance) across continents (Fig. 2a): the Great Plains in North America, the savannas in Southern Brazil, the savannas in Africa, the grasslands in Central Asia, and Northern Australia. Meanwhile, we found the main $C_4$ crop zones were in central North America, the Sahel region, and the west coast of India (Fig. 2c). The disagreement between remote sensing-based grassland fraction maps (Fig. S4), along with the uncertainty in the $AC_4/AC_3$ - $C_4$ coverage relationship (Fig.1b), incurred uncertainties in the $C_4$ natural grass distribution (Fig. 2b)—the uncertainty typically ranges between 1 and 3% of the land surface area, though in regions like Australia the uncertainty could be as high as 6–7% (Fig. 2b).

## The changes in $C_4$ vegetation distribution

Based on our simulation of $C_4$ natural grass distribution for the past two decades and the $C_4$ cropland distribution from the LUHv2 dataset (see Methods), we found the overall area of $C_4$ vegetation decreased from $17.7 \pm 1.4\%$ (mean $\pm$ one standard deviation) in 2001–2005 to $17.1 \pm 1.4\%$ in 2015–2019, as a net effect of a decrease in $C_4$ natural grasses from $15.0 \pm 1.3\%$ to $14.2 \pm 1.3\%$, and an increase in $C_4$ crops from $2.6 \pm 0.3\%$ to $3.0 \pm 0.3\%$ (Fig. 3b). The change in $C_4$ shows large spatial heterogeneity (Fig. 3a). In particular, $C_4$ natural grass area decreased all over the globe, except for the central Europe and parts of the western U.S. (Fig. 3c). $C_4$ crop area increased in most parts of the world, except for central North America where there was the largest decrease, and Europe where there were slight decreases (Fig. 3e).

The increase in $C_4$ crop area was often at the expense of decreasing $C_4$ natural grasses, as we found that in regions where there were both $C_4$ crops and $C_4$ natural grasses, more than 50% of the regions showed $C_4$ natural grasses decreased but $C_4$ crops increased. Meanwhile, only 14% showed that $C_4$ crops and $C_4$ natural grasses increased simultaneously, 28% showed they both decreased, and only 7% of the region showed that $C_4$ natural grasses increased and $C_4$ crops decreased (Fig. 3d). Our attribution analysis suggested that elevated $CO_2$ was the dominant reason for the decrease in $C_4$ natural grass distribution, while the impacts of temperature and water stress (i.e., soil moisture and vapor pressure deficit) were positive (i.e., increase $C_4$ coverage) over the study period (Fig. 3f). Most of the increase in $C_4$ crops, as we analyzed from another independent dataset

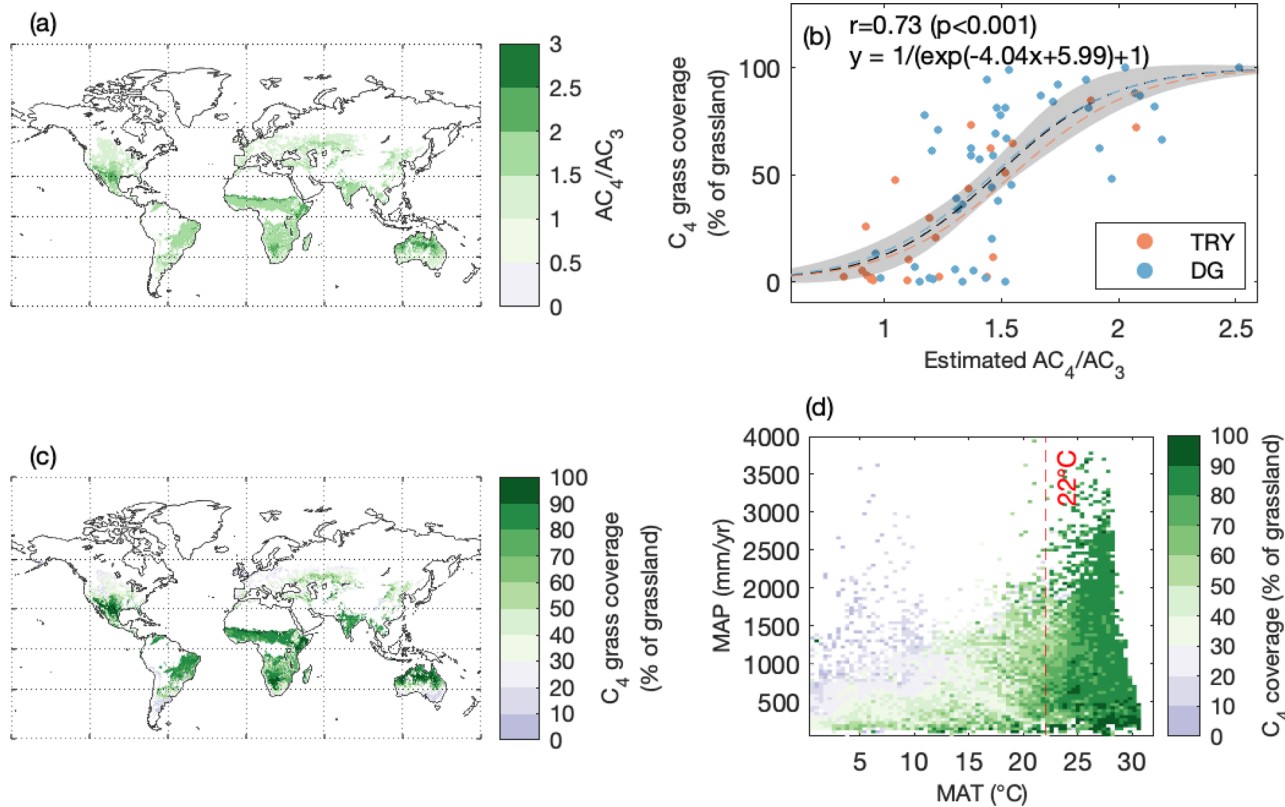

**Fig. 1 | $C_4$ natural grass coverage estimated by the optimality model. a** the ratio of $C_4$ to $C_3$ photosynthesis estimated by the optimality model ($AC_4/AC_3$) over global non-woody regions; (**b**) the relationship between observed $C_4$ coverage (% of grassland) and estimated $C_4/C_3$ photosynthetic ratio by the optimality model. $C_4$ coverage observation obtained from difference sources (i.e., TRY, DG datasets; please see methods); gray shaded area indicates the uncertainty range for the relationship between $AC_4/AC_3$ and $C_4$ coverage (i.e., 95% confidence interval). The black line represents the regression using both the TRY and DG datasets, while the red and blue dash lines represent the regression using either the TRY or the DG dataset. **c** $C_4$ grass coverage (% of grassland) over the globe, which can be regarded as the potential $C_4$ area abundance when grassland covers 100% of the land surface; (**d**) $C_4$ coverage in a climate space of mean annual temperature (MAT: °C) and mean annual precipitation (MAP: mm/yr).

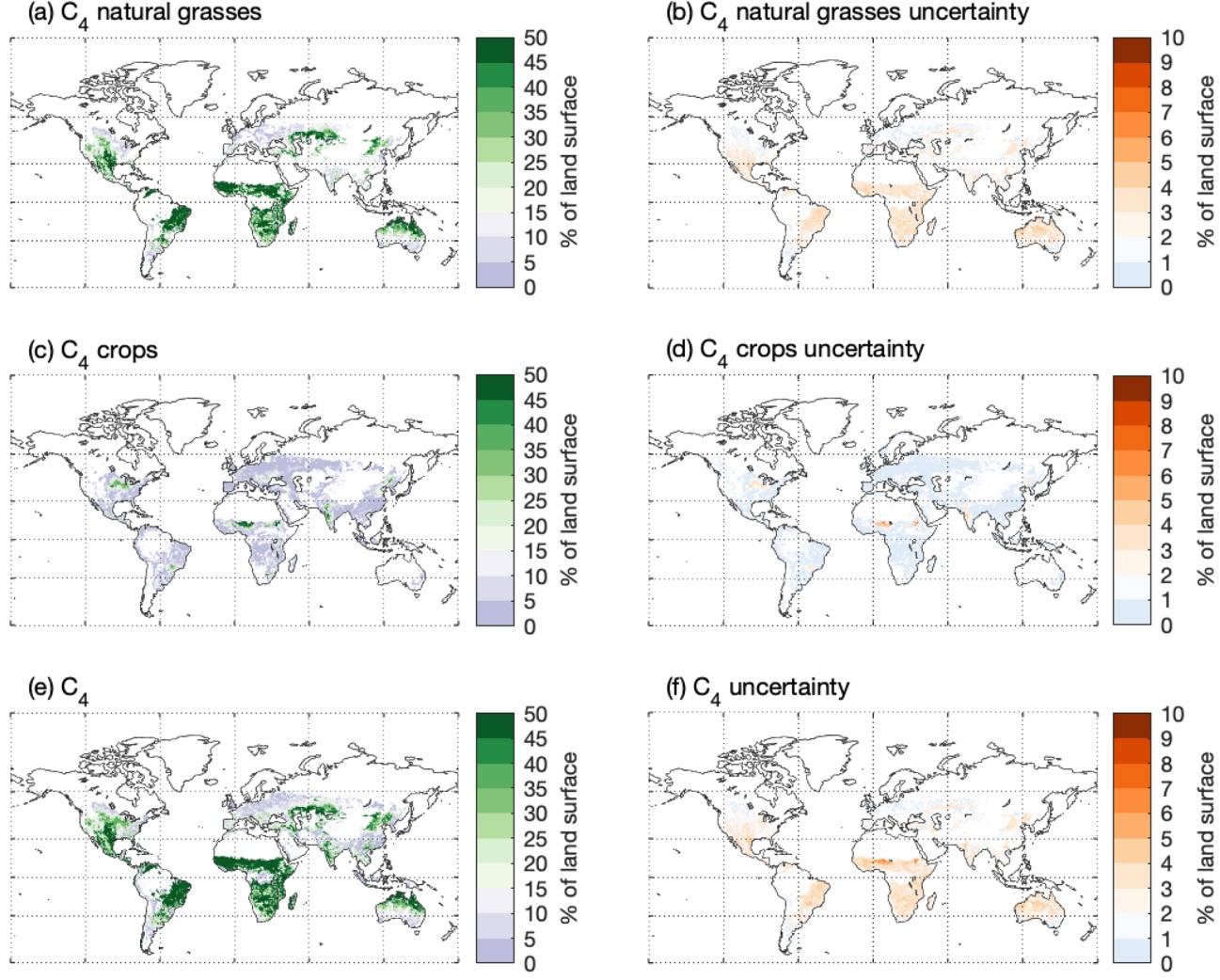

**Fig. 2 | The modeled global distribution of C$_4$ vegetation and associated uncertainties.** The area occupied by (**a**) C$_4$ natural grasses, (**c**) C$_4$ croplands and (**e**) all C$_4$ vegetation (unit: % of the land surface). The uncertainties of the area abundance of (**b**) C$_4$ natural grasses, (**d**) C$_4$ croplands and (**f**) all C$_4$ vegetation (unit: % of the land surface).

on major C$_4$ crop distributions[32], came from the expansion of maize in South America and eastern Europe (Fig. 3f; Fig. S9; see Methods).

### The contribution of C$_4$ vegetation to global photosynthesis

The changes in the C$_4$ area can cause associated changes in total C$_4$ photosynthesis, thus impacting global carbon cycle dynamics. Current ensemble of DGVMs predicted that C$_4$ vegetation contributed from 2% to 40% of global photosynthesis, on 7% to 23% of the global vegetated land surface area (Figs. S6, S7). The large spread of model estimates indicates the various assumptions adopted in C$_4$ vegetation distribution and potentially the different parameterizations for C$_4$ photosynthesis in DGVMs. Despite these wide inter-model variations, it is possible to infer emergent constraints on C$_4$ vegetation contributions to the carbon cycle. To quantify the contribution of C$_4$ photosynthesis to global photosynthesis, we established an emergent constraint ($p < 0.01$) between the DGVM-simulated occupied area and percentage contribution of C$_4$ natural grasses and crops to global photosynthesis, respectively. We found that with a 1% increase in area, C$_4$ natural grass contribution to global photosynthesis increased by 1.10% (Fig. 4a), while the contribution of C$_4$ crops increased by 1.16% (Fig. 4b). We also conducted a grid cell-level emergent constraint analysis and acquired similar ranges of slopes (Fig. S10). The lower coefficient of emergent constraint for C$_4$ grass (i.e., 1.10) than the

coefficient for C$_4$ crop (i.e.,1.16) suggests that croplands tend to have a higher ecosystem photosynthetic rate compared to grasslands over the same area.

We further explored the changes in the coefficients of emergent constraints over the past two decades. We note that for C$_4$ grasslands and C$_4$ croplands, the coefficients all slightly decreased in the past two decades. The coefficient of C$_4$ crops decreased from 1.16 to 1.15, while the coefficient of C$_4$ grasses decreased from 1.11 to 1.10 from 2001 to 2019 (Fig. 4c). Interestingly, the coefficients are all greater than 1, highlighting that the per unit area photosynthetic rate of C$_4$ is generally higher than that of the remaining C$_3$ vegetation. With the likely increase of global photosynthesis in recent decades, the decreasing coefficients of C$_4$ indicated that C$_4$ photosynthesis increased at a slower pace than other (mostly C$_3$) vegetation. By applying the estimated area of C$_4$ to the annual emergent constraint coefficients (Fig. 4c), we found that the global C$_4$ natural grass contribution to photosynthesis decreased from 16.5 ± 1.5% (mean ± one standard deviation) in 2001–2005 to 15.5 ± 1.5% in 2015–2019, the C$_4$ crop contribution to photosynthesis increased from 3.0 ± 0.3% to 3.4 ± 0.4%, and in total the C$_4$ contribution to global GPP decreased from 19.7 ± 1.9% to 19.0 ± 1.9% (Fig. 4d). The reported value is greater than the ensemble mean reported by the DGVMs (14 ± 13%), and is within the range of previously modeled values (18–23%[2–4]).

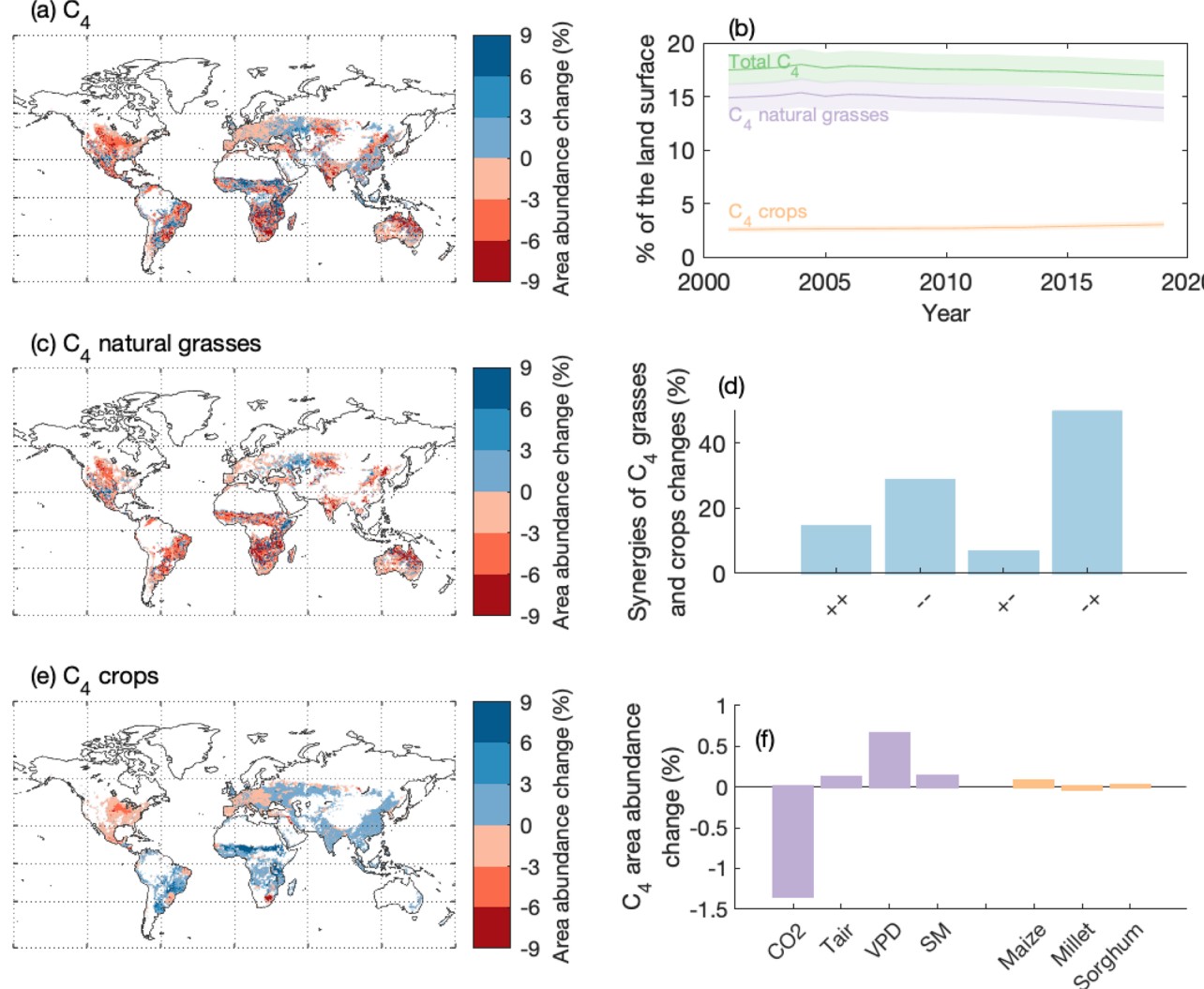

**Fig. 3 | Changes in the global distribution of C$_4$ vegetation between 2001–2019.** Spatial distributions of changes in (**a**) total C$_4$ vegetation, (**c**) C$_4$ natural grasses and (**e**) C$_4$ crops from 2001 to 2019; **b** The changes in the total area of C$_4$ vegetation, C$_4$ natural grasses and C$_4$ crops, in percentages of global vegetated land surface; **d** the synergies of changes in C$_4$ natural grasses and C$_4$ crops, where ++ means both C$_4$ natural grasses and C$_4$ crops area abundance increased, – – means both decreased, + – means C$_4$ natural grasses increased and C$_4$ crops abundance decreased, – + means the opposite; **f** the drivers for the change in C$_4$ natural grasses

and C$_4$ crops area abundances. Climate drivers include atmospheric CO$_2$ concentration, air temperature (T$_{air}$), vapor pressure deficit (VPD) and soil moisture (SM) during the growing seasons. In (**b**), the uncertainty for C$_4$ crop is 10% of the C$_4$ crop area reported, the uncertainty for C$_4$ natural grass area is the combination of the uncertainty in remote sensing-based grassland fraction and the uncertainty of the AC$_4$/AC$_3$ · C$_4$ coverage relationship (Fig. 1b), and uncertainty for C$_4$ vegetation is the combination of uncertainties of C$_4$ crop and C$_4$ natural grass areas.

## Discussion

In this study, we estimated the global C$_4$ vegetation distribution and quantified the changes in C$_4$ vegetation distribution and photosynthesis over the past two decades, using an optimality photosynthesis model, photosynthetic pathway records from global/regional databases and remote sensing observations. On average, from 2001 to 2019, C$_4$ plants occupied 17.5 ± 1.4% of the global vegetated surface and contributed 19.5 ± 1.9% of global photosynthesis, within the range of previous estimates (i.e., 18–23% for photosynthesis) but are greater than the estimates from the ensemble mean of DGVMs (13 ± 8% for area and 14 ± 13% for photosynthesis). C$_4$ total area and C$_4$ contribution to global photosynthesis both decreased over this period (i.e., 0.6% of the land surface and 0.7% of global GPP), which resulted from the increases in C$_4$ crop area and its contribution to global photosynthesis, and the decreases in C$_4$ natural grass area and its contribution to global photosynthesis.

Our study suggests that the decrease in C$_4$ natural grass distribution was primarily driven by elevated CO$_2$, in accordance with

previous theoretical and experimental works that showed C$_4$ advantage in carbon assimilation over C$_3$ decreased with rising CO$_2$[48–50]. There is also evidence showing many grassland and savanna areas have been invaded by C$_3$ woody species, with increased atmospheric CO$_2$ proposed as a major driver for the encroachment[51,52]. The decrease in the emergent constraint coefficients demonstrated that the impact of elevated CO$_2$ on C$_4$ and C$_3$ were included in most DGVMs (Fig. 4c). Evidence for the historical expansion of C$_4$ over geological time scales seems to support our conclusion, in particular for Africa where CO$_2$ dominates the C$_4$ grassland expansion or decline[50,53,54], while in Central Asia[55], Australia[56], central China[57] and central US grasslands[58] reports show hydroclimatic change impacted C$_4$ grass distribution. Our results also show that the changes in soil moisture and VPD over the past two decades largely induced positive impacts on C$_4$ grass distribution, though globally their impacts were unable to offset the negative changes driven by elevated CO$_2$ (Fig. 3f; Fig. S8).

To assess the distribution of C$_4$ vegetation, we noted that there were three types of abundance used in previous literatures: for

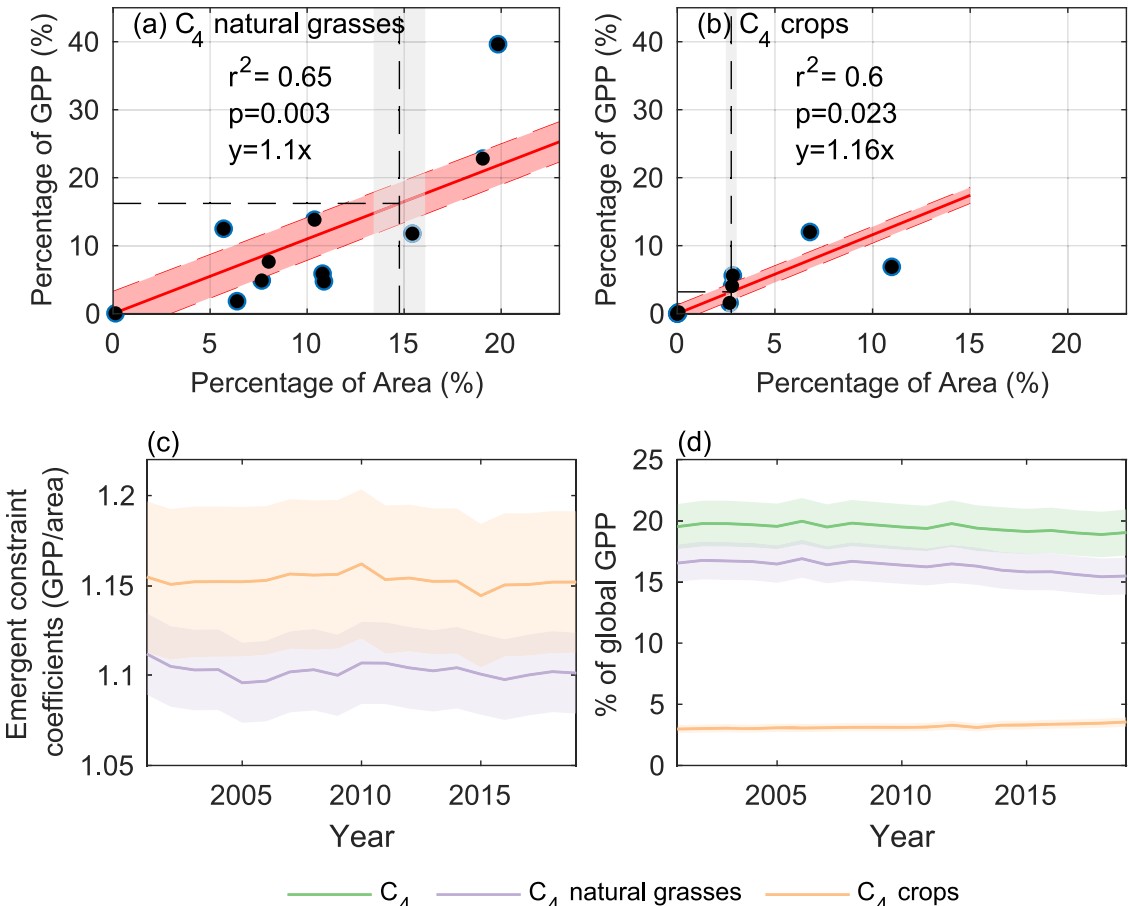

**Fig. 4 | The contribution of C₄ vegetation to global photosynthesis.** The emergent constraints between the percentage of area occupied and the percentage of global photosynthesis contributed by (**a**) C₄ natural grasses and (**b**) C₄ crops, based on the estimates from an ensemble of the DGVMs; **c** changes in emergent constraint coefficients (i.e., the slopes of the linear regressions in (**a**, **b**) from 2001 to 2019 for C₄ natural grasses and C₄ crops). The uncertainties in (**a**, **b**, **c**) were quantified as one standard error (i.e., SE) by bootstrapping models when getting emergent constraint; **d** The contributions of C₄, C₄ natural grasses and C₄ crops to global GPP from 2001 to 2019, whereas the uncertainty was quantified as one SE by bootstrapping the uncertainty range of C₄ areas and the uncertainty range of the emergent constraint coefficients.

modeling practice, we often used the area abundance, that is, the percentage of the land surface occupied by C₄ plants. However, the area abundance was challenging to observe over large scales, and it was rare to have direct observations of large-scale area abundance other than approximations from remote sensing[32]. Most observations of C₄ presence and cover are at the species level and, therefore, observation-based studies often report the relative species richness of C₄ plants[13,26]. A few studies reported biomass abundance, i.e., the percentage of local biomass contributed by C₄ plants, and biomass abundance is often much higher than the species abundance[27,29], echoing some studies suggesting that species abundance should be used with caution to infer biomass abundance and productivity[56]. In this study, we developed a conversion factor to translate C₄ species richness into C₄ area abundance, using plot-level concurrent measurements of both from NutNet. We found 1% increase in C₄ species richness led to 1.51 ± 0.15% in C₄ area abundance (Fig. S3). The conversion factor enabled us to translate global observations of C₄ species richness into C₄ coverage to develop the AC₄/AC₃ - C₄ coverage relationship in our study (Fig. 1b, S1).

Compared to the previously estimated C₄ vegetation distribution from a crossover-temperature model[2] (Fig. S5), our study provided a similar area occupied by C₄ vegetation (20.5 million km² versus 21.1 million km²). We find a lower estimate for Africa, where our estimate of C₄ area abundance showed a more nuanced gradient compared to the estimate from the crossover-temperature model (Fig. S5). We suspect this is partly due to their difference in relating C₄ photosynthetic advantage to C₄ grass coverage (% of grassland covered by C₄ grasses) – while our study used the AC₄/AC₃ - C₄ grass coverage relationship to gradually adjust C₄ grass coverage, the crossover-temperature model assumes all grasslands in the pixel are C₄ grassland as long as the monthly climate satisfies the crossover criteria (e.g., mean daytime air temperature is >22 °C and precipitation in that same month is ≥25 mm). Therefore, for regions where monthly climate meets the crossover criteria (e.g., sub-Sahel Africa), the crossover-temperature model tends to estimate 100% C₄ grass, however, it was not the case for the optimality approach in those regions (Fig. 1c). Meanwhile, we estimated considerably higher C₄ grass cover in Central Asia, which is consistent with the reported prevalence of C₄ species in the region[26,59] – the high abundance in these inland regions were potentially due to harsh environments characterized by high maximum temperature and aridity levels, which favor the growth of C₄ plants over C₃. We also note that our approach may underestimate C₄ grass distribution in some mesic savanna ecosystems—such as the longleaf pine savannas in the Southeastern US—where a C₄ understory exists beneath a C₃ canopy[60,61]. The underestimation is likely because the optimality model predicted no photosynthetic advantage for C₄ plants over C₃ under low light conditions in understory (Fig. 1a), and the remote sensing products reported low grassland fraction in the region (Fig. S4).

In our study, we estimated the annual distribution of $C_4$ grasses using the growing season mean climate; however, many locations have seasonal shifts between $C_4$ and $C_3$ grass dominance depending on seasonal climate variations[4]. For example, in the grasslands of southeast Australia, a recent study suggests $C_4$ dominance is the highest in summer when there is high temperature and low precipitation, while other seasons have more $C_3$ vegetation[28]. Therefore, for modeling the seasonal variation of carbon fluxes from seasonal changes in $C_4$ grass distribution, the crossover-temperature approach based on monthly climate and weighted by a vegetation index like NDVI could be more useful[2]. We acknowledge that the distribution of our observations was not uniform across the globe, with North America being better represented compared to other regions (Fig. S2). As an additional test to validate our estimation of $C_4$ vegetation distribution, we compared the $C_4$ grass coverage estimated in our study (Fig. 1c) with the $C_4$ coverage estimated from isotopic measurements and remote sensing in Australia[62] (Fig. S11). This validation demonstrates a strong agreement ($r = 0.69$, $p < 0.01$) between the two independent estimates, affirming the robustness of our estimates in under-sampled regions.

We also need to highlight that the optimality approach was based on the assumption that $C_4$ grass distribution is determined by the photosynthetic advantage of $C_4$ compared to $C_3$, and the photosynthetic advantage is largely dependent on local climate[44]. While this assumption is also adopted by the crossover-temperature model, it neglects the role of grass phylogeny in determining $C_4$ grass distributions. Some studies have suggested that grass clades (i.e., Pooideae for $C_3$, and PACCMAD for $C_3$ and $C_4$) are perhaps more critical than photosynthetic pathway for determining $C_4$ and $C_3$ grass distributions, at least along temperature gradients[19,63,64]. This lack of consideration on phylogeny in $C_4$ grass distribution models may impact predictions of future distributions, as $C_3$ grasses from certain clades have less competitive disadvantage compared to $C_4$ grasses in a warmer world.

Other environmental changes that can impact $C_4$ grass distribution include fire and nitrogen deposition. Fire can influence $C_4$ coverage and productivity either through creating open canopies for light capture by $C_4$ plants or as a characteristic of semi-arid environments that provide a photosynthetic advantage for $C_4$[65]. Recent woody plant encroachment suggests fire had a central role[51,66] in the formation of grasslands and the rise of $C_4$ dominant grasslands in late Neogene[67] and late Miocene[68]. For recent decades, since globally the trend of fire occurrence is still very uncertain with strong regional variations in the trend[69], we were unable to quantify its impact on $C_4$ grass distribution. Neither the crossover-temperature $C_4$ model, nor the optimality model we used, incorporates the role of fire in $C_4$ dynamics at present. However, since our approach used annual grassland fractional maps based on satellite remote sensing, the impacts of fire at the annual scale were implicitly considered. Some DGVMs have incorporated fire-relevant processes, however, we found they estimated lower $C_4$ grass abundance (i.e., 9.6 ± 6.7%) than those models that do not include fires (i.e.,11.8 ± 3.2%). Additionally, since $C_4$ plants have higher photosynthetic nitrogen use efficiency than $C_3$[70], anthropogenic nitrogen deposition[71] might have impacted the relative advantage of $C_4$ to $C_3$ photosynthesis. Previous studies have suggested $C_4$ plants tend to have a higher photosynthetic rate than $C_3$ across a spectrum of nitrogen supply - meaning $C_4$ plants have photosynthetic advantage on infertile soils and the advantage will be enhanced by increased nitrogen availability, which can be used to increase $C_4$ leaf area[72]. In this study, we used a data-driven product of leaf nitrogen content and remote sensing leaf area index to simulate $C_4$ grass distribution (see Methods), which may have implicitly accounted for the effect of nitrogen deposition or limitation on $C_4$ photosynthesis.

In this study, we used LUHv2—the primary dataset used in current global carbon cycle modeling and climate forecasting[36,41]—to examine the changes in $C_4$ cropland and reported an increase in $C_4$ cropland area. However, we note a key source of uncertainty in this dataset: the historical simulation of $C_4$ crop distribution used a constant fraction of $C_4$ crop cover for each global grid cell, based only on observations circa 2000[41]. Therefore, the increase in $C_4$ crop area in LUHv2 could just reflect an increase in all croplands rather than real $C_4$ expansion. To reduce the uncertainty caused by this issue, we used another cropland dataset[39], which dynamically simulated the area of 17 major crop types (including main $C_4$ crops maize, millet and sorghum) based on annual FAO census of crop harvested areas. This analysis confirmed our results regarding the increase in $C_4$ cropland mainly due to the expansion of maize (Fig. 3f), with a similar spatial pattern reported (Fig. S9). However, the sum of the three main $C_4$ species only increased by 0.1%, relatively lower than what we see from the LUHv2 dataset (i.e., 0.4%). We can conclude there was an increase in $C_4$ cropland, though the magnitude of increase should be subject to further examinations.

In conclusion, we used a combination of plant photosynthetic pathway records, remote sensing, and an optimality-based photosynthesis model to estimate the global $C_4$ coverage and the magnitudes of $C_4$ photosynthesis and their variations over the past two decades. We infer that $C_4$ vegetation covered on average 17.5% of the global land surface over the period from 2001 to 2019, while $C_4$ grass cover decreased due to elevated $CO_2$ and $C_4$ crop cover increased because of corn (maize) expansion. We predict that $C_4$ photosynthesis accounted for 19.5% of the global total photosynthesis, with an increased contribution from $C_4$ crops and a decrease from $C_4$ natural grasses during this period. Our study offers an updated and more observationally constrained estimate of $C_4$ vegetation distribution and photosynthesis, thereby improving our understanding of potential future $C_4$ changes and enhancing the quantification of the global carbon budget.

## Methods
### The overarching framework
The distribution of $C_4$ vegetation overwhelmingly consists of $C_4$ natural grasses and $C_4$ crops. To estimate the $C_4$ natural grass distribution, we first used an optimality photosynthesis model[44] to simulate the optimal photosynthetic assimilation rates of $C_4$ and $C_3$ plants (noted as $AC_3$ and $AC_4$, respectively) using $0.5 \times 0.5$ degree gridded historical climate (i.e., CRU-JRA2020), soil[73] and leaf nitrogen content[74]. We calculated the ratio of $AC_4$ to $AC_3$, and established a statistically significant ($p < 0.01$) relationship between $AC_4/AC_3$ and the observed $C_4$ coverage from multiple databases—the TRY database[45] and the DG dataset[25] based on an assumption that larger $AC_4/AC_3$ indicates higher $C_4$ grass coverage (% of grassland covered by $C_4$ grasses). Using the $AC_4/AC_3$ - $C_4$ coverage relationship, we estimated the $C_4$ grass coverage (a.k.a. potential $C_4$ abundance when grasses cover 100% of the land surface) from estimated $AC_4/AC_3$ for the globe. We lastly overlaid the $C_4$ coverage map to a global map of grassland fraction from remote sensing to acquire actual $C_4$ grass abundance (% of the land surface covered by $C_4$). The workflow is presented in Fig. S1. Meanwhile, for $C_4$ crop distribution, we directly used the estimates from the LUHv2-2019 dataset, in which $C_4$ crop distribution is estimated from FAO survey and satellite remote sensing.

### Processing observational $C_4$ records
We acquired 61,588 georeferenced records of photosynthetic pathways from the TRY database (last accessed 2022 June), among them, there were 2269 records of $C_4$. The time range of the record covers roughly the past 50 years. We first removed the woody species from the records, based on species names and an index table from the TRY database (https://www.try-db.org/TryWeb/Data.php#3), as our study aimed to examine $C_4$ grass distribution and the global cover of $C_4$ forests is not extensive[47]. After the step, we kept 13,919 records for non-woody species, among which 1963 were $C_4$. We further removed 82 records that belong to major $C_4$ crops (i.e., maize, sugarcane, millet,

and sorghum) and kept 1881 records. We then aggregate these records to $10 \times 10$ degree cells, in each cell we calculate the species richness of $C_4$ (i.e., number of $C_4$ species/total number of herbaceous species, the numbers were derived from the available records in the TRY database). The gridded values of $C_4$ species richness would be further used to constrain the optimality model to estimate global $C_4$ grassland coverage. Here we use the large-size grid cell to make sure there were enough samples in each cell to acquire a meaningful estimate of $C_4$ abundance – in this analysis, each cell should have at least 50 species (i.e., $C_3$ and $C_4$ in total). We used 1619 (out of 1881) $C_4$ species records in this aggregation step, and obtained 23 $10 \times 10$ degree cells for the analysis (Fig. S2).

Note here the $C_4$ abundance from the TRY database was species richness, not equal to the area abundance that is often used in DGVMs. To acquire $C_4$ grass coverage (% of grassland covered by $C_4$) from $C_4$ species richness (% of grass species that is $C_4$), we used an open dataset from the global nutrient network (NutNet) that has paired $C_4$ species richness and $C_4$ area abundance (% of the land surface covered by $C_4$) to infer their relationship[46] (Fig. S3). The dataset includes species-specific coverage records as well as the grass species richness data collected in 25 $m^2$ plots across 34 sites. Each site has between 1 and 6 control plots. We only used the data from the control plots, excluding plots that underwent nutrient addition treatments. To avoid the uneven distribution of data samples, we grouped the paired observations by their $C_4$ species richness, and for each species richness we get a mean $C_4$ area abundance and the standard deviation of the $C_4$ grass coverage. We then conducted 1000 linear fittings (i.e., with an intercept of 0, since $C_4$ grass coverage should be 0 when $C_4$ species abundance is 0), and for each fitting we used randomly sampled $C_4$ grass coverage values (i.e., based on mean and the standard deviation) value against $C_4$ species richness values. The slopes of the linear regressions represented a conversion factor between $C_4$ species richness and $C_4$ grass coverage (Fig. S3).

In addition to the TRY database that has a global representation, we also used a gridded $C_4$ grass coverage data compiled for the contiguous United States (denoted as the DG dataset)[23]. The DG dataset provides $C_4$ grass coverage (% of grassland) aggregated at a 100 km resolution grid, which was sampled from roughly 40,000 plots over the past 40 years. Please note that the DG dataset only surveyed $C_4$ grass species. We used the DG dataset and the TRY database to establish the relationship between $AC_4/AC_3$ and $C_4$ grass coverage (Fig. 1b).

### Processing cropland and land use data

We used a gridded $C_4$ crop distribution from the LUHv2 dataset (version: LUHv2-GCB2019)[36]. It was estimated based on the FAO census and national reporting of >170 major crop types (including the main $C_4$ types), supplemented by the total cropland area collated by HYDE3.2[40] which came from FAO census and remote sensing products. The $C_4$ crop fraction of each grid cell was only acquired based on observations circa 2000[37] and the fraction was kept constant over the study period. The most recent versions of the LUHv2 dataset have been used in CMIP6 for the IPCC AR6 report and the global carbon budget. Since the LUHv2 dataset did not contain an estimate of uncertainty, we relied on an independent study that compared four different land use products (including LUHv2) and reported the uncertainty of cropland estimation between products was about 10%[75]. We thus used 10% to represent the uncertainty range of the $C_4$ cropland area.

In addition to LUHv2, we used another open dataset reporting the area of 17 main crop types from 1961 to 2014[39]. Unlike the LUHv2 dataset which almost exclusively relied on observations circa 2000 to quantify the $C_4$ crop fraction, this other dataset used annual FAO census records for crop area fraction estimation, including those of the three primary $C_4$ crops: maize, millet, and sorghum. We used this

dataset to examine the changes in global $C_4$ croplands and compare to the values obtained from the LUHv2 dataset.

### The optimality model for $C_4$ and $C_3$ photosynthesis

We used optimal $C_3$ and $C_4$ photosynthesis models to simulate optimal $C_3$ and $C_4$ photosynthesis[44]. The soil-plant-air water continuum was incorporated in $C_3$ photosynthesis models[76] and $C_4$ photosynthesis models[77] to examine interactions of $CO_2$, water availability, light and temperature. The model considered optimal stomatal resistance and leaf/fine-root allocation to maximize the carbon gain regarding water loss, and successfully predicted the ancient distribution of $C_4$ species in Oligocene and Miocene[44].

In the current study, we improved the modeling processes through the following aspects. (1) We used different parameters for $C_3$ and $C_4$ species specifically to better represent the diversity of $C_3$ and $C_4$ species variability (Table S2 in the Supplementary Note). (2) We considered the effects of nitrogen availability and optimal nitrogen allocation between $C_3$ and $C_4$ species. Specifically, we adjusted the maximum carboxylation rate ($V_{cmax}$) and maximum electron transport rate ($J_{max}$) values using optimal $J_{max}/V_{cmax}$ ratio (i.e., 2.1 for $C_3$ and 5.0 for $C_4$, which were supported by both measurements and theoretical modeling)[78,79] and available leaf nitrogen content for $C_3$ and $C_4$ respectively. (3) Since a large majority of $C_4$ species are herbaceous, when we modeled closed canopy biomes (e.g., those pixels dominated by tree and shrubs), we used estimates of understory photosynthetic active radiation (PAR) to model the relative advantage of the herbaceous species. A full model description and the parameterization is in the Supplementary Note.

Using the models, we were able to calculate the optimal assimilation rates for $C_3$ and $C_4$ (i.e., $AC_3$ and $AC_4$) over the globe at the 0.5-degree resolution (i.e., dependent on the spatial resolution of climate input), where the relative advantage of $C_4$ to $C_3$ is defined as $AC_4/AC_3$. The simulation was conducted at an annual time step and there was no need for model initialization. When establishing the relationship between $AC_4/AC_3$ and $C_4$ grass coverages (Fig. 1b) from the DG and TRY datasets, we aggregated the simulations from 0.5-degree to 1-degree (approximately 100 km at the equator) and 10-degree. As the relationships derived from both 10-degree (i.e., TRY) and 1-degree (i.e., DG) data were similar, we assumed that the relationship is scale-independent. Consequently, we applied it to 0.5-degree estimates of $AC_4/AC_3$ to infer global $C_4$ grass coverage. We also assumed that the relationship between $AC_4/AC_3$ and $C_4$ grass coverage was time-invariant.

### Running the optimality model

We used annual growing season average soil water potential, vapor pressure deficit (VPD), 2 m daytime air temperature ($T_{air}$), photosynthetic active radiation (PAR) and leaf nitrogen content as inputs for the optimality photosynthesis model. The growing season was defined using the MODIS phenology product (MCD12Q2)[80].

$T_{air}$ was acquired from the CRU-JRA2020 dataset. VPD was estimated using the specific humidity and air temperature from CRU-JRA2020. Soil water potential was estimated from soil texture properties from soil grids and global soil water content datasets, using the Clapp & Hornberger equation[81]. The global soil water content datasets came from GLEAM v3[82]. To avoid extreme low soil water potential that does not allow plant growth in the optimality model, we set the minimal soil water potential to $-3$ MPa. The leaf nitrogen content was acquired from a machine learning upscaled leaf traits product[74]. PAR was also acquired from the CRU-JRA2020 dataset, which is a reanalysis from CRU[83] and JRA[84]. We directly used PAR for 'open' ecosystems (i.e., grasslands, savannas), however, for dense forests and shrublands we used understory PAR (i.e., as $C_4$ grasses often exist in understories), which was derived from PAR and multi-year average

MODIS LAI[85] (i.e., assume they are overstory LAI) following a radiation gradient mandated by the Beer's Law.

We ran the photosynthetic optimality models multiple times in the process. We first modeled the growing season $AC_4/AC_3$ in the study period using the climatology of the variables mentioned from 2001 to 2019. To model the growing season $AC_4/AC_3$ from 2001 to 2019, for each year we used the 20-year climatology (i.e., 20 years before the target year) of the driving variables. In addition, we also conducted simulation for four scenarios, in which we replaced the climate input for 2001 simulation with the $CO_2$, $T_{air}$, VPD and soil moisture from 2019, respectively. Then we used $AC_4/AC_3$ to estimate the $C_4$ grass distribution for each year or for each scenario. By calculating the difference between the $C_4$ grass distribution of four scenarios and the $C_4$ grass distribution in 2001, we quantified the contribution of $CO_2$, $T_{air}$, VPD and soil moisture to the changes in $C_4$ grass distribution.

### Remote sensing estimates of global grassland fraction

Multiple remote sensing products provide information on grassland distributions. Some directly provide continuous fraction values (i.e., GLC[86] at 100 meter and Dynamic World[87] at 10 meter) and some provide categoric information on grassland and savannas (i.e., MODIS[88] at 500 meter and ESA-CCI[89] and 300 meter). For the former, we can directly calculate the grassland fraction value at 0.5-degree resolution; for the latter, we assign the grassland/savanna type pixel to 100% and others to 0% grassland, and then obtain the mean value for each 0.5 grid cell. We found that those four estimates of grassland fraction vary considerably (Fig. S4). Based on a visual comparison of the four estimates (i.e., GLC, Dynamic World, MODIS and ESA-CCI) against vegetation map estimates[90], we found Dynamic World and ESA-CCI substantially underestimate grassland fraction. We therefore used only MODIS and GLC estimates of grassland fractions in our study.

The MODIS grassland fraction product is available from 2001 to 2019. The GLC product is only available from 2015 to 2019. To extend the GLC product back to 2001, we employed a random forest approach to estimate GLC estimates based on surface reflectance, climate, and soil type and extrapolate it to 2001 (i.e., the training accuracy is 99% and the validation accuracy is 95%). We used the average of MODIS and GLC estimates to represent the grassland fraction. To quantify the uncertainty of the approach, for each pixel we bootstrapped 1000 times between the MODIS estimate and the GLC estimate, and use the one standard deviation of these 1000 values to represent the uncertainty in grassland fraction.

### Dynamic Global Vegetation Models (DGVMs)

We used 11 DGVMs participating in the global carbon project[91](Table S1) in our study. Though all of the 11 DGVMs provided simulations for $C_4$ natural grasses, only 7 of them have simulations for $C_4$ crops (Table S1). We established an emergent constraint between $C_4$ area and $C_4$ photosynthesis contribution using the estimates from the model ensemble. We used the S3 scenario (i.e., considering elevated $CO_2$, climate change and land use change) of model simulations in our analysis.

### Emergent constraint approach

The emergent constraint technique is widely used in climate and modeling communities to infer unobserved quantities of interest in land surface processes[92,93]. The underlying assumption is that although there is a large spread in the model estimates of an observed variable X and an unobserved variable Y across models, the relationship linking the two is tightly constrained across models. Based on the strong and robust relationship across models between X and Y, observations of X can be used to generate a constraint on unobserved Y. This approach has been termed 'emergent' because the functional relationship cannot be diagnosed from a single model, but rather emerges from the spread of the model estimates. The emergent constraint identified in this study links the contribution of $C_4$ grasses/crops to total GPP to the percentages of area covered by $C_4$ grasses/crops as estimated from the ensemble of DGVM simulations.

### Reporting summary

Further information on research design is available in the Nature Portfolio Reporting Summary linked to this article.

## Data availability

The global $C_4$ vegetation distribution map is available at https://zenodo.org/records/10516423. The CS $C_4$ map was acquired from https://daac.ornl.gov/cgi-bin/dsviewer.pl?ds_id=932. The CRU TS4.02 climate data is available at https://crudata.uea.ac.uk/cru/data/hrg/, the soil moisture data can be downloaded from https://www.gleam.eu/#datasets. The global dataset of leaf photosynthetic pathway was acquired from the TRY database https://www.try-db.org/TryWeb/Home.php, by selecting those records with the field "photosynthesis pathway (traitID: 22)". The DG dataset was obtained from the supporting information of https://onlinelibrary.wiley.com/doi/10.1111/jbi.13061. The subset of the observations from the nutrient network (NutNet) are accessible at https://portal.edirepository.org/nis/mapbrowse?packageid=edi.1037.2.

## Code availability

The code for analysis is available at https://github.com/lxzswr/C4distribution/ and https://zenodo.org/records/10516423. The code of optimality photosynthesis model is available at https://github.com/zhouhaoran06/C3C4OptPhotosynthesis-. All maps in the study were generated with the assistance of the 'M_Map' package in Matlab, the instruction of which is available at https://www.eoas.ubc.ca/-rich/map.html.

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

## Acknowledgements

X.L., T.W.S., J.T. and R.Z. are supported by the NUS Presidential Young Professorship awarded to X.L. (A-0003625-03-00), a Tier II research grant from the Ministry of Education (A-8001551-00-00) and a Singapore Energy Center Core project (A-8000179-00-00). NGS acknowledges support from the National Science Foundation (DEB-2045968) and Texas Tech University. T.K. and N.G.S. also acknowledge support from LEMONTREE (Land Ecosystem Models based On New Theory, Observations, and Experiments) project, funded through the generosity of Eric and Wendy Schmidt by recommendation of the Schmidt Futures program. T.K. acknowledges additional support from a NASA Carbon Cycle Science Award 80NSSC21K1705, and the RUBISCO SFA, which is sponsored by the Regional and Global Model Analysis (RGMA) Program in the Climate and Environmental Sciences Division (CESD) of the Office of Biological and Environmental Research (BER) in the U.S. Department of Energy (DOE) Office of Science. C.J.S. and D.M.G. are grateful for support from the National Science Foundation (Award 1926431). We thank the TRY database and the global nutrient network for making their data available to support the study. X.L. thanks Prof. Graham Farquhar, Prof. Belinda Medlyn, Prof. Shuli Niu and Prof. Anthony Walker for sharing their insights on the study through several personal communications.

## Author contributions

XL conceived the idea. XL and HZ designed the study and carried out the analysis. XL, HZ, DG, TK, NS, SS and CS participated in discussions at various stages. SS provided the TRENDY v9 DGVMs simulations. JT, TS, RZ and NS aided in data collection and analysis. All authors contributed to writing.

## Competing interests

The authors declare no competing interests.
