## [Peer Review File · Nature Communications]

Mapping the global distribution of C4 vegetation using observations and optimality theoryEditorial Note: As Reviewer 1 and co-reviewer became authors after the first round of review, they were no longer consulted as reviewers.

Reviewer #1 (Remarks to the Author):

[See attached PDF]

Review of Luo et al “Global C4 distribution estimate constrained by observations and optimality theory”

General comments directed to the editor:

This manuscript presents an optimality-based estimate of global C4 GPP and distributions. Attempts are made to constrain the estimates based on satellite data and vegetation surveys. Based on their approach, the authors assess a lowering of previous global C4 GPP estimates. The central premise of this manuscript is that widely used maps of the distribution of C4 and C3 vegetation are out of date, and the authors seek to make improvements in this area. The paper presents a method for mapping C4 vegetation cover and productivity based on a C4 photosynthesis model merged to a hydraulic model and assuming some optimality principle, principally the widely used (though rarely questioned) assumption that plants maximize photosynthesis, in this case by optimizing stomatal conductance and the root/shoot allocation. There is certainly a need for different approaches to model C3 and C4 spatial distributions and for a renewed focus on the contribution of C4 grasses to regional-to-global C cycling. And there are certainly some good ideas in here that deserve to be pursued – e.g., the emergent constraints framework could be very useful as has been shown by some of the co-authors previously, but it was barely explained or justified here so in the end it added little to the paper. The authors correctly identify that the representation of C4 grasses in modeling efforts has room for improvement. And while we heartily agree new efforts are needed to better understand the biogeography and ecology of C4 grasses, this manuscript is flawed in many regards, from conception to interpretation as we detail below. We find there are multiple additional problems with the analysis and manuscript in its current form, from the framing of the new approach to the comparison of the Still et al. legacy approach to the convoluted methods to the data comparisons and ultimately to the conclusions that are drawn.

To start with, it's not at all clear that yet another GPP modeling approach is necessary to make significant progress in this area of land surface modeling and biogeography. A good summary of some of the challenges in modeling C3 and C4 grass biogeography can be found in Still et al *Glob Ecol Biogeo* (2018), a paper which is not cited in this manuscript but should have been as it directly addresses many of the issues raised here such as the extreme model-to-model differences in how DGVMs predict C4 biogeography and productivity. The central conclusion from that earlier paper is *not* that GPP modeling needs improvement – indeed few areas of land models have received more attention than photosynthesis! – but that many basic aspects of grass ecology and demography need much more attention in order to better capture C3 and C4 biogeography across spatial and temporal scales. Another paper that addresses some of the issues raised here was not cited. (Powell, Yoo and Still *Ecosphere* 2012). In this 2012 paper, updated MODIS-based remote sensing datasets of vegetation cover fraction, new landcover maps, and new crop type (i.e., harvested areas of C3 and C4 crops) maps were used, as compared to the maps of Still et al. 2003. The 2012 paper also discusses ways to add uncertainty to such land cover maps, an often-ignored aspect of global-scale work.

The implicit framing of the paper as a C3/C4 competition model leading to C3/C4 distributions is incorrect. True competition, as defined by e.g., community ecologists would imply that the model captures dynamics of resource use that reduce the availabilities of those resources below the tolerances of other plants and thereby lead to different competitive outcomes. Similar to other models, the method presented here estimates C3 and C4 productivity and a calculation of the

degree to which physiology suggests that one photosynthetic type would be favored over another in terms of relative GPP.

We list below additional major concerns and issues we see with this paper:

The comparison with Still et al. 2003. The map they downloaded and compared to represents the C4 percentage of the vegetation, NOT the C4 percentage of the land area as is apparently presented in their Fig. 1b. No mention was given of conversion from % of veg to % of area in the Methods, and again that map appears to be % of vegetation. So the comparison is literally apples to oranges. In studies that build on Still et al 2003, it is fairly common to see comparisons of a wide range potential models in the context of relevant vegetation survey data or C4 proxies that span environmental gradients at the continental scale. For example, see Murphy and Bowman (2007) or Munroe et al (2022). Finally, it's worth noting in this discussion that the Still et al. 2003 distribution map was meant to be incorporated in land models which then simulate sub-daily, or monthly and inter annual variations in productivity, transportation, respiration, C allocation, etc. There is also the grassmapR package (available at: <https://github.com/rebeccapowell/grassmapr>) which allows inter annual variation in productivity via incorporation of user-selected vegetation indices.

Validation and application of the optimality model. Optimality is an interesting way to calculate relative productivity levels for C4 plants but it cannot assess, for example, nitrogen competition (e.g., Sage and Pearcy 1987), the complexities of tree-grass dynamics (e.g., Ludwig et al 2001) and how they influence C3/C4 dynamics broadly (Cerling et al 2011, Sankaran et al 2004), or how the tree-grass dynamics vary by ecosystem and continent (including C3/C4 ratios; Lehmann et al 2014; Griffith et al 2015), the role of seasonal priority effects and other nonlinearities in plant community composition, or even light competition. Shade from tree cover often favors C3 plants in the understory in systems where there are C3 grasses in the community, but this is not the case everywhere. At the end of the day, this is a leaf-level model that has not been compared to flux data (at least no mention is given of comparisons to show how well the model works) - especially as it is principally a photosynthesis model and does not simulate ecosystem C or nutrient cycles or water cycles. Furthermore, the optimality approach does not actually capture the nuances mentioned in e.g., Reich et al Science (2018). It also does not capture the dynamics reported in Morgan et al Nature (2011) or Knapp et al PNAS (2020) which are crucial competition-based mechanisms that should greatly influence the trajectory of climate-driven distribution and productivity changes. Similarly, the general balance of CO₂, temperature, and seasonal rainfall distribution changes for determining the trajectory of C3/C4 ratios has not been solved, though the authors of this current paper claim they have done so (see further concerns below). Finally, the land models using the Still et al. 2003 or similar PFT maps predict photosynthesis and related quantities at much shorter (often sub-hourly) model time steps, as opposed to this approach which calculates GPP at growing season mean time steps (apparently – see specific comments below about model time steps) and thus misses many non-linearities between light and GPP, temperature and respiration and photorespiration, and ET and VPD, just to name a few.

The paper is seemingly motivated by the false premise that previous activities in the field are “based on a temperature-only hypothesis and lack observational constraint” (L27). The COT approach – first outlined in Collatz et al. *Oecologia* (1998) has always included a precipitation threshold of 25mm in months that meet or exceed the temperature cutoff. Various values and types of crossover temperatures were explored and compared with plot data in another paper

not cited, Griffith et al. 2015. Additionally, stable isotopes from plant, soil, and herbivore tissues have been extensively compared to vegetation plot data at regional scales in this and other papers. They have also been compared directly of Land Surface Models (Still *et al.*, 2018):

FIGURE 3 Modelled C₄ plant functional type cover in the historical modern period (ca. 1850 CE) compared with available data (soil $\delta^{13}\text{C}$, bison tissue $\delta^{13}\text{C}$ and plot data; Cotton et al., 2016; Griffith et al., 2015) in the lower right panel. Circles represent modern grazer isotope data, and diamonds represent soil isotope data [Colour figure can be viewed at wileyonlinelibrary.com]

The vast majority of these studies are not constrained to annual temperature-only methods. With regards to the crossover temperature model in particular, it is typically applied at a monthly time step to capture important seasonal variation and includes critical precipitation thresholds that screen out Mediterranean climates where lack of summer rainfall leads to nearly 100% C3 dominance. We suggest that this might explain why the optimality-based estimates provided in the NetCDF file include numerous grid cells in California with 50-80% C4 abundance. The method have been compared extensively to proxy and measurement data at continental scales in conjunction with a wide range of disturbance, soil, and other factors(e.g., Murphy & Bowman, 2007; Griffith *et al.*, 2015).

Additionally, there are significant methodological issues. The datasets used are often inappropriate for the scale of analyses or are heavily biased. Third, the manuscript presentation is very unclear, especially with regards to the critical methodological details, which compounds the other issues with the paper. Finally, and most importantly, the end product of the work is a C4 distribution that contradicts both a modern understanding of C4 biogeography as well as 47+ years of distributional data for C4 floras.

The most important point we need to address is the overall C4 distribution generated by the author’s method.

Regardless of approach, the outcome is demonstrably incorrect. On line 173, the author’s state that “The lower estimates mainly came from Africa and Australia, as the CS map suggested a large portion of the two continents have C4 dominance while we only noted a few hotspots.” We suggest that this sentence highlights very clearly the first issue

with this distribution. Taken as a whole, the herbaceous layer in Africa is known to be dominated primarily by C4 grasses. One example showing this would be the grass flora botanical maps (digitized in Lehmann *et al.*, 2019; Griffith *et al.*, 2020) which represent herbaceous vegetation strata dominated by grasses. This can be seen in map to the right. Furthermore, Australia is comprised of large savanna ecosystems covering much of the north of the country, and the patterns and drivers of C4 distributions have been studied numerous times across the entire continent (Murphy & Bowman, 2007). The Cerrado is savanna ecosystem covering a large region of South America that is known to be C4 dominated and there are large regional scale data available there too. The optimality-based map also reverses the C4 gradient in China and put an unobserved spike of C4 dominance across Kazakhstan and Mongolia likely due to dry conditions.

The manuscript makes reference numerous times to a lack of large-scale observational constraints (e.g., L98). However, many of the studies cited in their manuscript as well as additional literature provide data at continental scales which constrain C4 biogeography but which provide the same types of data assembled by this study. We describe a small sample of these datasets below after first discussing some details regarding the observation constraint created in the current manuscript.

The primary observational constraint generated in this study is grid cell C4 area % estimate based on occurrences in the TRY plant database and modified using an offset created with Nutrient Network data. The manuscript boasts a dataset of >60,000 occurrences of photosynthetic pathways but also mention in the methods that this yields only <2000 C4 samples. This yields a dataset around 3% C4 and with what appears to be almost no samples in any major tropical savanna region. This appears to be reflected in Figure 1b which shows the observational relationship relies on 4 grid cells with over 50% C4. (Which data are these? Are the area abundances based on TRY? The biomass values do not fit.) The TRY data are upscaled from individual counts of species (which the authors call abundance) to 0.5 x 0.5 degree grid cells. So, ultimately, a leaf level dataset with unknown sampling bias within cells was then filtered to herbaceous plants and spatially aggregated, producing many more temperate grid cells than tropical ones. These are essentially incomplete regional herbaceous floras.

Complete floras are a common and robust method of understanding C4 biogeography over the last 47+ years. These form a large portion of a body of literature at continental scales that all use direct or proxy measurements of C4 cover abundances to fit and validate models. Here is a sample of some of these data.

The C4 grass flora of North America (Teeri & Stowe, 1976).

Global C4 in regional grass floras (Sage *et al.*, 1999).

Furthermore, C4 are a defining feature of tropical savannas (Ratnam *et al.*, 2011). The authors mention this, but seem to suggest that there may be regional scale C3 savannas.

Tropical C4 savannas cover a significant portion of the Earth's surface (Lehmann *et al.*, 2011).

In summary, the C4 biogeography is not constrained to a mean annual climatological temperature. There are also numerous continental and global datasets with similar qualities to the flora counts from the current study. Furthermore, the floral counts developed in this study are extremely biased to temperate ecosystems, among other major issues (including large scale mismatch, see specific comments). Finally, as shown above, the end product of the work is a C4 distribution map that contradicts known C4 biogeography. The overall impression is that an optimality method has been applied to summarize the tropics based on ~4 C4 floras in

temperate regions, therefore generating a hypothesis that South American, African, and Australian savannas do not exist.

Additional specific comments directed to the authors

L26-27 – As noted above, this assertion is incorrect.

L60 – Ecosystems with a surface grassy layer that is. Grasses - and C4 grasses in particular - are extremely rare in closed canopy tropical forest understories!

L65 – Thermal tolerance is not defined relative to water demand for transpiration. Rather it is most commonly assessed by fluorescence measurements of PSII. One could argue the opposite, in fact - with lower stomatal conductance and lower transpiration rates the cooling potential is less and from that perspective C4 plants would have a lower tolerance to higher temperatures. But this whole line of reasoning is very leaf-centric and needs to include a discussion of the tropical origins of grasses that evolved C4...lots of other characteristics predispose them to warm environments aside from stomatal conductance and WUE trade-offs. See a raft of papers by Sage, Edwards, Osborne, etc.

L68 - “C4 versus C3”

L52-78 - This line of reasoning focuses on the C4 pathway in isolation from all the factors and processes and anatomies that impact distributions. It’s also entirely free of phylogenetic considerations.

L84-86 - This misrepresents the minimum precipitation threshold of 25mm in the same month which is included. And the COT proposed originally by Collatz et al. 1998 was always meant to be adjusted as a function of changing CO2. Indeed, that is much of the focus of that paper on different time scales as is reflected in its title (“Effects of climate and atmospheric CO2 partial pressure on the global distribution of C4 grasses: present, past, and future.”)!

L88-89 - This list excludes the paper by Griffith et al. 2015 comparing different COT thresholds and comparing to plot data for validation.

L107-108 - See Still et al. 2003 for inclusion of FAO crop data and modeling of C3 and C4 crop productivity and biomass. See also papers you did not cite by Still and Powell (2009) and Powell et al. (2012) that explicitly include updated and spatially explicit crop type maps based on Monfreda et al. Adding changing crop cover is useful and novel but the implication that crop types have not been included before is misleading.

L110 - This is an overstatement. Not clear why or how you “directly quantify” this. And 1992-2016 is not exactly long term by most reckonings.

L119 – Again, 1992-2016 is not exactly long term.

L130 – in what dataset, the TRY estimate?

L135, L412 – this is not an abundance.

L145 – its “Great Plains” here and throughout

L147 - Meaning C4 shrubs....or?

Figure 1. Was the Still et al. map converted from the fraction of vegetation to the fraction of area? This looks an awful like the fraction of vegetation, which is quite a different thing from the fraction of area in regions with sparse vegetation.

Figure 2. Optimality does not seem like an adequate framework to assess vegetation turnover over this time period. This implies establishment of new plant communities, in which turnover is known to be non-linear and alternatively stable (Griffith *et al.*, 2015) and in which this level of regional change does not happen at this time scale (Griffith *et al.*, 2017) but over much longer time steps (Cotton *et al.*, 2016).

L177 - "The difference is because we considered multiple climate constraints in the optimality photosynthesis model rather than a single crossover temperature (22 °C)." – This would seem to be a false equivalency.

L193-194 - There was a well-documented expansion in C4 corn area after 2005 to supply more biofuels. This also contradicts an assertion you make later in the manuscript.

L204-205 - As noted in the Methods, it's unclear how much growing season average input values would vary...and thus very likely why they drove few of your changes relative to CO2.

L227-228 - What does a p value mean in this context? Just from a linear regression? How is this an emergent constraint and not just a statistical relationship? And these values come from the DGVM outputs or your own modeling? For that matter how well does your optimal model compare to the DGVM-modeled GPP?

L233-234 - This should surprise no one especially since for croplands that are watered and fertilized and with pests removed.

L261-262 - But as noted in the Methods and in this discussion, the DGVM outputs and assumed geographies vary so wildly and widely that an ensemble mean means very little. There is a large literature on ensemble means in the climate modeling literature and whether or how to include worse performing models in ensemble statistics. Clearly not all of these DGVMs perform well or capture the right C4 grass and crop biogeography.

L287-289 - Unclear why you say this since the late Miocene global C4 expansion is not clearly tied to lowering CO2 based on most paleo CO2 proxies. Most analyses point instead to global drying and loss of forest cover and expansion of grasslands and enhanced fires.

L300 - Neither the statement (that remote sensing assumes C4 vegetation is equivalent to savanna) nor the assertion given makes any sense.

L308 - See above comments about the problems with using these data in your analysis. Also, NutNet is a proper noun.

L319-322 - There are many papers, from the old P. Hattersley work to more recent ones based on stable isotope data (e.g., Bird et al. for soil C isotopes, Murphy and Bowman for kangaroo tissue) showing a great deal of C4 grass cover/biomass across most of Australia. Why do you not cite and discuss these papers in relation to the older Still et al. paper and your new map?

L327 - See other comments about this crude screen for growing season.

L322 – “upper” maybe Northern is what was intended?

L334 – Fire is actually a more important driver of C4 vegetation in mesic savannas. Fires are not very common in arid environments (e.g., deserts) with little plant cover as they are substrate limited. Do you mean semi-arid or seasonally dry?

L339 - Which models are you referring to as the “empirical models”?

L340 - Remote sensing datasets of herbaceous cover implicitly capture aspects of fire dynamics on tree:grass structure. Also, models like the SDVGM include fire and its impact on C4 distributions (see multiple papers by Bond, Midgley and Beerling on this, as well as Higgins and Scheiter)

Figure 1f. C4 biologists would not expect that mean annual temperature would split C3 and C4 plants in this way..

L363 - But in L193-194 you note a decline of C4 crop cover in central North America!

L376 - See other comments about problems with your approach to use observations. Why did you not try to compile or compare to soil or other $\delta^{13}C$ records as a constraint on C3/C4 fractions, as these provide some of the best constraints, albeit at small spatial scales?

L384-385 - This model was developed and initially applied on paleo time scales. Has it been validated against modern grass productivity and photosynthesis datasets at leaf and canopy scales?

L400 – 2269 C4 occurrences does not seem adequate. If this is the method, why not use GBIF species occurrences or any of the C4 floras that exist?

L404 - “aggregated”

L404-408 - Doesn't this approach have to assume the TRY records are an unbiased and complete sample of vegetation at any given point? That seems highly unlikely in the vast majority of sampling locations. Also what matters is relative cover or biomass, not relative species abundance given rank abundance relationships.

L408-410 - So highly heterogeneous spatial sampling was aggregated. How? And how were model predictions aggregated? Simple averaging or by trying to weight by the TRY sample maps? Note that a 10x10 degree cell is approximately 1 million km², and so having a few hundred records, each corresponding to sampling areas of one to a few square meters in most cases, means you have dramatically undersampled your area (i.e., if you have even 1000 records in a 10x10 degree cell, that is roughly 1000 square meters in an area of 1 trillion square meters!)

Figure S1 – This figure is unclear. What are the sizes of the rings? Why would the distributions of species that report traits mean anything? Any report of a species with a known photosynthetic pathway is a record of that pathway, so why use TRY and not something like GBIF? The map of these data being converted to grid cells should be shown somewhere. There data almost completely avoids tropical savannas on 3 tropical continents.

L416 – This is a completely different scale. This paper has nothing to do with C4. Did you code the C4 identities separately and not describe that process? Alternatively, did you get the C4

from the core NutNet dataset? The reviewer is one of the two people who coded this dataset for C3 and C4 and has done analyses with these data. They do not support this proposed optimality-biogeography. There are also authorship requirements for the use of these data, and so some description of how C4 was treated in this dataset is important from several angles. Furthermore, these plots are not 25 m² as stated. Species composition is assessed in 1m² plots as quadrants of 2.5x2.5 subplots within 25 m² units within blocks (personal experience). These plots are sampled through time. How was time treated? How were floral lists calculated through time and at what scale?

L425 – There is no reason to believe that the slope at 1m² grain size will relate to scaling at 0.5 degree grid cells or larger.

L431 - “were used”

L432 this is confusing

L452-453 - But you said above you fixed the C4 fraction for the LUhv2 dataset....?

L468-469 - Where do these optimal Jmax/Vmax ratios come, from? Need to justify these values.

L471 - Based on the herbaceous fraction maps or ...?

L475-477 - Monthly time steps? Over what time periods? How was disturbance like fire accounted for since it plays such a key role in C4 grass distributions?

L480-482 - VPD has a non-linear relationship with temperature so should be calculated at the shortest possible time step and then averaged to the seasonal value (i.e. that mean will be different from a mean calculated using seasonal mean Tair and RH, as was apparently done here). Also, did you calculate daytime VPD or include nighttime? Seems important to only calculate daytime means for stomatal conductance.

L482-484 - But what about precipitation thresholds? You will include many non-productive desert areas otherwise. Why not use a remotely sensed vegetation index or SIF to define your growing season as is typically done – 10C seems arbitrary and not justified for many systems.

L482 - So what is the basic time step of the model? Mean growing season (i.e., once/year)? It uses the Farquhar et al photosynthesis model which is meant to be run at much shorter time steps. Also, is this a big leaf implementation essentially?

L486-502 - IWe question the value and utility of growing season average input values and model outputs. This completely misses all of the dynamism of fast-response physiology and biophysics. Plants do NOT respond to growing season average climates. Seems likely the input values would not differ much from year to year when averaged to the growing season.

L499-500 - This statement betrays your lack of understanding of C4 grass ecology. In fact, C4 grasses - and grasses in general - are extremely uncommon in dense forest understories.

L511 - To be clear, you did not actually simulate competition and dispersal and the many factors important for biogeography, correct? You modeled relative C gain by each type and used that to set the distribution?

L517-524 – We're not clear about the value of including TRENDY model comparisons as the models vary so greatly in what they include (Table 1 and Fig. S3) regarding C₄ that ensemble means seem, well, meaningless.

Reviewed by Chris Still and Dan Griffith

References cited

Cotton JM, Cerling TE, Hoppe KA, Mosier TM, Still CJ. 2016. Climate, CO₂, and the history of North American grasses since the Last Glacial Maximum. *Science Advances* **2**: e1501346–e1501346.

Griffith DM, Anderson TM, Osborne CP, Strömberg CAE, Forrestel EJ, Still CJ. 2015. Biogeographically distinct controls on C₃ and C₄ grass distributions: merging community and physiological ecology: Climate disequilibrium in C₄ grass distributions. *Global Ecology and Biogeography* **24**: 304–313.

Griffith DM, Cotton JM, Powell RL, Sheldon ND, Still CJ. 2017. Multi-century stasis in C₃ and C₄ grass distributions across the contiguous United States since the industrial revolution. *Journal of Biogeography*.

Griffith DM, Osborne CP, Edwards EJ, Bachle S, Beerling DJ, Bond WJ, Gallaher TJ, Helliker BR, Lehmann CER, Leatherman L, et al. 2020. Lineage-based functional types: characterising functional diversity to enhance the representation of ecological behaviour in Land Surface Models. *New Phytologist* **228**: 15–23.

Lehmann CER, Archibald SA, Hoffmann WA, Bond WJ. 2011. Deciphering the distribution of the savanna biome. *New Phytologist*: 1–13.

Lehmann CER, Griffith DM, Simpson KJ, Anderson TM, Archibald S, Beerling DJ, Bond WJ, Denton E, Edwards EJ, Forrestel EJ, et al. 2019. Functional diversification enabled grassy biomes to fill global climate space. *bioRxiv*: 583625.

Murphy BP, Bowman DMJS. 2007. Seasonal water availability predicts the relative abundance of C₃ and C₄ grasses in Australia. *Global Ecology and Biogeography* **16**: 160–169.

Powell RL, Yoo E-H, Still CJ. 2012. Vegetation and soil carbon-13 isoscapes for South America: integrating remote sensing and ecosystem isotope measurements. *Ecosphere* **3**: 1–25.

Ratnam J, Bond WJ, Fensham RJ, Hoffmann WA, Archibald S, Lehmann CER, Anderson MT, Higgins SI, Sankaran M. 2011. When is a 'forest' a savanna, and why does it matter? *Global Ecology and Biogeography* **20**: 653–660.

Sage RF, Wedin DA, Li M. 1999. The Biogeography of C₄ Photosynthesis: Patterns and Controlling Factors. In: C₄ plant biology. San Diego: Academic Press, 313–373.

Still CJ, Cotton JM, Griffith DM. 2018. Assessing earth system model predictions of C₄ grass cover in North America: From the glacial era to the end of this century. *Global Ecology and Biogeography*.

Teeri JA, Stowe LG. 1976. Climatic patterns and the distribution of C₄ grasses in North America. *Oecologia* **23**: 1–12.

Sage, R.F. and R.W. Pearcy, The Nitrogen Use Efficiency of C₃ and C₄ Plants: II. Leaf Nitrogen Effects on the Gas Exchange Characteristics of *Chenopodium album* (L.) and *Amaranthus retroflexus* (L.), *Plant Physiol*

Morgan, J. A., LeCain, D. R., Pendall, E., Blumenthal, D. M., Kimball, B. A., Carrillo, Y., ... & West, M. (2011). C₄ grasses prosper as carbon dioxide eliminates desiccation in warmed semi-arid grassland. *Nature*, 476(7359), 202-205.

Knapp, A. K., Chen, A., Griffin-Nolan, R. J., Baur, L. E., Carroll, C. J., Gray, J. E., ... & Smith, M. D. (2020). Resolving the Dust Bowl paradox of grassland responses to extreme drought. *Proceedings of the National Academy of Sciences*, 117(36), 22249-22255. *Ecology*, Volume 84, Issue 3, July 1987, Pages 959–963

Ludwig, F., de Kroon, H., Prins, H. H., & Berendse, F. (2001). Effects of nutrients and shade on tree-grass interactions in an East African savanna. *Journal of Vegetation Science*, 12(4), 579-588.

Cerling, T. E., Wynn, J. G., Andanje, S. A., Bird, M. I., Korir, D. K., Levin, N. E., ... & Remien, C. H. (2011). Woody cover and hominin environments in the past 6 million years. *Nature*, 476(7358), 51-56.

Reviewer #2 (Remarks to the Author):

This paper uses an optimality-based model of C3 and C4 photosynthesis, together with constraints derived from data and DGVMs, to predict the relative coverage of C4 plants across the globe as well as their contribution to global photosynthesis. This computation of the contribution of grasses to global photosynthesis involves three conversions:

species abundance (TRY dataset) - species area coverage (EC1* - data-based) - global C4 coverage (optimality model) - global %GPP (EC2 - DGVM-based)

* EC = Emergent Constraint

The paper addresses an important but less-recognized issue regarding the contribution of C4 grasses to global photosynthesis, which have a significant impact on the global C budget, and is especially relevant in the context of increasing droughts and fires in a changing climate.

While I think that the overall methodology is sound, and perhaps the best one can do given the available data, my main concern is that uncertainties in the second emergent constraint (used in the last step) are hard to interpret, and therefore, the results in the last section could be subject to artefacts. If I understand correctly, the second emergent constraint that converts C4 area coverage to photosynthetic contribution is based on simulations with 11 DGVMs, and each of the 11 points represents the same quantity: {global % C4 area, global % C4 GPP}. Therefore, the spread in these data points (on both axes) is not natural variation, but rather, a simulation artefact resulting from different assumptions made in the DGVMs. Consequently, how would you rule out that the constraint is not simply 1:1? Perhaps it is possible to instead derive this constraint from grid-cell-level (rather than global) %GPP~%area predictions, which would make the constraint more robust?

Minor comments:

- 1) How do you ensure that you have obtained a reasonable estimate of species richness in each gridcell? You mention the rationale of choosing a 10x10 grid to obtain sufficient records, but plotting a species-saturation curve with cell-size on the x axis would give better confidence in the richness estimates.
- 2) I would prefer to denote "species abundance" with the more common term - "species richness". Abundance usually always refers to number of individuals.
- 3) Woody species are removed from the analysis - what is the reasoning for this? I can understand this step in closed-canopy gridcells, where competition is mainly in the understorey between herbaceous C3 and grass C4. But in open ecosystems, such as open savannas, wouldn't the existence of trees (containing a major chunk of biomass) point towards a competitive advantage for C3 over C4?
- 4) Why would you expect a linear relationship for the first emergent constraint? Why wouldn't grasses dominate the system if they're competitively superior ($AC4/AC3 > 1$) and vice versa?

Reviewer #3 (Remarks to the Author):

Key results

Using a combination of databases of photosynthetic pathways and C4 vegetation area abundance, a model based on photosynthetic optimality theory and remote sensing data products of herbaceous fraction, the authors calculated a new estimate of global C4 plant distribution. The authors found that global C4 plant coverage stabilized around 11% of the vegetated land surface over the period 1992-

2016. This coverage was the net effect of both an increase in C4 crops (e.g. maize) and a decrease in C4 grasses. The authors also estimated that C4 vegetation contributed to 12.5% of the global GPP, in line with the simulations from an ensemble of DVMs. Furthermore, the authors showed that elevated CO has driven a decrease in C4 natural grass distribution over 1992-2016, with only minimal contributions by rising temperature and changing water availability.

Significance and validity

By providing a new estimate of global C4 vegetation coverage and carbon assimilation, this work has the potential to provide a significant contribution to our understanding of global C4 vegetation dynamics and their role in the global carbon cycle. Furthermore, as this work is partially based on remote sensing observations of C4 vegetation fraction and MODIS LAI (to derive sub-canopy PAR), its outcome can be useful for evaluating dynamic vegetation models, which often only use climatic drivers and land use maps as the sole observational input. Overall I find the conceptual approach of this research interesting, and valid in theory. However, I have a few concerns and questions regarding parts of the methodology behind this paper, especially regarding the validity of using site-level data to derive relationships that are then used to produce global maps (see further). These could be resolved by clarifying parts of the methods section.

Data and methodology

- It is unclear to me why the authors aggregate point records from the TRY database to a very coarse $10^{\circ}\times 10^{\circ}$ grid, and why such an approach would be valid. The authors write that this is needed to make sure there are enough samples in each grid cell to acquire a meaningful estimate of C4 abundance (L408), but I have some concerns regarding the uncertainty that is generated by this aggregation. I expect that this uncertainty will be very high for regions that have strong climatic gradients and low data coverage, such as the Sahel.
- It is also unclear from the methods section how the authors link C4 area (or species) abundance (from the TRY database) with the C4/C3 photosynthesis ratio (from the optimality model). In other words, it is unclear how Fig 1d is produced. For example, is the model output also aggregated to the $10^{\circ}\times 10^{\circ}$ grid?
- Furthermore, these data are then used to derive the evolution of C4 vegetation abundance over 1996-2016, in combination with an optimality model. However, no information is given on the time-span of the TRY data records. The underlying assumption would be that the relationship shown in Fig 1d is time-invariant, but this is nowhere justified or mentioned.
- Similarly, it is not clear why the derived relationship between C4 species abundance and C4 area abundance (Fig S2) would be valid globally, as the 37 grassland sites in this nutrient database are mainly based in North-America and Europe (ref. 38). And of course, I have the same concerns regarding the aggregation of the 73 records of C4 abundance across the U.S. It would be good to have this addressed and to have some insight in the uncertainty that comes with these assumptions.

Clarity and context

Overall I find the manuscript well written, but some parts of the methods section need some clarification. Some parts of the discussion are also a bit repetitive of the results or methods sections. See detailed comments below.

Suggested improvements

- Justify why aggregating point records from the TRY database to a $10^{\circ}\times 10^{\circ}$ grid would be a valid (and necessary?) approach. Try to quantify the uncertainty of the species abundance in each $10^{\circ}\times 10^{\circ}$ gridcell, especially for regions which have strong climatic gradients and low data coverage. Maybe also add a figure (in supplementary materials) which shows the result of this aggregation procedure, e.g. maps containing the abundance and number of records in each gridcell.

- Same suggestion for the nutrient network relationship and the biomass abundance.
- Clarify how Fig 1d is produced. How exactly did the authors couple (modelled) AC4/AC3 to (observed) area abundance? Was the model output coarse-grained to $10^{\circ} \times 10^{\circ}$ as well? If so, why don't the authors simply use the (0.5°) model grid instead? Maybe also justify why the relationships shown in Fig 1d would be time-invariant.
- The authors mention the use of a satellite-based herbaceous fraction map (L393), but this is not referenced or clearly discussed further in the methods. Maybe add a paragraph with some more details?
- Please explain into more detail what is the "emergent constraint approach" and how it was implemented, because this may not be immediately clear for all readers.

Detailed comments

- L81-84: I think these statements should be a bit more nuanced. For example, LPJ-GUESS does use bioclimatic temperature limits for tropical trees and C4 grass survival ($>15.5^{\circ}\text{C}$; Smith et al. 2014), but the simulated dominant cover type (e.g. C3 vegetation vs C4 grass) emerges from competition for resources between cohorts of plant functional types. Likewise for the Ecosystem Demography model v2.2, which does not use such bioclimatic limits (Longo et al. 2019). These are of course just two examples.
- L86 and the rest of this paragraph: Are these all distribution models? Maybe clarify which kind of model you're referring to.
- L115 and elsewhere in the text: Just a very minor comment about the language. Please don't simply write "C4" but add vegetation/grass/photosynthesis/... after it as well.
- L119: Please provide a reference for the global herbaceous fraction map.
- L122: Please clarify what an "emergent constraint approach" is and how it's implemented. Maybe in the methods section? Is this simply the slope of the linear regression in Fig 3a,b?
- L132: Just wondering, why do you use AC4/AC3 instead of $\text{AC4}/(\text{AC3}+\text{AC4})$?
- L135-138: This is already specified in the methods. Not sure if this needs to be repeated here.
- L140: Just a semantics comment, but please clarify what you mean by observational constraint. Don't you mean conversion factor, as you're not really constraining the model parameters or the model output against observations?
- L142: Minor remark: these values are not directly obvious from Fig 1. I suppose these can be obtained by averaging Fig 1 over all land area?
- L143: Just for clarity, here you applied the herbaceous fraction map (EAS-CCI) to obtain C4 grass from the C4 area abundance map, right?
- L203: It would be interesting to see if there's any difference in attribution between different regions or aridity zones. Maybe add a map in supplement for each driver driver?
- L237: Please clarify in the caption which model these figures are based on. I suppose the optimality model? Just asking because in the paragraph where this figure is referenced (L221-234) you discuss DGVM output. Perhaps it would be good to clarify that in this paragraph as well then.
- L284: Do you mean "considerable" instead of "considered"?
- L307: From here on, this paragraph mainly repeats what is already mentioned elsewhere. Consider rephrasing or removal.
- L314: Paragraph is also a bit repetitive from the results section..
- L350: Just for clarity, MODIS LAI is only used to derive understorey PAR for model input, right?
- L359-362: Are these new results? Maybe move this to the methods and results sections?
- L382: Just a minor suggestion: it can maybe be nice to add a little flowchart to accompany this overview. That will make it easier to keep track and understand how the different datasets and model are combined in this study..
- L387: Can you specify which leaf traits were used?
- L388: Please clarify how this relationship is established.
- L413: Please provide units for area abundance.
- L479: I suppose this model doesn't need any initialization? (e.g. spinup phase)

- L523: Maybe elaborate a little bit on the S3 scenario
- L571: Figure S1a needs to be improved. The figure has a very low resolution and it is not explained in the caption what the red circles in this figure mean. The light gray points are nearly invisible against the white background. Fig S1b would benefit from having density plots along the x- and y-axes, as well as a better explanation of the data used. For instance, are these the original 61588 records of photosynthetic pathways, or only the 13919 non-woody ones?
- L598: Please make these maps larger

References

Longo, M., Knox, R.G., Medvigy, D.M., Levine, N.M., Dietze, M.C., Kim, Y., Swann, A.L.S., Zhang, K., Rollinson, C.R., Bras, R.L., Wofsy, S.C., Moorcroft, P.R., 2019. The biophysics, ecology, and biogeochemistry of functionally diverse, vertically and horizontally heterogeneous ecosystems: The Ecosystem Demography model, version 2.2-Part 1: Model description. *Geoscientific Model Development* 12, 4309–4346. <https://doi.org/10.5194/gmd-12-4309-2019>

Smith, B., Wårlind, D., Arneth, A., Hickler, T., Leadley, P., Siltberg, J., Zaehle, S., 2014. Implications of incorporating N cycling and N limitations on primary production in an individual-based dynamic vegetation model. *Biogeosciences* 11, 2027–2054. <https://doi.org/10.5194/bg-11-2027-2014>

Global C₄ distribution estimate constrained by observations and optimality theory
NCOMMS-23-03867-T
Response to Reviewers

=====
We are grateful to the editor and the reviewers for their time spent evaluating our manuscript, and the constructive comments provided. We have carefully considered all the comments and have used them to improve the manuscript. Please kindly see our point-by-point response to reviewer comments below.
=====

Reviewer #1 (Remarks to the Author):

Review of Luo et al “Global C₄ distribution estimate constrained by observations and optimality theory”
General comments directed to the editor:

R1C1: This manuscript presents an optimality-based estimate of global C₄ GPP and distributions. Attempts are made to constrain the estimates based on satellite data and vegetation surveys. Based on their approach, the authors assess a lowering of previous global C₄ GPP estimates. The central premise of this manuscript is that widely used maps of the distribution of C₄ and C₃ vegetation are out of date, and the authors seek to make improvements in this area. The paper presents a method for mapping C₄ vegetation cover and productivity based on a C₄ photosynthesis model merged to a hydraulic model and assuming some optimality principle, principally the widely used (though rarely questioned) assumption that plants maximize photosynthesis, in this case by optimizing stomatal conductance and the root/shoot allocation. There is certainly a need for different approaches to model C₃ and C₄ spatial distributions and for a renewed focus on the contribution of C₄ grasses to regional-to-global C cycling. And there are certainly some good ideas in here that deserve to be pursued – e.g., the emergent constraints framework could be very useful as has been shown by some of the co-authors previously, but it was barely explained or justified here so in the end it added little to the paper. The authors correctly identify that the representation of C₄ grasses in modeling efforts has room for improvement. And while we heartily agree new efforts are needed to better understand the biogeography and ecology of C₄ grasses, this manuscript is flawed in many regards, from conception to interpretation as we detail below. We find there are multiple additional problems with the analysis and manuscript in its current form, from the framing of the new approach to the comparison of the Still et al. legacy approach to the convoluted methods to the data comparisons and ultimately to the conclusions that are drawn.

Answer: We appreciate the detailed comments from the reviewer(s), who have pointed out many aspects of the manuscript that would benefit from improved clarifications. We are particularly grateful to the reviewers for taking the time to communicate with us following the review and agreeing to co-author with us to ensure the quality of our study. The reviewers have also recommended additional data to help us better validate the optimality model.

One key point in our methodology, which emerged from our discussion with reviewer #1 and had caused many confusions in the review letter, is that the remote sensing grassland product (ESA-CCI) we used gave an egregiously low (and possibly wrong) estimate of global grassland coverage. We have therefore compared four widely used remote sensing land cover products (ESA-CCI, GLC, MODIS and Dynamic World) and their estimates of grassland fraction (Fig. S4 below). We found the ESA-CCI underestimated the global grassland fraction, particularly in C₄ abundant savanna ecosystems in the tropics. This issue consequently caused the missing of tropical savannas in our previous C₄ map. In our revised manuscript, we replaced the ESA-CCI grassland fraction map with the MODIS and GLC grassland fraction maps (also quantifying the uncertainty

incurred by their difference), and updated the C_4 map (Fig. 2a and b). The updated map includes tropical savannas and is consistent with our current understanding of C_4 biogeography, thus significantly improving our results.

Figure S4. The global grassland fraction provided by four mainstream remote sensing products. (a) ESA-CCI; (b) Copernicus GLC; (c) MODIS Land Cover and (d) Dynamic World.

Figure 2. The global distribution of C₄ vegetation and its uncertainties. The area occupied by (a) C₄ natural grasses, (c) C₄ croplands and (e) all C₄ vegetation (unit: % of the land surface). The uncertainties of the area abundance of (b) C₄ natural grasses, (d) C₄ croplands and (f) all C₄ vegetation (unit: % of the land surface).

It is important to note that the issue in our original C₄ map was solely caused by the issue of remote sensing grassland maps, and **our optimality model results are robust**. To support the argument, we added a new figure dedicated to the results from the optimality model (Fig. 1a), and the C₄ coverage estimated from the optimality model using the AC₄/AC₃ - C₄ coverage relationship (Fig. 1b and c). The C₄ coverage (% of grassland covered by C₄) clearly follows a temperature gradient, as expected, and aligns with our current understanding of C₄ biogeography (Fig. 1d).

Figure 1. C₄ natural grass coverage estimated by the optimality model. (a) the ratio of C₄ photosynthesis to C₃ photosynthesis estimated by the optimality model (AC_4/AC_3) over global non-woody regions; (b) the relationship between observed C₄ coverage (% of grassland) and estimated C₄/C₃ photosynthetic ratio by the optimality model. C₄ coverage observation obtained from difference sources (i.e., TRY, DG; please see methods); gray shaded area indicates the uncertainty range for the relationship between AC_4/AC_3 and C₄ coverage (i.e., 95% confidence interval); (c) C₄ grass coverage (% of grassland) over the globe, which can be regarded as the potential C₄ area abundance when grass covers 100% of the land surface; (f) C₄ coverage in a climate space of mean annual temperature (MAT: °C) and mean annual precipitation (MAP: mm/yr).

R1C2: To start with, it's not at all clear that yet another GPP modeling approach is necessary to make significant progress in this area of land surface modeling and biogeography. A good summary of some of the challenges in modeling C₃ and C₄ grass biogeography can be found in Still et al Glob Ecol Biogeo (2018), a paper which is not cited in this manuscript but should have been as it directly addresses many of the issues raised here such as the extreme model-to-model differences in how DGVMs predict C₄ biogeography and productivity. The central conclusion from that earlier paper is *not* that GPP modeling needs improvement – indeed few areas of land models have received more attention than photosynthesis! – but that many basic aspects of grass ecology and demography need much more attention in order to better capture C₃ and C₄

biogeography across spatial and temporal scales. Another paper that addresses some of the issues raised here was not cited. (Powell, Yoo and Still Ecosphere 2012). In this 2012 paper, updated MODIS-based remote sensing datasets of vegetation cover fraction, new landcover maps, and new crop type (i.e., harvested areas of C3 and C4 crops) maps were used, as compared to the maps of Still et al. 2003. The 2012 paper also discusses ways to add uncertainty to such land cover maps, an often-ignored aspect of global-scale work.

Answer: We fully agree with Still et al. 2018 that the problem in estimating global C₄ photosynthesis is more about the C₄ biogeography than GPP modelling (and note that Chris Still is now one of the co-authors of this study). We hope to clarify that we did not develop a new GPP model -- the optimality model is still based on the classic Collatz et al (Collatz, Ribas-Carbo, and Berry 1992) and von Caemmerer (Caemmerer 2000) framework for C₄ photosynthesis at the leaf level. Our estimation of C₄ GPP is based on a statistical approach - an emergent constraint. To clarify the point, we have added a new section "Emergent constraint approach" in Methods.

We added the two papers mentioned by the reviewer. The Still et al. 2018 was added to highlight the importance of C₄ biogeography in estimating C₄ productivity, and the Powell et al. 2012 was added in a statement about regional efforts in mapping C₄ crop.

R1C3: The implicit framing of the paper as a C3/C4 competition model leading to C3/C4 distributions is incorrect. True competition, as defined by e.g., community ecologists would imply that the model captures dynamics of resource use that reduce the availabilities of those resources below the tolerances of other plants and thereby lead to different competitive outcomes. Similar to other models, the method presented here estimates C3 and C4 productivity and a calculation of the degree to which physiology suggests that one photosynthetic type would be favored over another in terms of relative GPP.

Answer: We rephrased the wording to avoid the implicit framing of competition between C₃ and C₄ throughout the manuscript, and clarified that our assumption is that photosynthetic advantage of C₄ over C₃ can be related to C₄ coverage over large scales. This assumption was adopted by other C₄ distribution models (i.e., such as the crossover temperature hypothesis). We nevertheless agree with the reviewer that the consideration of grass phylogeny is critical to the distribution of C₄, and thus have added a new section about it in Discussion.

"We also need to highlight that optimality approach is based on the assumption that C₄ distribution is determined by the photosynthetic advantage of C₄ compared to C₃, and the photosynthetic advantage is largely dependent on local climate (Zhou et al. 2018). While this assumption is also adopted by the crossover-temperature hypothesis, it neglects the impacts of grass phylogeny on C₄ distribution. Some studies have suggested that the clades (i.e., Pooideae for C₃, and PACCMAD for C₃ and C₄) which C₄ or C₃ belong to is perhaps more critical than photosynthetic advantage for determining C₄ distribution, at least along the temperature gradient (Edwards and Still 2008; Edwards and Smith 2010; Pau, Edwards, and Still 2013). This lack of consideration on phylogeny in C₄ distribution models might impact our prediction of future C₄, as C₃ from certain clade have less competitive disadvantage to C₄ in a warmer world."

We list below additional major concerns and issues we see with this paper:

R1C4: The comparison with Still et al. 2003. The map they downloaded and compared to represents the C4 percentage of the vegetation, NOT the C4 percentage of the land area as is apparently presented in their Fig. 1b. No mention was given of conversion from % of veg to % of area in the Methods, and again that map appears to be % of vegetation. So the comparison is literally apples to oranges. In studies that build on Still et al 2003, it is fairly common to see comparisons of a wide range potential models in the context of relevant

vegetation survey data or C4 proxies that span environmental gradients at the continental scale. For example, see Murphy and Bowman (2007) or Munroe et al (2022). Finally, it's worth noting in this discussion that the Still et al. 2003 distribution map was meant to be incorporated in land models which then simulate sub-daily, or monthly and inter annual variations in productivity, transportation, respiration, C allocation, etc. There is also the grassmapR package (available at: <https://github.com/rebeccapowell/grassmapr>) which allows inter annual variation in productivity via incorporation of user-selected vegetation indices.

Answer: We apologize for this oversight. We now present the Still et al. map both in the units of % vegetated land surface and % of land surface (Fig. S5). To avoid confusion, we reported the absolute land surface area covered by C₄ – our estimate is 19.3 million km² and the Still et al. estimate is 21.1 million km². We added a new section in Discussion to indicate that our optimality model simulates at yearly time step (based on growing season climate) and is meant to provide annual baseline of C₄ distribution, while Still model is easier to be applied at month scale “Therefore, for modeling the seasonal variation of carbon fluxes from seasonal changes in C₄ distribution, the crossover temperature approach based on monthly climate could be useful here”.

R1C5: Validation and application of the optimality model. Optimality is an interesting way to calculate relative productivity levels for C4 plants but it cannot assess, for example, nitrogen competition (e.g., Sage and Pearcy 1987), the complexities of tree-grass dynamics (e.g., Ludwig et al 2001) and how they influence C3/C4 dynamics broadly (Cerling et al 2011, Sankaran et al 2004), or how the tree-grass dynamics vary by ecosystem and continent (including C3/C4 ratios; Lehmann et al 2014; Griffith et al 2015), the role of seasonal priority effects and other nonlinearities in plant community composition, or even light competition. Shade from tree cover often favors C3 plants in the understory in systems where there are C3 grasses in the community, but this is not the case everywhere. At the end of the day, this is a leaf-level model that has not been compared to flux data (at least no mention is given of comparisons to show how well the model works) - especially as it is principally a photosynthesis model and does not simulate ecosystem C or nutrient cycles or water cycles. Furthermore, the optimality approach does not actually capture the nuances mentioned in e.g., Reich et al Science (2018). It also does not capture the dynamics reported in Morgan et al Nature (2011) or Knapp et al PNAS (2020) which are crucial competition-based mechanisms that should greatly influence the trajectory of climate-driven distribution and productivity changes. Similarly, the general balance of CO₂, temperature, and seasonal rainfall distribution changes for determining the trajectory of C3/C4 ratios has not been solved, though the authors of this current paper claim they have done so (see further concerns below). Finally, the land models using the Still et al. 2003 or similar PFT maps predict photosynthesis and related quantities at much shorter (often sub- hourly) model time steps, as opposed to this approach which calculates GPP at growing season mean time steps (apparently – see specific comments below about model time steps) and thus misses many non-linearities between light and GPP, temperature and respiration and photorespiration, and ET and VPD, just to name a few.

Answer: After our discussion with the reviewer, we note that the lack of confidence in our optimality approach and the speculation of the reasons above mostly come from the underestimation of grassland fractions from remote sensing. Please see our answers to R1C1.

We added a statement in Discussion to clarify that the crossover temperature approach (Still et al.) can perform simulation at time scales shorter than year (please see our response to R1C4).

We fully agree that optimality does not capture all the processes in C₄ plants, in particular the impacts of grass phylogeny (please see our response to R1C3).

R1C6: The paper is seemingly motivated by the false premise that previous activities in the field are “based on a temperature-only hypothesis and lack observational constraint” (L27). The COT approach – first outlined in

Collatz et al. *Oecologia* (1998) has always included a precipitation threshold of 25mm in months that meet or exceed the temperature cutoff. Various values and types of crossover temperatures were explored and compared with plot data in another paper not cited, Griffith et al. 2015. Additionally, stable isotopes from plant, soil, and herbivore tissues have been extensively compared to vegetation plot data at regional scales in this and other papers. They have also been compared directly of Land Surface Models (Still *et al.*, 2018):

The vast majority of these studies are not constrained to annual temperature-only methods. With regards to the crossover temperature model in particular, it is typically applied at a monthly time step to capture important seasonal variation and includes critical precipitation thresholds that screen out Mediterranean climates where lack of summer rainfall leads to nearly 100% C3 dominance. We suggest that this might explain why the optimality-based estimates provided in the NetCDF file include numerous grid cells in California with 50-80% C4 abundance. The method have been compared extensively to proxy and measurement data at continental scales in conjunction with a wide range of disturbance, soil, and other factors(e.g., Murphy & Bowman, 2007; Griffith *et al.*, 2015).

Answer: We apologize for inaccurately describing the crossover model. We should have highlighted that the crossover temperature model (Still *et al.* 2003) considers not only temperature but also precipitation and CO₂. We have added a new statement in Introduction – “the crossover-temperature hypothesis, which assumes a particular month is determined to favor C₄ grasses when the mean daytime temperature was > 22°C and precipitation is ≥ 25 mm. This approach is based on each pathway’s relative carbon assimilation as a function of temperature, and thus the crossover temperature is dependent on atmospheric CO₂ concentration with higher crossovers at higher CO₂ levels.”

R1C7: Additionally, there are significant methodological issues. The datasets used are often inappropriate for the scale of analyses or are heavily biased. Third, the manuscript presentation is very unclear, especially with regards to the critical methodological details, which compounds the other issues with the paper. Finally, and most importantly, the end product of the work is a C4 distribution that contradicts both a modern understanding of C4 biogeography as well as 47+ years of distributional data for C4 floras.

Answer: We added a flow chart to improve the presentation of the methodology (Fig. S1). Regarding the ground dataset issue, please refer to our response to R1C9 below.

Figure S1. The workflow of estimating C₄ natural grass distribution and GPP from observations and the optimality model. Green boxes indicate observations, yellow boxes indicate models. Units and relationships between variables are highlighted in *italic*.

R1C8: *The most important point we need to address is the overall C4 distribution generated by the author’s method.*

Regardless of approach, the outcome is demonstrably incorrect. On line 173, the author’s state that “The lower estimates mainly came from Africa and Australia, as the CS map suggested a large portion of the two continents have C4 dominance while we only noted a few hotspots.” We suggest that this sentence highlights

very clearly the first issue with this distribution. Taken as a whole, the herbaceous layer in Africa is known to be dominated primarily by C4 grasses. One example showing this would be the grass flora botanical maps (digitized in Lehmann *et al.*, 2019; Griffith *et al.*, 2020) which represent herbaceous vegetation strata dominated by grasses. This can be seen in map to the right. Furthermore, Australia is comprised of large savanna ecosystems covering much of the north of the country, and the patterns and drivers of C4 distributions have been studied numerous times across the entire continent (Murphy & Bowman, 2007). The

Cerrado is savanna ecosystem covering a large region of South America that is known to be C4 dominated and there are large regional scale data available there too. The optimality-based map also reverses the C4 gradient in China and put an unobserved spike of C4 dominance across Kazakhstan and Mongolia likely due to dry conditions.

Answer: We agree with the reviewer that the original C₄ distribution included areas that go against expectations, and we have addressed the issue using high quality remote sensing-based grass fraction maps. Please see our response to R1C1.

R1C9: The manuscript makes reference numerous times to a lack of large-scale observational constraints (e.g., L98). However, many of the studies cited in their manuscript as well as additional literature provide data at continental scales which constrain C4 biogeography but which provide the same types of data assembled by this study. We describe a small sample of these datasets below after first discussing some details regarding the observation constraint created in the current manuscript.

The primary observational constraint generated in this study is grid cell C4 area % estimate based on occurrences in the TRY plant database and modified using an offset created with Nutrient Network data. The manuscript boasts a dataset of >60,000 occurrences of photosynthetic pathways but also mention in the methods that this yields only <2000 C4 samples. This yields a dataset around 3% C4 and with what appears to be almost no samples in any major tropical savanna region. This appears to be reflected in Figure 1b which shows the observational relationship relies on 4 grid cells with over 50% C4. (Which data are these? Are the area abundances based on TRY? The biomass values do not fit.) The TRY data are upscaled from individual counts of species (which the authors call abundance) to 0.5 x 0.5 degree grid cells. So, ultimately, a leaf level dataset with unknown sampling bias within cells was then filtered to herbaceous plants and spatially aggregated, producing many more temperate grid cells than tropical ones. These are essentially incomplete regional herbaceous floras.

Complete floras are a common and robust method of understanding C4 biogeography over the last 47+ years. These form a large portion of a body of literature at continental scales that all use direct or proxy measurements of C4 cover abundances to fit and validate models. Here is a sample of some of these data.

The C4 grass flora of North America (Teeri & Stowe, 1976).

Global C_4 in regional grass floras (Sage *et al.*, 1999).

Answer: We fully agree with the reviewer that regarding the original AC_4/AC_3 — C_4 coverage relationship (referred as “observational constraint” in the original manuscript but we decided to not use the term in revision), there were two potential issues – 1) the sparsity of aggregated TRY samples over space and 2) whether the observation constraint established for a 10 degree grid is applicable at 0.5 degree for global simulation (mentioned also by reviewer #2 and 3).

To address the issues, we introduced a new dataset in our analysis - the C_4 coverages collected for the contiguous United States (the DG dataset)(Griffith *et al.* 2017) (Fig. S2). The dataset directly provides C_4 coverage values on 100 km grid cells across the US, based on 40,000 plots. Interestingly, we found the AC_4/AC_3 - C_4 coverage relationship we got using the DG dataset is VERY similar to the one we had when using the TRY database (i.e., compare the blue and red dash lines in Fig.1b). This new result greatly improves the robustness of the AC_4/AC_3 - C_4 coverage relationship, as we used data of different spatial representativeness (i.e., US versus Globe) and got the same relationship, and the AC_4/AC_3 - C_4 coverage relationship is scale independent (i.e., 100 km versus 10 degree).

Figure S2. The distribution of C₄ observations used in the study. “TRY raw” means the locations where there are C₄ observations, “TRY” is the aggregated TRY raw observations at 10 degree spatial resolution (i.e., each aggregated grid cell for TRY has more than 200 C₄ observations), “DG” means the C₄ observations compiled for North America at 100 km spatial resolution.

Figure 1(b) the relationship between observed C₄ coverage (% of grassland) and estimated C₄/C₃ photosynthetic ratio by the optimality model. C₄ coverage observation obtained from difference sources (i.e., TRY, DG; please see methods); gray shaded area indicates the uncertainty range for the relationship between AC₄/AC₃ and C₄ coverage (i.e., 95% confidence interval).

R1C10: Furthermore, C₄ are a defining feature of tropical savannas (Ratnam *et al.*, 2011). The authors mention this, but seem to suggest that there may be regional scale C₃ savannas.

Fig. 3 Observed and predicted extent of C₄ savannas across Africa, Australia and South America. (a) The observed extent of savanna was mapped as a product of the classification process outlined in the Materials and Methods section. (b) The predicted extent of C₄ savannas ranges from 0% to 100% with increments of 20% as shown in the figure key. The predicted extent was calculated from the statistical models as outlined in the Results section.

Tropical C₄ savannas cover a significant portion of the Earth’s surface (Lehmann *et al.*, 2011).

Answer: Please see our response to R1C1, where we showed in our updated C₄ map that tropical savannas are fully included.

Figure 2. The global distribution of C₄ vegetation and its uncertainties. The area occupied by (a) C₄ natural grasses, (c) C₄ croplands and (e) all C₄ vegetation (unit: % of the land surface). The uncertainties of the area abundance of (b) C₄ natural grasses, (d) C₄ croplands and (f) all C₄ vegetation (unit: % of the land surface).

R1C11: In summary, the C₄ biogeography is not constrained to a mean annual climatological temperature. There are also numerous continental and global datasets with similar qualities to the flora counts from the current study. Furthermore, the floral counts developed in this study are extremely biased to temperate ecosystems, among other major issues (including large scale mismatch, see specific comments). Finally, as shown above, the end product of the work is a C₄ distribution map that contradicts known C₄ biogeography. The overall impression is that an optimality method has been applied to summarize the tropics based on ~4 C₄ floras in temperate regions, therefore generating a hypothesis that South American, African, and Australian savannas do not exist.

Answer: We acknowledge the main concerns from the reviewers, in particular the flaw in the original C₄ map. We found this flaw is solely caused by biases in the remote sensing-based grassland fraction map we used. Now we have corrected the issue – Please see our answer to R1C1. Regarding the sampling issue, please see our answer to R1C9.

Additional specific comments directed to the authors

R1C12: L26-27 – As noted above, this assertion is incorrect.

Answer: We removed the statement.

R1C13: L60 – Ecosystems with a surface grassy layer that is. Grasses - and C₄ grasses in particular - are extremely rare in closed canopy tropical forest understories!

Answer: For the tropical understory grass, we fully understand C₄ should be minimal there since there is not enough light - we calculated understory PAR mainly to maintain the spatial continuity in our model, not having to manually preclude any regions from the examination of optimality theory on C₄ - in fact the optimality

model works well there, C₄ distribution is almost 0% for tropical forest understories (Please refer to the Fig. 1a in our response to R1C1).

R1C14: L65 – Thermal tolerance is not defined relative to water demand for transpiration. Rather it is most commonly assessed by fluorescence measurements of PSII. One could argue the opposite, in fact - with lower stomatal conductance and lower transpiration rates the cooling potential is less and from that perspective C₄ plants would have a lower tolerance to higher temperatures. But this whole line of reasoning is very leaf-centric and needs to include a discussion of the tropical origins of grasses that evolved C₄...lots of other characteristics predispose them to warm environments aside from stomatal conductance and WUE trade-offs. See a raft of papers by Sage, Edwards, Osborne, etc.

Answer: We removed the statement about the thermal tolerance.

R1C15: L68 - "C₄ versus C₃"

Answer: We revised the statement to read "the relative advantage of C₄ photosynthesis to C₃ photosynthesis."

R1C16: L52-78 - This line of reasoning focuses on the C₄ pathway in isolation from all the factors and processes and anatomies that impact distributions. It's also entirely free of phylogenetic considerations.

Answer: We agree with the reviewer that our reasoning is primarily focused on the different climate sensitivities of C₄ and C₃ photosynthesis, and how these differences potentially impact the C₄ distribution.

We added a new section in Discussion to include the phylogenetic consideration:

"We also need to highlight that optimality approach is based on the assumption that C₄ distribution is determined by the photosynthetic advantage of C₄ compared to C₃, and the photosynthetic advantage is largely dependent on local climate(Zhou et al. 2018). While this assumption is also adopted by the crossover-temperature hypothesis and many C₄ distribution models, it neglects the impacts of grass phylogeny on C₄ distribution. Some studies have suggested that the clades (i.e., Pooideae for C₃, and PACCMAD for C₃ and C₄) which C₄ or C₃ belong to is perhaps more critical than photosynthetic advantage for determining C₄ distribution, at least along the temperature gradient (Edwards and Still 2008; Edwards and Smith 2010; Pau, Edwards, and Still 2013). This lack of consideration on phylogeny in C₄ distribution models might impact our prediction of future C₄, as C₃ from certain clade have less competitive disadvantage to C₄ in a warmer world."

R1C17: L84-86 - This misrepresents the minimum precipitation threshold of 25mm in the same month which is included. And the COT proposed originally by Collatz et al. 1998 was always meant to be adjusted as a function of changing CO₂. Indeed, that is much of the focus of that paper on different time scales as is reflected in its title ("Effects of climate and atmospheric CO₂ partial pressure on the global distribution of C₄ grasses: present, past, and future.")!

Answer: Following the suggestion from the reviewer, we changed our statement to "the crossover-temperature hypothesis, which assume a particular month is determined to favor C₄ grasses when the mean daytime temperature is > 22°C and precipitation in that same month is ≥ 25 mm(Still et al. 2003; Wei et al. 2014). This approach is based on each pathway's relative carbon assimilation as a function of temperature, and thus the crossover temperature is dependent on atmospheric CO₂ concentration with higher crossovers at higher CO₂ levels."

R1C18: L88-89 - This list excludes the paper by Griffith et al. 2015 comparing different COT thresholds and comparing to plot data for validation.

Answer: We added Griffith et al. 2015 in the list.

R1C19: L107-108 - See Still et al. 2003 for inclusion of FAO crop data and modeling of C₃ and C₄ crop

productivity and biomass. See also papers you did not cite by Still and Powell (2009) and Powell et al. (2012) that explicitly include updated and spatially explicit crop type maps based on Monfreda et al. Adding changing crop cover is useful and novel but the implication that crop types have not been included before is misleading. **Answer:** We added the Still et al. 2003, Still and Powell 2009 and Powell 2012 in the statement “Except a few studies that have examined the C₄ crop distribution for certain years(Still et al. 2003; R. Powell and Still 2009; R. L. Powell, Yoo, and Still 2012), changes in global C₄ crop distribution and the related contribution to global photosynthesis have yet to be evaluated.”

R1C20: L110 - This is an overstatement. Not clear why or how you “directly quantify” this. And 1992- 2016 is not exactly long term by most reckonings.

Answer: We removed “directly” and “long-term” in the statement as suggested.

R1C21: L119 – Again, 1992-2016 is not exactly long term. L130 – in what dataset, the TRY estimate?

Answer: We removed the term “long-term”, and added the names of datasets to be used in the sentence.

R1C22: L135, L412 – this is not an abundance.

Answer: We changed the term “area abundance”, to “C₄ coverage”, and added the unit of C₄ coverage “% of grassland covered by C₄”.

R1C23: L145 – its “Great Plains” here and throughout L147 - Meaning C₄ shrubs....or?

Answer: We changed “Great Prairie” to “Great Plains” throughout, and updated the original L147 to “the savannas in Africa”

R1C24: Figure 1. Was the Still et al. map converted from the fraction of vegetation to the fraction of area? This looks awful like the fraction of vegetation, which is quite a different thing from the fraction of area in regions with sparse vegetation.

Answer: We moved the Still et al maps to Fig. S5, and reported the value in both % of vegetated surface area and % of land surface area. We have also moved the comparison between the Still et al map and our map to Discussion. Please also see our response to R1C4.

Figure S5. The C₄ distribution estimated by the crossover temperature hypothesis. (a) C₄ area abundance in % of the vegetated surface; (b) C₄ area abundance in % of the land surface.

R1C25: Figure 2. Optimality does not seem like an adequate framework to assess vegetation turnover over this time period. This implies establishment of new plant communities, in which turnover is known to be non-linear and alternatively stable (Griffith *et al.*, 2015) and in which this level of regional change does not happen at this time scale (Griffith *et al.*, 2017) but over much longer time steps (Cotton *et al.*, 2016).

Answer: Please refer to our responses to R1C1, which suggest the optimality model is a reliable approach to estimate C₄ coverage...the original Figure 2 does not reflect that because it is affected by the wrong remote sensing grassland fraction map.

Additionally, we hope to clarify that we used 20-year climatology prior to each year in the optimality model to simulate AC_4/AC_3 , not the year-to-year climate. It can avoid the dramatic changes in C_4 distribution caused by climate variability and in theory allow enough time for the plant communities to develop and grow C_4 at places where C_4 has a photosynthetic advantage.

R1C26: L177 - "The difference is because we considered multiple climate constraints in the optimality photosynthesis model rather than a single crossover temperature (22 °C)." – This would seem to be a false equivalency.

Answer: We have removed the statement as suggested.

R1C27: L193-194 - There was a well-documented expansion in C_4 corn area after 2005 to supply more biofuels. This also contradicts an assertion you make later in the manuscript.

Answer: We have clarified in Discussion that globally there is a net increase in C_4 corn area – it resulted from an increase in corn area in the eastern Europe and South America, and a decrease in corn area in the U.S.

Figure S8. The changes in the area of three major C_4 crops from 2001 to 2014. (a) Maize, (b) Millet and (c) Sorghum.

R1C28: L204-205 - As noted in the Methods, it's unclear how much growing season average input values would vary...and thus very likely why they drove few of your changes relative to CO_2 .

Answer: In the revised manuscript, we now use a remote sensing-based phenology product to define growing season. We also plotted the global average of growing season T_{air} , rainfall, VPD and soil water potential (Ψ) here for your reference (Fig. R1). We found there were clear changes in growing season climate from 2001 to 2019.

Figure R1. The growing season climate (i.e., air temperature, rainfall, VPD and soil water potential (psi)) from 2001 to 2019.

R1C29: L227-228 - What does a p value mean in this context? Just from a linear regression? How is this an emergent constraint and not just a statistical relationship? And these values come from the DGVM outputs or your own modeling? For that matter how well does your optimal model compare to the DGVM-modeled GPP?

Answer: The p value here indicates the statistical strength of the emergent constraint. We added a new section in the Method to introduce the emergent constraint approach.

“Emergent constraint approach

The emergent constraint technique is widely used in climate and modelling communities to infer unobserved quantities of interest in land surface processes (Cox 2019; Eyring et al. 2019). The underlying assumption is that although there is a large spread in the model estimates of an observed variable X and an unobserved variable Y across models, the relationship linking the two is tightly constrained across models. Based on the strong and robust relationship across models between X and Y, observations of X can be used to generate a constraint on Y. This approach has been termed ‘emergent’ because the functional relationship cannot be diagnosed from a single model, but rather emerges from the spread of the model estimates. The emergent constraint identified in this study links the contribution of C₄ grasses/crops to total GPP to the percentages of area covered by C₄ grasses/crops.”

In this case, we used the output of C₄ area and C₄ GPP from 11 DGVMs. Since the optimality model predicts only leaf level maximum photosynthesis, it is not directly comparable to the outputs of DGVMs (i.e., often available at canopy scale). Nevertheless, we are interested in the photosynthetic advantage of C₄ versus C₃ (AC₄/AC₃), not the absolute magnitude of AC₄ and AC₃ in this case.

R1C30: L233-234 - This should surprise no one especially since for croplands that are watered and fertilized and with pests removed.

Answer: We are glad to see the difference between C₄ croplands and C₄ natural grasslands can be detected and quantified using the emergent constraint approach. The quantitative result (i.e., the slope of the emergent constraint relationship), despite being unsurprising, allows us to further estimate the contribution of C₄ to global GPP.

R1C31: L261-262 - But as noted in the Methods and in this discussion, the DGVM outputs and assumed

geographies vary so wildly and widely that an ensemble mean means very little. There is a large literature on ensemble means in the climate modeling literature and whether or how to include worse performing models in ensemble statistics. Clearly not all of these DGVMs perform well or capture the right C₄ grass and crop biogeography.

Answer: We fully agree with the reviewers that the DGVMs have different levels of accuracy in estimating C₄ biogeography and carbon fluxes. However, we currently lack effective tools to determine which models perform better. Considering all models used here are selected in the prestigious Global Carbon Project (Friedlingstein et al. 2020), we believe they embodied a good representation of current understanding on C₄ biogeography and carbon fluxes from the global modelling community.

Nevertheless, we did use a **drop-one bootstrapping approach** to remove a random member from the model ensemble when quantifying the slopes of the emergent constraint, and used the bootstrapped result to get the uncertainty range of the emergent constraint slopes (Fig. 4c). This source of uncertainty has been included the uncertainty of the final C₄ contribution to global GPP.

R1C32: L287-289 - Unclear why you say this since the late Miocene global C₄ expansion is not clearly tied to lowering CO₂ based on most paleo CO₂ proxies. Most analyses point instead to global drying and loss of forest cover and expansion of grasslands and enhanced fires.

Answer: We rephrased the statement to indicate there is a spatial variation in the reasons for C₄ expansion, and highlighted the role of hydroclimate in C₄ expansion in central US and Asia.

R1C33: L300 - Neither the statement (that remote sensing assumes C₄ vegetation is equivalent to savanna) nor the assertion given makes any sense.

Answer: We have removed the statement to avoid confusion. The main purpose of the statement was to explain how previous studies define C₄ area as savannas without really examining the C₄ grass characteristics. For example, MODIS defined savanna and woody savanna as “tree cover 10-30% (canopy > 2m)” and “tree cover 30-60% (canopy > 2m)” (https://lpdaac.usgs.gov/documents/101/MCD12_User_Guide_V6.pdf), with no consideration of grassland at all.

R1C34: L308 - See above comments about the problems with using these data in your analysis. Also, NutNet is a proper noun.

Answer: Please kindly see our response to R1C9, where we used additional data to validate the model. We correct the typo as suggested.

R1C35: L319-322 - There are many papers, from the old P. Hattersley work to more recent ones based on stable isotope data (e.g., Bird et al. for soil C isotopes, Murphy and Bowman for kangaroo tissue) showing a great deal of C₄ grass cover/biomass across most of Australia. Why do you not cite and discuss these papers in relation to the older Still et al. paper and your new map?

Answer: We removed the statement about Australia, as our updated C₄ map (Fig. 2a) is very much aligned with the estimates from Still et al. paper in Australia.

R1C36: L327 - See other comments about this crude screen for growing season.

Answer: We used the MODIS phenology to define the growing season and reran the simulations.

R1C37: L322 – “upper” maybe Northern is what was intended?

Answer: Yes, we changed “upper” to “Northern”.

R1C38: L334 – Fire is actually a more important driver of C₄ vegetation in mesic savannas. Fires are not very common in arid environments (e.g., deserts) with little plant cover as they are substrate limited. Do you mean

semi-arid or seasonally dry?

Answer: We changed “arid” to “semi-arid” as suggested.

R1C39: L339 - Which models are you referring to as the “empirical models”?

Answer: We updated the term to “crossover temperature C_4 model”.

R1C40: L340 - Remote sensing datasets of herbaceous cover implicitly capture aspects of fire dynamics on tree:grass structure. Also, models like the SDVGM include fire and its impact on C_4 distributions (see multiple papers by Bond, Midgley and Beerling on this, as well as Higgins and Scheiter)

Answer: We added a statement in Discussion – “However, since our approach and the crossover temperature approach (Still et al. 2003) used remote sensing-based grassland fraction maps, the impacts of fire at the annual scale might have been implicitly considered.”.

Yes, we acknowledge that some DGVMs include fire. We had in fact a statement about fire in DGVMs in Discussions. “Some DGVMs have incorporated fire-relevant processes, however, we found they estimated lower C_4 grass distribution (i.e., $9.6 \pm 6.7\%$) than those models that do not include fires (i.e., $11.8 \pm 3.2\%$).”

R1C41: Figure 1f. C_4 biologists would not expect that mean annual temperature would split C_3 and C_4 plants in this way..

Answer: Please see our response to R1C1, where we found C_4 coverage follows a temperature gradient.

R1C42: L363 - But in L193-194 you note a decline of C_4 crop cover in central North America!

Answer: Please see our response to R1C27.

R1C43: L376 - See other comments about problems with your approach to use observations. Why did you not try to compile or compare to soil or other $\delta^{13}C$ records as a constraint on C_3/C_4 fractions, as these provide some of the best constraints, albeit at small spatial scales?

Answer: Please see our response to R1C9. Additionally, we have also compared AC_4/AC_3 against $\delta^{13}C$ in bison tissues (a robust proxy for C_4 coverage) in North America (Fig. R2). The strong relationship further indicates that AC_4/AC_3 is a strong proxy for C_4 coverage.

Figure R2. The relationship between the ratio of C_4 photosynthesis to C_3 photosynthesis (AC_4/AC_3) and the $\delta^{13}C$ from bison tissues.

R1C44: L384-385 - This model was developed and initially applied on paleo time scales. Has it been validated against modern grass productivity and photosynthesis datasets at leaf and canopy scales?

Answer: The optimality model includes key modules FvCB for C_3 and von Caemmerer (2000) for C_4 , which are widely used and validated in photosynthesis studies. We do not expect the time scale to be an issue here considering the photosynthesis principle is the same over time. We parameterized the model using contemporary measurements of C_4 species, which can represent the current condition. In addition, we are interested in the ratio (AC_4/AC_3) rather than the absolute magnitude of AC_4 and AC_3 .

R1C45: L400 – 2269 C4 occurrences does not seem adequate. If this is the method, why not use GBIF species occurrences or any of the C4 floras that exist?

Answer: Please see our response to R1C9.

R1C46: L404 - “aggregated”

Answer: We corrected the typo as suggested.

R1C47: L404-408 - Doesn't this approach have to assume the TRY records are an unbiased and complete sample of vegetation at any given point? That seems highly unlikely in the vast majority of sampling locations. Also what matters is relative cover or biomass, not relative species abundance given rank abundance relationships.

L408-410 - So highly heterogeneous spatial sampling was aggregated. How? And how were model predictions aggregated? Simple averaging or by trying to weight by the TRY sample maps? Note that a 10x10 degree cell is approximately 1 million km², and so having a few hundred records, each corresponding to sampling areas of one to a few square meters in most cases, means you have dramatically undersampled your area (i.e., if you have even 1000 records in a 10x10 degree cell, that is roughly 1000 square meters in an area of 1 trillion square meters!)

Answer: Please see our response to R1C9.

R1C48: Figure S1 – This figure is unclear. What are the sizes of the rings? Why would the distributions of species that report traits mean anything? Any report of a species with a known photosynthetic pathway is a record of that pathway, so why use TRY and not something like GBIF? The map of these data being converted to grid cells should be shown somewhere. There data almost completely avoids tropical savannas on 3 tropical continents.

Answer: We have removed Figure S1 and added a new figure (Fig. S2) to show the location of samples. Please also see our response to R1C9.

R1C49: L416 – This is a completely different scale. This paper has nothing to do with C4. Did you code the C4 identities separately and not describe that process? Alternatively, did you get the C4 from the core NutNet dataset? The reviewer is one of the two people who coded this dataset for C3 and C4 and has done analyses with these data. They do not support this proposed optimality-biogeography. There are also authorship requirements for the use of these data, and so some description of how C4 was treated in this dataset is important from several angles.

Furthermore, these plots are not 25 m² as stated. Species composition is assessed in 1m² plots as quadrants of 2.5x2.5 subplots within 25 m² units within blocks (personal experience). These plots are sampled through time. How was time treated? How were floral lists calculated through time and at what scale?

Answer: We acquired an open subset of NutNet data from the link <https://portal.edirepository.org/nis/mapbrowse?packageid=edi.1037.2>. We did not use the Nutnet observations when establishing the AC₄/AC₃ - C₄ coverage relationship (Fig. 1b), but only used it to infer an empirical relationship between C₄ species richness and C₄ coverage (Fig. S3). We apologize for the confusions here, and added a flow chart (Figure. S1) to illustrate how we used the different observations in the study.

R1C50: L425 – There is no reason to believe that the slope at 1m² grain size will relate to scaling at 0.5 degree grid cells or larger.

Answer: After discussing with the reviewer, we deem that NutNet could be representative considering the plots are often set up at homogeneous grassland landscapes and we only used control plots without added fertilizer. Furthermore, the similarity in the AC₄/AC₃ - C₄ coverage relationship when using DG dataset and TRY dataset (the latter must convert species richness to C₄ coverage using the NutNet data) gave us confidence

about the approach. Please also see Fig.1b in our response to R1C9.

R1C51: L431 - "were used" L432 this is confusing

Answer: We have deleted the paragraph as the content on biomass abundance is a bit distracting.

R1C52: L452-453 - But you said above you fixed the C4 fraction for the LUHv2 dataset....?

Answer: The LUHv2 used a fixed C4 crop fraction (Monfreda, Ramankutty, and Foley 2008) throughout years. Here we are introducing a new dataset that does not use fixed C4 crop fraction (Jackson et al. 2019), and the data served as an independent validation to the results from LUHv2.

R1C53: L468-469 - Where do these optimal J_{max}/V_{max} ratios come, from? Need to justify these values. L471 - Based on the herbaceous fraction maps or ...?

Answer: These values were acquired from measurements for nearly 40 C_3 and C_4 grasses in a recent study (Zhou, Akçay, and Helliker 2023) and these ratios were also validated by more measurements and theoretical analyses on optimal J_{max}/V_{cmax} for C_3 and C_4 (Pignon and Long 2020; Zhou, Akçay, and Helliker 2023). We have added these references in the manuscript to justify the choice. We did not apply grassland fraction maps when running the optimality model.

R1C54: L475-477 - Monthly time steps? Over what time periods? How was disturbance like fire accounted for since it plays such a key role in C_4 grass distributions?

Answer: We produced the map on annual time step. We did not consider the role of fire, but had a section on fire in Discussion. Please also see our response to R1C40.

R1C55: L480-482 - VPD has a non-linear relationship with temperature so should be calculated at the shortest possible time step and then averaged to the seasonal value (i.e., that mean will be different from a mean calculated using seasonal mean T_{air} and RH, as was apparently done here). Also, did you calculate daytime VPD or include nighttime? Seems important to only calculate daytime means for stomatal conductance.

Answer: We redid the analysis as the reviewer suggested. The VPD was calculated based on daytime temperature.

R1C56: L482-484 - But what about precipitation thresholds? You will include many non-productive desert areas otherwise. Why not use a remotely sensed vegetation index or SIF to define your growing season as is typically done – 10C seems arbitrary and not justified for many systems.

Answer: We did not apply any precipitation threshold when running the optimality model. We agree with the reviewer, and have rerun the model using the MODIS phenology product.

R1C57: L482 - So what is the basic time step of the model? Mean growing season (i.e., once/year)? It uses the Farquhar et al photosynthesis model which is meant to be run at much shorter time steps. Also, is this a big leaf implementation essentially?

Answer: Our approach provides annual simulation of C_4 area using growing season climate. We added relevant clarifications in Methods and Discussion. Since we are only interested in the ratio AC_4/AC_3 , the scale (leaf vs canopy) would have limited influence on the result as leaf area be cancelled out in the ratio.

R1C58: L486-502 - We question the value and utility of growing season average input values and model outputs. This completely misses all of the dynamism of fast-response physiology and biophysics. Plants do NOT respond to growing season average climates. Seems likely the input values would not differ much from year to year when averaged to the growing season.

Answer: Please see our response to R1C28, where we show there was considerable change in climate.

R1C59: L499-500 - This statement betrays your lack of understanding of C4 grass ecology. In fact, C4 grasses - and grasses in general - are extremely uncommon in dense forest understories.

Answer: We understand that in understories of dense forests there is unlikely to be C₄ plants. Please see our responses to R1C13.

R1C60: L511 - To be clear, you did not actually simulate competition and dispersal and the many factors important for biogeography, correct? You modeled relative C gain by each type and used that to set the distribution?

Answer: We have updated the statement to avoid confusion – it now reads “To model the growing season AC₄/AC₃ from 2001 to 2019.”

R1C61: L517-524 – We’re not clear about the value of including TRENDY model comparisons as the models vary so greatly in what they include (Table 1 and Fig. S3) regarding C4 that ensemble means seem, well, meaningless.

Answer: The wide range of estimates for C₄ area and C₄ GPP from TRENDY DGVMs indicate C₄ process is a main source of uncertainty in carbon cycle simulations, and it is critical for us to narrow down the uncertainty. In this study we used an emergent constraint approach to do so. Please also see our answer to R1C29.

Reviewed by Chris Still and Dan Griffith

References cited

Cotton JM, Cerling TE, Hoppe KA, Mosier TM, Still CJ. 2016. Climate, CO₂, and the history of North American grasses since the Last Glacial Maximum. *Science Advances* 2: e1501346–e1501346.

Griffith DM, Anderson TM, Osborne CP, Strömberg CAE, Forrester EJ, Still CJ. 2015. Biogeographically distinct controls on C₃ and C₄ grass distributions: merging community and physiological ecology: Climate disequilibrium in C₄ grass distributions. *Global Ecology and Biogeography* 24: 304–313.

Griffith DM, Cotton JM, Powell RL, Sheldon ND, Still CJ. 2017. Multi-century stasis in C₃ and C₄ grass distributions across the contiguous United States since the industrial revolution. *Journal of Biogeography*.

Griffith DM, Osborne CP, Edwards EJ, Bachle S, Beerling DJ, Bond WJ, Gallaher TJ, Helliker BR, Lehmann CER, Leatherman L, et al. 2020. Lineage-based functional types: characterising functional diversity to enhance the representation of ecological behaviour in Land Surface Models. *New Phytologist* 228: 15–23.

Lehmann CER, Archibald SA, Hoffmann WA, Bond WJ. 2011. Deciphering the distribution of the savanna biome. *New Phytologist*: 1–13.

Lehmann CER, Griffith DM, Simpson KJ, Anderson TM, Archibald S, Beerling DJ, Bond WJ, Denton E, Edwards EJ, Forrester EJ, et al. 2019. Functional diversification enabled grassy biomes to fill global climate space. *bioRxiv*: 583625.

Murphy BP, Bowman DMJS. 2007. Seasonal water availability predicts the relative abundance of C₃ and C₄ grasses in Australia. *Global Ecology and Biogeography* 16: 160–169.

Powell RL, Yoo E-H, Still CJ. 2012. Vegetation and soil carbon-13 isoscapes for South America: integrating remote sensing and ecosystem isotope measurements. *Ecosphere* 3: 1–25.

Ratnam J, Bond WJ, Fensham RJ, Hoffmann WA, Archibald S, Lehmann CER, Anderson MT, Higgins SI, Sankaran M. 2011. When is a 'forest' a savanna, and why does it matter? *Global Ecology and Biogeography* 20: 653–660.

Sage RF, Wedin DA, Li M. 1999. The Biogeography of C4 Photosynthesis: Patterns and Controlling Factors. In: *C4 plant biology*. San Diego: Academic Press, 313–373.

Still CJ, Cotton JM, Griffith DM. 2018. Assessing earth system model predictions of C4 grass cover in North America: From the glacial era to the end of this century. *Global Ecology and Biogeography*.

Teeri JA, Stowe LG. 1976. Climatic patterns and the distribution of C4 grasses in North America.

Oecologia 23: 1–12.

Sage, R.F. and R.W. Pearcy, The Nitrogen Use Efficiency of C3 and C4 Plants: II. Leaf Nitrogen Effects on the Gas Exchange Characteristics of *Chenopodium album* (L.) and *Amaranthus retroflexus* (L.), *Plant Physiol*

Morgan, J. A., LeCain, D. R., Pendall, E., Blumenthal, D. M., Kimball, B. A., Carrillo, Y., ... & West, M. (2011). C4 grasses prosper as carbon dioxide eliminates desiccation in warmed semi- arid grassland. *Nature*, 476(7359), 202-205.

Knapp, A. K., Chen, A., Griffin-Nolan, R. J., Baur, L. E., Carroll, C. J., Gray, J. E., ... & Smith, M.

D. (2020). Resolving the Dust Bowl paradox of grassland responses to extreme drought. *Proceedings of the National Academy of Sciences*, 117(36), 22249-22255. *Physiology*, Volume 84, Issue 3, July 1987, Pages 959–963

Ludwig, F., de Kroon, H., Prins, H. H., & Berendse, F. (2001). Effects of nutrients and shade on tree-grass interactions in an East African savanna. *Journal of Vegetation Science*, 12(4), 579- 588.

Cerling, T. E., Wynn, J. G., Andanje, S. A., Bird, M. I., Korir, D. K., Levin, N. E., ... & Remien, C.

H. (2011). Woody cover and hominin environments in the past 6 million years. *Nature*, 476(7358), 51-56.

=====
Reviewer #2 (Remarks to the Author):

This paper uses an optimality-based model of C3 and C4 photosynthesis, together with constraints derived from data and DGVMs, to predict the relative coverage of C4 plants across the globe as well as their contribution to global photosynthesis. This computation of the contribution of grasses to global photosynthesis involves three conversions:

species abundance (TRY dataset) - species area coverage (EC1* - data-based) - global C4 coverage (optimality model) - global %GPP (EC2 - DGVM-based)

* EC = Emergent Constraint

The paper addresses an important but less-recognized issue regarding the contribution of C4 grasses to global photosynthesis, which have a significant impact on the global C budget, and is especially relevant in the context of increasing droughts and fires in a changing climate.

Answer: We appreciate the positive feedback from the reviewer and for recognizing the importance of the study. We have carefully followed the comments from the reviewer to improve the manuscript.

R2C1: While I think that the overall methodology is sound, and perhaps the best one can do given the available data, my main concern is that uncertainties in the second emergent constraint (used in the last step) are hard to interpret, and therefore, the results in the last section could be subject to artefacts. If I understand correctly, the second emergent constraint that converts C4 area coverage to photosynthetic contribution is based on simulations with 11 DGVMs, and each of the 11 points represents the same quantity: {global % C4 area, global % C4 GPP}. Therefore, the spread in these data points (on both axes) is not natural variation, but rather, a simulation artefact resulting from different assumptions made in the DGVMs. Consequently, how

would you rule out that the constraint is not simply 1:1? Perhaps it is possible to instead derive this constraint from grid-cell-level (rather than global) %GPP~%area predictions, which would make the constraint more robust?

Answer: We agree with the reviewer that the emergent constraint approach is based on the spread of simulations from 11 DGVMs. The variation in the simulations is primarily caused by the different parameterization and structures of models regarding C₄ distribution and photosynthesis, not caused by the natural variation of these processes (nor did we expect to). It is important to note however that we do not assume a 1:1 relationship, but instead assume that the relationship is well described by a linear function, with support provided by the low p value (see also our response to R2C5 where we test the impact of the assumption of linearity). Though there was uncertainty in the linear relationship across models (Fig. 3), but the powerful aspect of the emergent constraint technique is that this uncertainty is propagated directly through to the uncertainty on the constrained target variable. We have added the following section in Methods to clarify the validity of using emergent constraint approach, in particular when we are unsure which model gives good or bad representations of C₄ distribution and photosynthesis.

“Emergent constraint approach

The Emergent constraint technique is widely used in climate and modelling communities to infer unobserved quantities of interest in land surface processes (Cox 2019; Eyring et al. 2019). The underlying assumption is that although there is a large spread in the model estimates of an observed variable X and an unobserved variable Y across models, the relationship linking the two is tightly constrained across models. Based on the strong and robust relationship across models between X and Y, observations of X can be used to generate a constraint on Y. This approach has been termed ‘emergent’ because the functional relationship cannot be diagnosed from a single model, but rather emerges from the spread of the model estimates. The emergent constraint identified in this study links the contribution of C₄ grasses/crops to total GPP to the percentages of area covered by C₄ grasses/crops.”

Following the suggestion of the reviewer, we conducted a grid-cell level emergent constraint (EC) analysis (Fig. S9). Our result showed that on pixels where there is a significant relationship between % C₄ area and % C₄ GPP, the slope of the ECs distributed in a small range of 0.6-1.5. The mean slope for C₄ grass is 1.11, for C₄ crop is 1.2 – similar to what we have acquired when using the global total values (Fig. 4).

Figure S9. The emergent constraint at the pixel level. The slopes of the emergent constraint for (a) C₄ grass and (b) C₄ crop. We only show the slopes there the relationship between C₄ area abundance and C₄ GPP

percentage are significant ($p < 0.05$). The numbers of DGVMs that have simulations for (c) C_4 grass and (d) C_4 crop.

However, we cannot directly apply this grid cell-level ECs to estimate global C_4 GPP%, as the EC can only be established for certain pixels (not including all the C_4 pixels in our C_4 map). This is further challenged by the fact that many pixels only have a few DGVM simulations (Fig. S9 c and d). Therefore, in order to infer the C_4 global photosynthesis, we had to rely on the global scale simulations. We have added the following statement in Results. “We also conducted a grid cell level emergent constraint analysis and acquired similar ranges of slopes”

Minor comments:

R2C2: 1) How do you ensure that you have obtained a reasonable estimate of species richness in each gridcell? You mention the rationale of choosing a 10x10 grid to obtain sufficient records, but plotting a species-saturation curve with cell-size on the x axis would give better confidence in the richness estimates.

Answer: In our study, we used estimated photosynthetic ratio of C_4 to C_3 to infer the C_4 coverage (% C_4 of grasslands) of each grid cell. We fully agree with the reviewer since this relationship is established on a 10 degree grid, the relationship may be not applicable to the grid cell we study (0.5 degree). The reviewers suggest another method, that is the extrapolate the relationship by cell-size, however, we found it is difficult to do so for the TRY database due to the limited samples.

To address the issues, we introduced a new dataset - the C_4 coverages collected for the contiguous United States (the DG dataset)(Griffith et al. 2017) (Fig. S2). The dataset directly provides C_4 coverage values on 100 km grid cells across the US, based on 40,000 plots. Interestingly, we found the AC_4/AC_3 - C_4 coverage relationship we built using the DG dataset is VERY similar to the one we had when using the TRY database (i.e., compare the blue and red dash lines in Fig.1b). The new result greatly improved the robustness of our approach, as we used data of different spatial representativeness (i.e., US versus Globe) and got the same relationship, and the AC_4/AC_3 - C_4 coverage relationship is scale independent (i.e., 100 km versus 10 degree).

Figure S2. The distribution of C_4 observations used in the study. “TRY raw” means the locations where there are C_4 observations, “TRY” is the aggregated TRY raw observations at 10 degree spatial resolution (i.e., each aggregated grid cell for TRY has more than 200 C_4 observations), “DG” means the C_4 observations compiled for North America at 100 km spatial resolution.

Figure 1(b) the relationship between observed C_4 coverage (% of grassland) and estimated C_4/C_3 photosynthetic ratio by the optimality model. C_4 coverage observation obtained from difference sources (i.e., TRY, DG; please see methods); gray shaded area indicates the uncertainty range for the relationship between AC_4/AC_3 and C_4 coverage (i.e., 95% confidence interval).

R2C3: 2) I would prefer to denote "species abundance" with the more common term - "species richness". Abundance usually always refers to number of individuals.

Answer: Thank you. We changed "species abundance" to "species richness" throughout the manuscript.

R2C4: 3) Woody species are removed from the analysis - what is the reasoning for this? I can understand this step in closed-canopy gridcells, where competition is mainly in the understorey between herbaceous C_3 and grass C_4 . But in open ecosystems, such as open savannas, wouldn't the existence of trees (containing a major chunk of biomass) point towards a competitive advantage for C_3 over C_4 ?

Answer: We apologize for the confusions due to the lack of details in our method. Our study is aimed at acquiring the distribution of C_4 plants, which are almost all grass species (Sage and Sultman 2016) and therefore only exist in the grassland fraction of each pixel. In our method, as long as a pixel has a fraction of grassland (including open or woody savannas, forest understoreys), we used the optimality model to estimate its C_4 coverage (% of grassland covered by C_4) in the pixel. We did not assess the competitive advantage of grasslands (either C_4 or C_3) versus non-grasslands (C_3 woody) in each pixel, as the fraction of grassland/non-grassland was directly prescribed by remote sensing observations.

We added a flow chart (Fig. S1) in our study to better illustrate our method.

Figure S1. The workflow of estimating C₄ natural grass distribution and GPP from observations and the optimality model. Green boxes indicate observations, yellow boxes indicate models. Units and relationships between variables are highlighted in italic.

R2C5: 4) Why would you expect a linear relationship for the first emergent constraint? Why wouldn't grasses dominate the system if they're competitively superior ($AC_4/AC_3 > 1$) and vice versa?

Answer: Following the reviewer's comment, we tested multiple linear and non-linear models to fit the C₄ coverage to AC_4/AC_3 . Statistics of the models, including AIC, R² and RMSE, suggest that the linear model is strong enough to describe the relationship (Table R1) – with increase in AC_4/AC_3 , there is an increase in C₄ species richness (+%) and C₄ coverage (+%). Interestingly, we barely see a place that has AC_4/AC_3 greater than 2.5 (Fig. 1a), and that happen to be the point where C₄ coverage approaches 100%.

Table R1. The statistics of three models used to fit AC_4/AC_3 to C₄ coverage.

Type of models	Equation	R ²	RMSE	AIC
Linear	$f(x) = 0.68 * x - 0.54$	0.51	0.24	-187.5
Polynomial (Degree = 2)	$f(x) = -0.26 * x^2 + 1.47 * x - 1.12$	0.52	0.24	-184.0
Best fitting model	$-0.46 * (\sin(x - \pi)) + -0.04 * ((x - 10)^2) + 2.93 * (1)$	0.53	0.24	-183.9

=====

Reviewer #3 (Remarks to the Author):

Key results

R3C1: Using a combination of databases of photosynthetic pathways and C4 vegetation area abundance, a model based on photosynthetic optimality theory and remote sensing data products of herbaceous fraction, the authors calculated a new estimate of global C4 plant distribution. The authors found that global C4 plant coverage stabilized around 11% of the vegetated land surface over the period 1992-2016. This coverage was the net effect of both an increase in C4 crops (e.g. maize) and a decrease in C4 grasses. The authors also estimated that C4 vegetation contributed to 12.5% of the global GPP, in line with the simulations from an ensemble of DVMs. Furthermore, the authors showed that elevated CO₂ has driven a decrease in C4 natural grass distribution over 1992-2016, with only minimal contributions by rising temperature and changing water availability.

Significance and validity

By providing a new estimate of global C4 vegetation coverage and carbon assimilation, this work has the potential to provide a significant contribution to our understanding of global C4 vegetation dynamics and their role in the global carbon cycle. Furthermore, as this work is partially based on remote sensing observations of C4 vegetation fraction and MODIS LAI (to derive sub-canopy PAR), its outcome can be useful for evaluating dynamic vegetation models, which often only use climatic drivers and land use maps as the sole observational input. Overall I find the conceptual approach of this research interesting, and valid in theory. However, I have a few concerns and questions regarding parts of the methodology behind this paper, especially regarding the validity of using site-level data to derive relationships that are then used to produce global maps (see further). These could be resolved by clarifying parts of the methods section.

Answer: We thank the reviewer for the encouraging comments and have endeavored to use their comments to improve the manuscript.

Data and methodology

R3C2: - It is unclear to me why the authors aggregate point records from the TRY database to a very coarse 10°x10° grid, and why such an approach would be valid. The authors write that this is needed to make sure there are enough samples in each grid cell to acquire a meaningful estimate of C4 abundance (L408), but I have some concerns regarding the uncertainty that is generated by this aggregation. I expect that this uncertainty will be very high for regions that have strong climatic gradients and low data coverage, such as the Sahel.

Answer: We agree with the reviewer that only using TRY database is not ideal, considering we can only aggregate them to 10 degrees to establish the $AC_4/AC_3 - C_4$ coverage relationship. To test whether the relationship and its associated uncertainty is scale dependent, we introduced a new dataset - the C₄ coverages collected for the contiguous United States (the DG dataset)(Griffith et al. 2017). The dataset directly provides C₄ coverage values on 100 km grid cells across the US, based on 40,000 plots. Interestingly, we found the $AC_4/AC_3 - C_4$ coverage relationship we built using the DG dataset is VERY similar to the one we had when using the TRY database (i.e., compare the blue and red dash lines in Fig.1b). The new result greatly enhanced the robustness of our approach, as we used data of different spatial representativeness (i.e., US versus Globe) and different spatial resolutions (i.e., 100 km versus 10 degree), but got the same $AC_4/AC_3 - C_4$ coverage relationship.

Figure 1(b) the relationship between observed C_4 coverage (% of grassland) and estimated C_4/C_3 photosynthetic ratio by the optimality model. C_4 coverage observation obtained from difference sources (i.e., TRY, DG; please see methods); gray shaded area indicates the uncertainty range for the relationship between AC_4/AC_3 and C_4 coverage (i.e., 95% confidence interval).

Based on the new analysis, we have updated the uncertainty in our C_4 grassland map (Fig. 2a and b), which shows larger uncertainty in parts of Sahel as the reviewer expected. Part of the reason is the relatively limited samples under the very high AC_4/AC_3 conditions there.

Figure 2. The global distribution of C_4 vegetation and its uncertainties. The area occupied by (a) C_4 natural grasses, (c) C_4 croplands and (e) all C_4 vegetation (unit: % of the land surface). The uncertainties of the area abundance of (b) C_4 natural grasses, (d) C_4 croplands and (f) all C_4 vegetation (unit: % of the land surface).

R3C3: - It is also unclear from the methods section how the authors link C_4 area (or species) abundance (from the TRY database) with the C_4/C_3 photosynthesis ratio (from the optimality model). In other words, it is unclear how Fig1d is produced. For example, is the model output also aggregated to the $10^\circ \times 10^\circ$ grid?

Answer: We apologize for the lack of clarity in the method description. Regarding this specific question, we obtained AC_4/AC_3 firstly at 0.5 degree (as climate input is at 0.5 degree) and then aggregated that to 10 degree when comparing with the TRY database observations. We have added a statement to clarify that in Methods – “When establishing the AC_4/AC_3 - C_4 coverage relationship (Fig.1b) from the TRY and DG datasets, we aggregated the simulations from 0.5 degree to 10 degree and 1 degree (approximately to 100 km).”

We have added a flow chart (Fig. S1) in our study to better illustrate our method.

Figure S1. The workflow of estimating C_4 natural grass distribution and GPP from observations and the optimality model. Green boxes indicate observations, yellow boxes indicate models. Units and relationships between variables are highlighted in italic.

R3C4: - Furthermore, these data are then used to derive the evolution of C_4 vegetation abundance over 1996-2016, in combination with an optimality model. However, no information is given on the time-span of the TRY data records. The underlying assumption would be that the relationship shown in Fig 1d is time-invariant, but this is nowhere justified or mentioned.

Answer: We added the in Methods that “the time range of TRY observations covers the past 50 years” and “The DG dataset... was sampled from roughly 40,000 plots over the past 40 years”.

Considering that only limited number of observations have time stamps, it is challenging to test whether the relationship changes with time. Alternatively, we did a simple bootstrapping test -- We randomly removed 10% of the samples, assuming they came from different time periods, then examined whether the relationship would change (Fig. R3). We found our test has minimal impacts on the AC_4/AC_3 - C_4 coverage relationship, and the impact is not greater than the uncertainty range we already quantified (as the confidence interval of the linear regression). This test implies the relationship is potentially time invariant.

Figure R3. Bootstrapping test of the relationship between AC_4/AC_3 and C_4 coverage. (Left) We randomly removed 10% of the samples and rebuilt linear the relationship. We tested 100 times; (Right) Same as Fig. 1b, the relationship used in our study.

In theory, we would also argue the relationship is unlikely to change with time, as the photosynthesis advantage of C_4 over C_3 is essentially determined by leaf anatomy, which does not change over the time scale of our study. Nevertheless, we have added in Methods that “we assume the relationship is time-invariant”.

R3C5: - Similarly, it is not clear why the derived relationship between C_4 species abundance and C_4 area abundance (Fig S2) would be valid globally, as the 37 grassland sites in this nutrient database are mainly based in North-America and Europe (ref. 38). And of course, I have the same concerns regarding the aggregation of the 73 records of C_4 abundance across the U.S. It would be good to have this addressed and to have some insight in the uncertainty that comes with these assumptions.

Answer: We removed the part related to the C_4 biomass abundance, since it caused some distractions.

We also added a figure (Fig. S2) to show the distribution of NutNet observations, and it covers not only North America and Europe, but also has representations in Australia, Asia and to a less extent in Africa and South America. We hope to clarify that we did not use NutNet observations when establishing the $AC_4/AC_3 - C_4$ coverage relationship (Fig. 1b), but only used it to infer an empirical relationship between C_4 species richness and C_4 coverage (Fig. S3). We apologize for the confusions here, and added a flow chart to illustrate how we used the different observations in the study (Fig. S1).

Figure S2. The distribution of C_4 observations used in the study. “TRY raw” means the locations where there are C_4 observations, “TRY” is the aggregated TRY raw observations at 10 degree spatial resolution (i.e., each aggregated grid cell for TRY has more than 200 C_4 observations), “DG” means the C_4 observations compiled for North America at 100 km spatial resolution.

Additionally, our result on the $AC_4/AC_3 - C_4$ coverage relationship (Fig. 1b) potentially suggests that the C_4 richness – C_4 coverage relationship (Fig. S3) we got from NutNet is reasonable, because in Fig. 1b, the relationships solely based on TRY and DG are very similar (i.e., red and blue dash lines). Since the C_4 richness – C_4 coverage relationship was only used for the TRY dataset to convert species richness to C_4 coverage, while the DG dataset directly provided C_4 coverage, the similarity of the relationships of TRY and DG in Fig. 1b indicates the effectiveness of the conversion factor (Fig. S3) we got from NutNet.

Figure 1(b) the relationship between observed C_4 coverage (% of grassland) and estimated C_4/C_3 photosynthetic ratio by the optimality model. C_4 coverage observation obtained from difference sources (i.e., TRY, DG; please see methods); gray shaded area indicates the uncertainty range for the relationship between AC_4/AC_3 and C_4 coverage (i.e., 95% confidence interval).

Clarity and context

R3C6: Overall I find the manuscript well written, but some parts of the methods section need some clarification. Some parts of the discussion are also a bit repetitive of the results or methods sections. See detailed comments below.

Answer: Thank you for the comment, we have improved the manuscript following your suggestions.

Suggested improvements

R3C7: - Justify why aggregating point records from the TRY database to a $10^\circ \times 10^\circ$ grid would be a valid (and necessary?) approach. Try to quantify the uncertainty of the species abundance in each $10^\circ \times 10^\circ$ gridcell, especially for regions which have strong climatic gradients and low data coverage. Maybe also add a figure (in supplementary materials) which shows the result of this aggregation procedure, e.g. maps containing the abundance and number of records in each gridcell.

Answer: We appreciate the comment from the reviewer, please kindly refer to our answer to R3C2. We added a new figure (Fig. S2 in our response to R3C5) to show the locations of the ground observations we used.

R3C8: - Same suggestion for the nutrient network relationship and the biomass abundance.

Answer: Please kindly refer to our answers to R3C5. We have removed the part about biomass abundance.

R3C9: Clarify how Fig 1d is produced. How exactly did the authors couple (modelled) AC_4/AC_3 to (observed) area abundance? Was the model output coarse-grained to $10^\circ \times 10^\circ$ as well? If so, why don't the authors simply use the (0.5°) model grid instead? Maybe also justify why the relationships shown in Fig 1d would be time-invariant.

Answer: Please kindly refer to our answer to R3C3 regarding how we found the relationship is scale independent. And refer to our response to R3C4 on whether the time range could be an issue.

R3C10: - The authors mention the use of a satellite-based herbaceous fraction map (L393), but this is not referenced or clearly discussed further in the methods. Maybe add a paragraph with some more details?

Answer: We apologize for the missing of reference. We have now added a new section in Methods to describe the remote sensing-based grassland fraction maps we used. Please see below:

“Remote sensing estimates of global grassland fraction

Multiple remote sensing products provide information on grassland distribution. Some directly provide continuous fraction value (i.e., GLC(Tsendbazar et al. 2021) at 100 meter and Dynamic World(Brown et al. 2022) at 10 meter) and some provide categoric information on grassland and savannas (i.e., MODIS(Sulla-Menashe and Friedl, n.d.) at 500 meter and ESA-CCI(ESA 2017) and 300 meter). For former, we can directly calculate the fraction value at 0.5 degree, for the latter, we assign the grassland/savanna type pixel to 100% and others to 0% grassland, and then obtain the mean value for each 0.5 grid cell. Nevertheless, we are surprised to find those four estimates of grassland fraction vary considerably (Fig. S4). By visually compare the four estimates (i.e., GLC, Dynamic World, MODIS and ESA-CCI) against some ground-based estimates, we found Dynamic world and ESA-CCI substantially underestimate grassland fraction. We therefore move on to use only MODIS and GLC estimates of grassland fractions in our study.

MODIS grassland fraction is available from 2001 to 2019. GLC is only available from 2015-2019. To extend GLC to 2001, we employed a random forest approach to estimate GLC estimates based on surface reflectance, climate and soil and extrapolate it to 2001 (i.e., the training accuracy is 99% and the validation accuracy is 95%). We used the average of MODIS and GLC estimates to represent the grassland fraction. To quantify the uncertainty of the approach, for each pixel we bootstrapped 1000 times between the MODIS estimate and the GLC estimate, and use the one standard deviation of these 1000 values to represent the uncertainty in grassland fraction.”

R3C11: - Please explain into more detail what is the “emergent constraint approach” and how it was implemented, because this may not be immediately clear for all readers.

Answer: We added a new statement about the emergent constraint approach in Methods. Please see below:

“Emergent constraint approach

The emergent constraint technique is widely used in climate and modelling communities to infer unobserved quantities of interest in land surface processes(Cox 2019; Eyring et al. 2019). The underlying assumption is that although there is a large spread in the model estimates of an observed variable X and an unobserved variable Y across models, the relationship linking the two is tightly constrained across models. Based on the strong and robust relationship across models between X and Y, observations of X can be used to generate a constraint on Y. This approach has been termed ‘emergent’ because the functional relationship cannot be diagnosed from a single model, but rather emerges from the spread of the model estimates. The emergent constraint identified in this study links the contribution of C4 grasses/crops to total GPP to the percentages of area covered by C4 grasses/crops.”

Detailed comments

R3C12: - L81-84: I think these statements should be a bit more nuanced. For example, LPJ-GUESS does use bioclimatic temperature limits for tropical trees and C4 grass survival (>15.5°C; Smith et al. 2014), but the simulated dominant cover type (e.g. C3 vegetation vs C4 grass) emerges from competition for resources between cohorts of plant functional types. Likewise for the Ecosystem Demography model v2.2, which does not use such bioclimatic limits (Longo et al. 2019). These are of course just two examples.

Answer: We thank the reviewer for the comments. It was indeed challenging to identify how DGVMs consider the C_4 distribution process as those are not often explicitly described. We updated the statement and added “Some cohort-based models further consider competition for resources (Smith et al. 2014) and disturbances (Longo et al. 2019) in C_4 distribution simulations”.

R3C13: - L86 and the rest of this paragraph: Are these all distribution models? Maybe clarify which kind of model you’re referring to.

Answer: Yes, they are all distribution models. We have updated the statement to clarify that.

R3C14: - L115 and elsewhere in the text: Just a very minor comment about the language. Please don’t simply write “ C_4 ” but add vegetation/grass/photosynthesis/... after it as well.

Answer: Thank you for the suggestion. We have updated the text where applicable.

R3C15: - L119: Please provide a reference for the global herbaceous fraction map.

Answer: Please kindly refer to our response to R3C10.

R3C16: - L122: Please clarify what an “emergent constraint approach” is and how it’s implemented. Maybe in the methods section? Is this simply the slope of the linear regression in Fig 3a,b?

Answer: Please kindly see our response to R3C11.

R3C17: - L132: Just wondering, why do you use AC_4/AC_3 instead of $AC_4/(AC_3+AC_4)$?

Answer: If we use $AC_4/(AC_3+AC_4)$, the metric will not indicate the photosynthetic advantage of C_4 over C_3 , but indicate the advantage of C_4 over a grassland where C_3 and C_4 are each 50%.

R3C18: - L135-138: This is already specified in the methods. Not sure if this needs to be repeated here.

Answer: We have removed this part.

R3C19: - L140: Just a semantics comment, but please clarify what you mean by observational constraint. Don’t you mean conversion factor, as you’re not really constraining the model parameters or the model output against observations?

Answer: Thank you for the suggestion. We have replaced “observation constraint” with “the $AC_4/AC_3 - C_4$ coverage relationship” throughout the manuscript.

R3C20: - L142: Minor remark: these values are not directly obvious from Fig 1. I suppose these can be obtained by averaging Fig 1 over all land area?

Answer: We updated our text to clarify that the values here means % land surface area covered by C_4 . It is not the average of (the original) Fig. 1, as the area of each pixel is impacted by map projection. We calculated the total area of C_4 by summing the C_4 area in each pixel, and then divided the total C_4 area by the total land surface area to get the percentage.

R3C21: - L143: Just for clarity, here you applied the herbaceous fraction map (EAS-CCI) to obtain C_4 grass from the C_4 area abundance map, right?

Answer: Yes, we first acquired C_4 coverage (% of grassland covered by C_4), and then multiplied the value by the grassland fraction maps from remote sensing (% of land surface covered by grassland) to get the C_4 area abundance (% of land surface covered by C_4). We have added a flow chart (please see Fig. S1 in our answer to R3C3) to better present the process.

R3C22: - L203: It would be interesting to see the if there's any difference in attribution between different regions or aridity zones. Maybe add a map in supplement for each driver driver?

Answer: Sure, we added a new figure (Fig. S7) to show the spatial variation of each contributing factor.

Figure S7. The impacts of climate drivers on C₄ natural grasslands change from 2001 to 2019. (a) CO₂, (b) air temperature, (c) VPD and (d) soil moisture.

R3C23: - L237: Please clarify in the caption which model these figures are based on. I suppose the optimality model? Just asking because in the paragraph where this figure is referenced (L221-234) you discuss DGVM output. Perhaps it would be good to clarify that in this paragraph as well then.

Answer: Thank you, we have added that the emergent constraint was developed based on an ensemble of DGVMs.

R3C24: - L284: Do you mean “considerable” instead of “considered”?

Answer: We changed “considered” to “included”. We only meant that CO₂ is a process included in DGVMs, however, we are unsure if it has “considerable” impacts in all DGVMs.

R3C25: - L307: From here on, this paragraph mainly repeats what is already mentioned elsewhere. Consider rephrasing or removal.

Answer: We removed the part as suggested.

R3C26: - L314: Paragraph is also a bit repetitive from the results section..

Answer: We removed the paragraph as suggested.

R3C27: - L350: Just for clarity, MODIS LAI is only used to derive understorey PAR for model input, right?

Answer: Yes, we only used LAI for estimating understorey PAR.

R3C28: - L359-362: Are these new results? Maybe move this to the methods and results sections?

Answer: This dataset (Jackson et al. 2019) has been first introduced in Results (Figure 3f). Here we are expanding on the result to further explain the difference between LUHv2 (i.e., it used a fixed C₄ crop fraction) and Jackson et al 2019 (i.e., it used a dynamic C₄ crop fraction), and the implication for our conclusion.

R3C29: - L382: Just a minor suggestion: it can maybe be nice to add a little flowchart to accompany this overview. That will make it easier to keep track and understand how the different datasets and model are combined in this study..

Answer: We fully agree with the reviewer and have added a flowchart (Fig. S1). Please see it in our answer to R3C3.

R3C30: - L387: Can you specify which leaf traits were used?

Answer: We updated the statement to specify “leaf nitrogen content”

R3C31: - L388: Please clarify how this relationship is established.

Answer: We updated the statement to “(we establish the relationship)...based on an assumption that larger AC_4/AC_3 indicates higher C_4 grass coverage (% of grassland covered by C_4 grasses). Using the $AC_4/AC_3 - C_4$ coverage relationship, we estimated the C_4 grass coverage (a.k.a. potential C_4 abundance when grass covers 100% of the land surface) from estimated AC_4/AC_3 for the globe”. We also pointed readers to the flow chart Fig. S1 on how to establish the relationship.

R3C32: - L413: Please provide units for area abundance.

Answer: We added the unit of the area abundance (% of land surface area covered by C_4), following the suggestion from the reviewer.

R3C33: - L479: I suppose this model doesn't need any initialization? (e.g. spinup phase)

Answer: Yes, there was no spin up. We added that in the method “The simulation was conducted at annual time step and there is no need for model initialization.”

R3C34: - L523: Maybe elaborate a little bit on the S3 scenario

Answer: We updated the statement to read “S3 scenario (i.e. considering elevated CO₂, climate change and land use change)...in our analysis”

R3C35: - L571: Figure S1a needs to be improved. The figure has a very low resolution and it is not explained in the caption what the red circles in this figure mean. The light gray points are nearly invisible against the white background. Fig S1b would benefit from having density plots along the x- and y-axes, as well as a better explanation of the data used. For instance, are these the original 61588 records of photosynthetic pathways, or only the 13919 non-woody ones?

Answer: We removed the original Figure S1a and replace it with a new figure (Fig. S2) to present the location of ground observations used.

R3C36: - L598: Please make these maps larger

Answer: We made the maps larger as the reviewer suggested.

Figure S8. The changes in the area of three major C4 crops from 2001 to 2014. (a) Maize, (b) Millet and (c) Sorghum.

References

Longo, M., Knox, R.G., Medvigy, D.M., Levine, N.M., Dietze, M.C., Kim, Y., Swann, A.L.S., Zhang, K., Rollinson, C.R., Bras, R.L., Wofsy, S.C., Moorcroft, P.R., 2019. The biophysics, ecology, and biogeochemistry of functionally diverse, vertically and horizontally heterogeneous ecosystems: The Ecosystem Demography model, version 2.2-Part 1: Model description. *Geoscientific Model Development* 12, 4309–4346. <https://doi.org/10.5194/gmd-12-4309-2019>

Smith, B., Wårlind, D., Arneth, A., Hickler, T., Leadley, P., Siltberg, J., Zaehle, S., 2014. Implications of incorporating N cycling and N limitations on primary production in an individual-based dynamic vegetation model. *Biogeosciences* 11, 2027–2054. <https://doi.org/10.5194/bg-11-2027-2014>

Brown, Christopher F., Steven P. Brumby, Brookie Guzder-Williams, Tanya Birch, Samantha Brooks Hyde, Joseph Mazzariello, Wanda Czerwinski, et al. 2022. “Dynamic World, Near Real-Time Global 10 m Land Use Land Cover Mapping.” *Scientific Data* 9 (1): 251. <https://doi.org/10.1038/s41597-022-01307-4>.

Caemmerer, Susanne von. 2000. *Biochemical Models of Leaf Photosynthesis*. Techniques in Plant Sciences 2. Collingwood: CSIRO Publishing.

Collatz, GJ, M Ribas-Carbo, and Ja Berry. 1992. “Coupled Photosynthesis-Stomatal Conductance Model for Leaves of C4 Plants.” *Functional Plant Biology* 19 (5): 519. <https://doi.org/10.1071/PP9920519>.

Cox, Peter M. 2019. “Emergent Constraints on Climate-Carbon Cycle Feedbacks.” *Current Climate Change Reports* 5 (4): 275–81. <https://doi.org/10.1007/s40641-019-00141-y>.

Edwards, Erika J., and Stephen A. Smith. 2010. “Phylogenetic Analyses Reveal the Shady History of C₄ Grasses.” *Proceedings of the National Academy of Sciences* 107 (6): 2532–37. <https://doi.org/10.1073/pnas.0909672107>.

- Edwards, Erika J., and Christopher J. Still. 2008. "Climate, Phylogeny and the Ecological Distribution of C₄ Grasses." *Ecology Letters* 11 (3): 266–76. <https://doi.org/10.1111/j.1461-0248.2007.01144.x>.
- ESA. 2017. "Land Cover CCI Product User Guide Version 2. Tech. Rep." <https://www.esa-landcover-cci.org/>.
- Eyring, Veronika, Peter M. Cox, Gregory M. Flato, Peter J. Gleckler, Gab Abramowitz, Peter Caldwell, William D. Collins, et al. 2019. "Taking Climate Model Evaluation to the next Level." *Nature Climate Change* 9 (2): 102–10. <https://doi.org/10.1038/s41558-018-0355-y>.
- Friedlingstein, Pierre, Michael O Sullivan, Matthew W Jones, Robbie M Andrew, and Judith Hauck. 2020. "Global Carbon Budget 2020" 2020: 3269–3340.
- Griffith, Daniel M., T. Michael Anderson, Colin P. Osborne, Caroline A. E. Strömberg, Elisabeth J. Forrester, and Christopher J. Still. 2015. "Biogeographically Distinct Controls on C₃ and C₄ Grass Distributions: Merging Community and Physiological Ecology: Climate Disequilibrium in C₄ Grass Distributions." *Global Ecology and Biogeography* 24 (3): 304–13. <https://doi.org/10.1111/geb.12265>.
- Griffith, Daniel M., Jennifer M. Cotton, Rebecca L. Powell, Nathan D. Sheldon, and Christopher J. Still. 2017. "Multi-century Stasis in C₃ and C₄ Grass Distributions across the Contiguous United States since the Industrial Revolution." *Journal of Biogeography* 44 (11): 2564–74. <https://doi.org/10.1111/jbi.13061>.
- Jackson, Nicole D., Megan Konar, Peter Debaere, and Lyndon Estes. 2019. "Probabilistic Global Maps of Crop-Specific Areas from 1961 to 2014." *Environmental Research Letters* 14 (9). <https://doi.org/10.1088/1748-9326/ab3b93>.
- Monfreda, Chad, Navin Ramankutty, and Jonathan A. Foley. 2008. "Farming the Planet: 2. Geographic Distribution of Crop Areas, Yields, Physiological Types, and Net Primary Production in the Year 2000: GLOBAL CROP AREAS AND YIELDS IN 2000." *Global Biogeochemical Cycles* 22 (1): n/a-n/a. <https://doi.org/10.1029/2007GB002947>.
- Pau, Stephanie, Erika J. Edwards, and Christopher J. Still. 2013. "Improving Our Understanding of Environmental Controls on the Distribution of C₃ and C₄ Grasses." *Global Change Biology* 19 (1): 184–96. <https://doi.org/10.1111/gcb.12037>.
- Pignon, Charles P., and Stephen P. Long. 2020. "Retrospective Analysis of Biochemical Limitations to Photosynthesis in 49 Species: C₄ Crops Appear Still Adapted to Pre-industrial Atmospheric [CO₂]." *Plant, Cell & Environment* 43 (11): 2606–22. <https://doi.org/10.1111/pce.13863>.
- Powell, Rebecca L., Eun-Hye Yoo, and Christopher J. Still. 2012. "Vegetation and Soil Carbon-13 Isoscapes for South America: Integrating Remote Sensing and Ecosystem Isotope Measurements." *Ecosphere* 3 (11): art109. <https://doi.org/10.1890/ES12-00162.1>.
- Powell, Rebecca, and Christopher Still. 2009. "Biogeography of C₃ and C₄ Vegetation in South America." *An. XIV Simpósio Brasileiro Sensoriamento Remoto* 14 (January).
- Sage, Rowan F., and Stefanie Sultmanis. 2016. "Why Are There No C₄ Forests?" *Journal of Plant Physiology* 203 (C): 55–68. <https://doi.org/10.1016/j.jplph.2016.06.009>.
- Still, Christopher J., Joseph A. Berry, G. James Collatz, and Ruth S. DeFries. 2003. "Global Distribution of C₃ and C₄ Vegetation: Carbon Cycle Implications." *Global Biogeochemical Cycles* 17 (1): 6-1-6–14. <https://doi.org/10.1029/2001gb001807>.
- Sulla-Menashe, Damien, and Mark A Friedl. n.d. "User Guide to Collection 6 MODIS Land Cover (MCD12Q1 and MCD12C1) Product."
- Tsendbazar, N., M. Herold, L. Li, A. Tarko, S. De Bruin, D. Masiliunas, M. Lesiv, et al. 2021. "Towards Operational Validation of Annual Global Land Cover Maps." *Remote Sensing of Environment* 266 (December): 112686. <https://doi.org/10.1016/j.rse.2021.112686>.
- Wei, Y., S. Liu, D. N. Huntzinger, A. M. Michalak, N. Viovy, W. M. Post, C. R. Schwalm, et al. 2014. "The North American Carbon Program Multi-Scale Synthesis and Terrestrial Model Intercomparison Project - Part 2: Environmental Driver Data." *Geoscientific Model Development* 7 (6): 2875–93. <https://doi.org/10.5194/gmd-7-2875-2014>.
- Zhou, Haoran, Erol Akçay, and Brent Helliker. 2023. "Optimal Coordination and Reorganization of Photosynthetic Properties in C₄ Grasses." *Plant, Cell & Environment* 46 (3): 796–811. <https://doi.org/10.1111/pce.14506>.
- Zhou, Haoran, Brent R. Helliker, Matthew Huber, Ashley Dicks, and Erol Akçay. 2018. "C₄ Photosynthesis and Climate through the Lens of Optimality." *Proceedings of the National Academy of Sciences of the United States of America* 115 (47): 12057–62. <https://doi.org/10.1073/pnas.1718988115>.

Reviewer #2 (Remarks to the Author):

The authors have done a very diligent and commendable job at addressing the reviewers' concerns. As a result, the manuscript now appears stronger and more convincing compared to the previous version. I have just one residual comment:

The newly added dataset has provided a much more granular picture of the C4 coverage \sim AC4 /AC3 relationship, which now also looks much more convincing. Note that the new data appears (at least visually) to follow a more logistic-like curve with a threshold AC4/AC3 \sim 1.5, below which you have C3 dominance and above which you get C4 dominance. This makes sense (conceptually) for 2 reasons: (1) a linear relationship is theoretically impossible, since you cannot possibly have C4 fractions above 100%. (2) As I mentioned in my previous review, even slight differences in competitive advantage should in principle bring about complete dominance of one species (or functional group) over others in the long run, unless there is a known mechanism (such as immigration, micro-habitats) that can sustain the less competitive species. Note that a ratio of AC3/AC4=1 does not necessarily represent the threshold value of equal competitive advantage, because these are just leaf-level rates and the scaling to the whole-plant level may incur additional costs/benefits. In conclusion, I would recommend using a logistic curve to fit this relationship for conceptual reasons, even if it doesn't turn out to be the best one statistically.

Jaideep Joshi

Reviewer #4 (Remarks to the Author):

I reviewed the manuscript 'Global C4 distribution estimate constrained by observations and optimality theory submitted for publication in Nature. I was specifically asked to review the macroecological methods (or rather how the comments by Reviewer 3 in regard to macroecological methods were dealt with) and therefore the focus of my review is mainly on those issues (but I've added some comments regarding some other issues as well).

1. I don't see a problem in aggregating TRY data for 10x10 degree cells for a global scale study IF the results reflect the coarseness as well. Unfortunately, I don't think this is the case. I also don't find the explanation given to Reviewer 3 in response to this matter sufficient (comparing TRY and Northern American dataset on 10x10 km scale) – Northern America is also well sampled in the TRY database, so a good correlation between the regional and global datasets is expected, but this doesn't answer the questions whether TRY represents less sampled regions and whether the AC4/AC3 and C4 coverage relationship applies to those regions as well. I don't think there's much to be done here other than state the limitations of the study clearly.

2. Also the details about the TRY data are not fully described. There's no information about:

- a. how the TRY data was assigned into grid cells – according to Fig. S2 there seem to be some gaps between the cells, why?
- b. how many grid cells were used – is it the 12 that shown on Fig. S2 or more (as it seems from Fig. 1b).
- c. where are these grid cells located globally, were all the datapoints included in the cells?
- d. is there any sampling bias for the data (e.g. only one photosynthesis type is sampled in a location – this bias I think the large cell size could possibly minimise), did the TRY data include both natural plants and crops?

3. The linear model to link AC4/AC3 and C4 grass coverage (Fig. 1b) should include a logit link since the response variable is a percentage (I don't think it currently includes it). I saw from a response to R2C5 that you also tested other models, but a simple linear regression shouldn't be an option

(currently values greater than 100% are predicted, which are not possible).

4. In the title and throughout the text I find the term 'C4 distribution' not meaningful, we can talk about C4 plants/vegetation/photosynthesis/etc., but C4 by itself is just a name or label.

Title: I understand that 'distribution estimate constrained by' is referring to the model, but it could be also understood that the ability to estimate C4 plant distribution is limited. Maybe use a more general title.

Line 50 – add a reference for this statement

Line 61 – use 'compared to'

Line 76 – use 'woody plant encroachment'

Line 155 – but when AC4 is 2.5 times greater, then the linear model predicts C4 coverage of >100%...

Lines 175-176 – do you mean that when 'grassland covers'?

Lines 206-207, 280-283 – what do the numbers after +- sign show? Is there a decrease in C4 area if the error is larger than the decrease?

Line 453 – I assume that this was georeferenced data? or how did you aggregate them into latitude x longitude cells?

Line 457 – this is not a very good explanation to exclude woody species, these could still be included in the total number of species, better to explain that you focused on herbaceous species and grasslands in general

Line 459 – I think 'species abundance' is not very accurate here, maybe better 'C4 species richness' (as is also in line 468)?

Line 460 – this needs further clarification, the 'total number of herbaceous species' is only species with photosynthesis information in TRY not all herbaceous species?

Line 464 – add information about how many cells were considered, according to Fig. S2 it seems 12, is that all?

Line 467 – unclear what is 'area abundance'

Line 469 – grass species or herbaceous species?

Line 470 – the sentence implies that reference 45 has explored C4 species richness, but there's no mention of C4 in that study

Line 471 – mention how many plots/sites

Line 488 – mention somewhere if the TRY and DG datasets included both natural and crop species

Line 540 – for the global map on Fig. 1c, how was this again on the 0.5x0.5 scale if the relationship behind it in on 100x100 km or 10x10 degree scale?

Figure 1:

Adjust the figures or axis titles, currently difficult to understand which text is part of a legend of a and which is the title for y-axis in b (same for c and d).

Figure 2:

Perhaps for uncertainty a different color scheme could be used, since it shows something different than the distribution maps.

Figure S2.

Are the aggregated TRY cells marked on the plot the only ones used? (in the Fig. 1 there seems to be more points than the 12 cells marked here). Better to show all cells on the map here to demonstrate global data coverage/availability.

Unclear how the grid cells were assigned (why there's a less than 10 degree gap between cells).

Was there enough data in the cell located mostly in the Atlantic ocean near Florida to accurately estimate relative abundance of C4 plants?

Mapping the global distribution of C₄ vegetation using observations and optimality theory
NCOMMS-23-03867A-Z
Response to Reviewers

=====
We thank the editor and the reviewers for their constructive comments on our manuscript. We have endeavored to address their remaining comments to further improve our study. Please kindly see our point-by-point response below.

=====
Reviewer #2 (Remarks to the Author):

R2C1: The authors have done a very diligent and commendable job at addressing the reviewers' concerns. As a result, the manuscript now appears stronger and more convincing compared to the previous version. I have just one residual comment:

The newly added dataset has provided a much more granular picture of the C₄ coverage ~ AC₄ / AC₃ relationship, which now also looks much more convincing. Note that the new data appears (at least visually) to follow a more logistic-like curve with a threshold AC₄/AC₃ ~ 1.5, below which you have C₃ dominance and above which you get C₄ dominance. This makes sense (conceptually) for 2 reasons: (1) a linear relationship is theoretically impossible, since you cannot possibly have C₄ fractions above 100%. (2) As I mentioned in my previous review, even slight differences in competitive advantage should in principle bring about complete dominance of one species (or functional group) over others in the long run, unless there is a known mechanism (such as immigration, micro-habitats) that can sustain the less competitive species. Note that a ratio of AC₃/AC₄=1 does not necessarily represent the threshold value of equal competitive advantage, because these are just leaf-level rates and the scaling to the whole-plant level may incur additional costs/benefits. In conclusion, I would recommend using a logistic curve to fit this relationship for conceptual reasons, even if it doesn't turn out to be the best one statistically.

Jaideep Joshi

Answer: We appreciate the time and effort from the reviewer to help us improve the study, and glad to know the study is more convincing to the reviewer now.

We fully agree with the reviewer that C₄ coverage percentage should not be greater than 100% - in the last version, we simply put a cap of 100% to the estimates from the linear model. In this round of the revision, we followed the reviewer's suggestion to use a logistic curve to build the AC₄/AC₃ - C₄ percentage relationship (please refer to the updated Fig. 1b below). After the update, we found the global C₄ area abundance (i.e., % of land surface covered by C₄) is 17.5%, decreasing from 17.7% in 2001-2005 to 17.1% in 2015-2019. The values changed slightly compared to using a linear model (global C₄ area abundance was 16.5% in the last version), but our conclusion remains the same. We have updated all relevant values throughout the manuscript.

Figure 1. C₄ natural grass coverage estimated by the optimality model. (a) the ratio of C₄ to C₃ photosynthesis estimated by the optimality model (AC₄/AC₃) over global non-woody regions; (b) the relationship between observed C₄ coverage (% of grassland) and estimated C₄/C₃ photosynthetic ratio by the optimality model. C₄ coverage observation obtained from difference sources (i.e., TRY, DG; please see methods); gray shaded area indicates the uncertainty range for the relationship between AC₄/AC₃ and C₄ coverage (i.e., 95% confidence interval); (c) C₄ grass coverage (% of grassland) over the globe, which can be regarded as the potential C₄ area abundance when grassland covers 100% of the land surface; (f) C₄ coverage in a climate space of mean annual temperature (MAT: °C) and mean annual precipitation (MAP: mm/yr).

=====

Reviewer #4 (Remarks to the Author):

I reviewed the manuscript 'Global C₄ distribution estimate constrained by observations and optimality theory submitted for publication in Nature. I was specifically asked to review the macroecological methods (or rather how the comments by Reviewer 3 in regard to macroecological methods were dealt with) and therefore the focus of my review is mainly on those issues (but I've added some comments regarding some other issues as well).

R4C1: 1. I don't see a problem in aggregating TRY data for 10x10 degree cells for a global scale study IF the results reflect the coarseness as well. Unfortunately, I don't think this is the case. I also don't find the explanation given to Reviewer 3 in response to this matter sufficient (comparing TRY and Northern American dataset on 10x10 km scale) – Northern America is also well sampled in the TRY database, so a good correlation between the regional and global datasets is expected, but this doesn't answer the questions whether TRY represents less sampled regions and whether the AC₄/AC₃ and C₄ coverage relationship applies to those regions as well. I don't think there's much to be done here other than state the limitations of the study clearly.

Answer: We thank the reviewer for their valuable comments on our study. We would like to first apologize for the misrepresentation of aggregated TRY data points in Fig. S2. By aggregating TRY records to 10-degree cells, we obtained 23 (not 12) valid data points to build the AC₄/AC₃-C₄ coverage relationship. Please kindly refer to

our answer to R4C2, where we identified the mistake in plotting Fig. S2. We have rectified the issue and provided an updated Fig. S2 (please see below).

Figure S2. The spatial distribution of C_4 observations used in the study. “TRY raw” means the locations where there are C_4 observations, “TRY” is the aggregated TRY raw observations at 10-degree spatial resolution (i.e., each aggregated grid cell for TRY has more than 50 species, including C_3 and C_4), “DG” means the C_4 observations compiled for North America at 100 km spatial resolution, “NutNet” indicates some sites from Nutrient Network where we obtained the observations of grass species richness and species coverage.

Based on the updated Fig. S2, we observed that out of the 23 TRY data points, 8 are located in North America, 10 in Eurasia, 1 in South America, and 4 in Oceania. Moreover, we identified discrepancies between the DG and TRY datasets in North America. In the eastern U.S., the DG dataset has fewer records, while in the western part of the U.S., the DG dataset contains more records than the TRY dataset. Therefore, we suggest that the AC_4/AC_3 ratio – C_4 coverage relationships from TRY and DG are not expected to be similar, as the TRY dataset offers broader, albeit sparser, spatial coverage compared to the DG dataset.

That being said, we do share the sentiment with the reviewer that regions outside North America are not well represented. In this round of revision, we found a recently published C_4 vegetation cover map of Australia generated by intensive isotopic measurements and remote sensing (Munroe et al. 2022) (Fig. R1). We compared our estimate of C_4 coverage against this independent C_4 map. The maps showed great agreement ($R = 0.69$, $p < 0.01$; Fig. S10).

Fig. R1. The Figure 6 in Munroe et al. 2022. Proportional (%) herbaceous C_4 cover (relative to herbaceous C_3 and C_4 herbaceous cover) extrapolated across Australia. Note that the gridded data from the figure is not

available, but paper provides regional average values of mean C₄ cover and herbaceous coverage for us to infer the regional average C₄ grass coverage.

Figure S10. The correlation between Australian C₄ grass coverage estimated by the optimality model used in our study (on the x-axis) and the C₄ grass coverage estimated by isotope measurements and remote sensing in Munroe et al. 2022 (on the y-axis). Each data point represents one of the continental Australian bioregions defined by the Interim Biogeographic Regionalisation for Australia version 7 (IBRA 7.0). It's important to note that the actual number of data points is lower than the total number of Australian bioregions due to some bioregions lacking C₄ estimates from our 0.5-degree C₄ map and others reporting C₄ grass coverage (isotope) exceeding 100%. The formula used for calculating C₄ grass coverage (isotope) is 'Mean_Proportional_C₄_Cover' divided by 'Mean_%_herbaceous_cover', with the values obtained from the supplementary table of Munroe et al. 2022.

Based on the analysis above, we have added the following statement in Discussion (Line 504-510):

“We acknowledge that the distribution of our observations was not uniform across the globe, with North America being better represented compared to other regions (Fig. S2). As an additional test to validate our estimation of C₄ vegetation distribution, we compared the C₄ grass coverage estimated in our study (Fig. 1c) with the C₄ coverage estimated from isotopic measurements and remote sensing in Australia (Fig. S10). This validation demonstrates a strong agreement ($r = 0.69$, $p < 0.01$) between the two independent estimates, affirming the robustness of our estimates in under-sampled regions”.

R4C2: 2. Also the details about the TRY data are not fully described. There's no information about:

- how the TRY data was assigned into grid cells – according to Fig. S2 there seem to be some gaps between the cells, why?
- how many grid cells were used – is it the 12 that shown on Fig. S2 or more (as it seems from Fig. 1b).
- where are these grid cells located globally, were all the datapoints included in the cells?
- is there any sampling bias for the data (e.g. only one photosynthesis type is sampled in a location – this bias I think the large cell size could possibly minimise), did the TRY data include both natural plants and crops?

Answer:

a, b) We apologize again for the confusion caused by the original Fig. S2. There should be 23 grid cells for aggregated TRY data in the figure. The original figure only showed 12 data points, as I was testing a case using 15-degree cells and a different minimal requirement of records for aggregation (therefore less cells!). Meanwhile, the size of the cells in the figure was not indicative of their geographic scopes, as we just used the red boxes as a symbol to present the locations of these cells.

In this round of revision, we rectified the issues above. We double checked that the data presented in Fig. S2 is the ones we obtained for the 10-degree cells. We have also made changes to the size of the cells, plotting

them using the geographic coordinates of their 10-degree boundaries. After doing so, we removed the gaps between the grid cells. Please kindly refer to our update Fig. S2 in our answer to R4C1.

c). Based on Fig. S2, we learned that not all C_4 data points in the raw TRY dataset were included, as some raw C_4 records are at places that are isolated or do not have enough number of total species records to be statistically robust ($n < 50$). In this step, we used 1619 out of the 1881 raw C_4 grass species records (86%). We added the following statement in Line 471-475.

“Here we use the large-size grid cell to make sure there were enough samples in cells to acquire a meaningful estimate of C_4 abundance – in this analysis, each cell should have at least 50 species (i.e., C_3 and C_4 in total). We used 1619 (out of 1881) C_4 species records in this aggregation step, and obtained 23 10x10 degree cells for the analysis (Fig. S2).”

We took the chance to correct a legacy issue in our description of the method. When determining whether a 10-degree cell is selected or not for our analysis, our criteria is that within that cell we “should have at least 50 species reported (i.e., C_3 and C_4 in total)”, not that “the cell should have more than 200 C_4 records”.

d). We did a brief examination and found on 67% of sites that have C_4 species records, they are accompanied with observations of other photosynthetic pathways. We agree with the reviewer that the sampling bias was further minimized by using the relatively large cells for aggregation.

R4C3: 3. The linear model to link AC_4/AC_3 and C_4 grass coverage (Fig. 1b) should include a logit link since the response variable is a percentage (I don't think it currently includes it). I saw from a response to R2C5 that you also tested other models, but a simple linear regression shouldn't be an option (currently values greater than 100% are predicted, which are not possible).

Answer: Following the comment and a similar comment from reviewer 2, we have updated our results to use a logistic equation to describe the $AC_4/AC_3 - C_4$ grass coverage relationship, which caps the C_4 grass coverage at 100% (Fig. 1b below). The values we reported changed slightly but our conclusion was unaffected. We would also like to clarify that we were unable to find studies on the hypothetical representation of the relationship between $AC_4/AC_3 - C_4$ grass coverage, or in general how photosynthetic advantage of species impacts species coverage (hope we did not overlook any). Hence, we initially relied on a data-driven linear equation, which, however, exhibited the issue of overshooting beyond 100%.

Figure 1. C₄ natural grass coverage estimated by the optimality model. (a) the ratio of C₄ to C₃ photosynthesis estimated by the optimality model (AC_4/AC_3) over global non-woody regions; (b) the relationship between observed C₄ coverage (% of grassland) and estimated C₄/C₃ photosynthetic ratio by the optimality model. C₄ coverage observation obtained from difference sources (i.e., TRY, DG; please see methods); gray shaded area indicates the uncertainty range for the relationship between AC_4/AC_3 and C₄ coverage (i.e., 95% confidence interval); (c) C₄ grass coverage (% of grassland) over the globe, which can be regarded as the potential C₄ area abundance when grassland covers 100% of the land surface; (f) C₄ coverage in a climate space of mean annual temperature (MAT: °C) and mean annual precipitation (MAP: mm/yr).

R4C4: 4. In the title and throughout the text I find the term ‘C4 distribution’ not meaningful, we can talk about C4 plants/vegetation/photosynthesis/etc., but C4 by itself is just a name or label.

Title: I understand that ‘distribution estimate constrained by’ is referring to the model, but it could be also understood that the ability to estimate C4 plant distribution is limited. Maybe use a more general title.

Answer: We have updated the title to “Mapping the global distribution of C₄ vegetation using observations and optimality theory”, following the suggestion from the reviewer.

R4C5: Line 50 – add a reference for this statement

Answer: Thank you. We added the following reference to support the statement.

Sage, R. F. The evolution of C₄ photosynthesis. *New Phytologist* 161, 341–370 (2004).

R4C6: Line 61 – use ‘compared to’

Answer: We apologize for the missing and have added ‘to’ here.

R4C7: Line 76 – use ‘woody plant encroachment’

Answer: We have changed ‘woody encroachment’ to ‘woody plant encroachment’ as suggested.

R4C8: Line 155 – but when AC₄ is 2.5 times greater, then the linear model predicts C₄ coverage of >100%...

Answer: In this revised version, we used a logistic equation for the fitting to ensure that predicted C₄ coverage is not greater than 100%. Please kindly refer to our answers to R4C3.

R4C9: Lines 175-176 – do you mean that when ‘grassland covers’?

Answer: we have changed ‘grass covers’ to ‘grassland covers’.

R4C10: Lines 206-207, 280-283 – what do the numbers after +/- sign show? Is there a decrease in C₄ area if the error is larger than the decrease?

Answer: “±” indicates that the numbers followed are the uncertainty range (i.e., usually one standard deviation) of our estimates. We have included clarifications in those lines to explain that the numbers following “±” represent one standard deviation. The standard deviation values reported for C₄ area changes are considerably smaller than the mean values. Therefore, it is improbable that they would impact the direction of C₄ area change.

R4C11: Line 453 – I assume that this was georeferenced data? or how did you aggregate them into latitude x longitude cells?

Answer: We have added ‘georeferenced’ in the description of the TRY records. We used the latitude and longitude information of each record to aggregate them into large cells.

R4C12: Line 457 – this is not a very good explanation to exclude woody species, these could still be included in the total number of species, better to explain that you focused on herbaceous species and grasslands in general

Answer: Per suggestion, we added a new statement “...as our study aimed to examine C₄ grass distribution...” in Line 465.

R4C13: Line 459 – I think ‘species abundance’ is not very accurate here, maybe better ‘C₄ species richness’ (as is also in line 468)?

Answer: Thank you, we have updated the term to “species richness”, consistent with other places.

R4C14: Line 460 – this needs further clarification, the ‘total number of herbaceous species’ is only species with photosynthesis information in TRY not all herbaceous species?

Answer: Yes, in this paragraph the values reported were derived from the TRY database. We added that “...the numbers were derived from the available records in the TRY database” in Line 468.

R4C15: Line 464 – add information about how many cells were considered, according to Fig. S2 it seems 12, is that all?

Answer: We added that “We obtained 23 10x10 degree cells for the analysis (Fig. S2)”

R4C16: Line 467 – unclear what is ‘area abundance’

Answer: We added “(% of the land surface covered by C₄)” to explain what is C₄ area abundance. We had explained the term in Line 181.

R4C17: Line 469 – grass species or herbaceous species?

Answer: We added “grass” before “species richness” to improve clarity.

R4C18: Line 470 – the sentence implies that reference 45 has explored C₄ species richness, but there’s no mention of C₄ in that study

Answer: In reference 45 (now the reference 46), there is a link to the grass species dataset they used, which contains information on C₄ species richness.

<https://portal.edirepository.org/nis/mapbrowse?packageid=edi.1037.2>. We added the link in our Data Availability statement.

R4C19: Line 471 – mention how many plots/sites

Answer: We added that “...across 34 sites. Each site has between 1 and 6 control plots” in Line 484.

R4C20: Line 488 – mention somewhere if the TRY and DG datasets included both natural and crop species

Answer: We added the following statements to clarify that crop species were not included in analysis for grass.

L466-468: For TRY database, “We further removed 82 records that belong to major C₄ crops (i.e., maize, sugarcane, millet, and sorghum) and kept 1881 records”.

L500: “Please note that the DG dataset only surveyed C₄ grass species”.

R4C21: Line 540 – for the global map on Fig. 1c, how was this again on the 0.5x0.5 scale if the relationship behind it in on 100x100 km or 10x10 degree scale?

Answer: We hope to clarify that the simulations from the optimality model was conducted at 0.5 degree, which is the common grid size for climate input and most vegetation model usage. We only downscaled the simulation (using spatial average) to 10 degree or 1 degree cells in order to support validation. We have added “As the relationships derived from both 10-degree (i.e., TRY) and 1-degree (i.e., DG) data were similar, we assumed that the relationship is scale-independent. Consequently, we applied it to 0.5-degree estimates of AC₄/AC₃ to infer global C₄ vegetation coverage” in Line 555-557.

R4C22: Figure 1:

Adjust the figures or axis titles, currently difficult to understand which text is part of a legend of a and which is the title for y-axis in b (same for c and d).

Answer: We have increased the horizontal gaps between the panels in Fig. 1, following the suggestion of the reviewer. Please see updated Fig. 1 in our answer to R4C3.

R4C23: Figure 2:

Perhaps for uncertainty a different color scheme could be used, since it shows something different than the distribution maps.

Answer: We have updated the color scheme for the uncertainty maps as suggested, please see below.

Figure 2. The modeled global distribution of C₄ vegetation and associated uncertainties. The area occupied by (a) C₄ natural grasses, (c) C₄ croplands and (e) all C₄ vegetation (unit: % of the land surface). The uncertainties of the area abundance of (b) C₄ natural grasses, (d) C₄ croplands and (f) all C₄ vegetation (unit: % of the land surface).

R4C24: Figure S2.

Are the aggregated TRY cells marked on the plot the only ones used? (in the Fig. 1 there seems to be more points than the 12 cells marked here). Better to show all cells on the map here to demonstrate global data coverage/availability.

Unclear how the grid cells were assigned (why there's a less than 10 degree gap between cells).

Was there enough data in the cell located mostly in the Atlantic ocean near Florida to accurately estimate relative abundance of C₄ plants?

Answer: Please kindly refer to our answer to R4C1 and R4C2, where we provided an updated and correct version of Fig. S2. In the correct version we do not have that cell near Atlantic Ocean.

Munroe, Samantha E. M., Greg R. Guerin, Francesca A. McInerney, Irene Martín-Forés, Nina Welti, Mark Farrell, Rachel Atkins, and Ben Sparrow. 2022. "A Vegetation Carbon Isoscape for Australia Built by Combining Continental-Scale Field Surveys with Remote Sensing." *Landscape Ecology* 37 (8): 1987–2006. <https://doi.org/10.1007/s10980-022-01476-y>.

Reviewer #4 (Remarks to the Author):

I reviewed the revised version of the manuscript 'Mapping the global distribution of C4 vegetation using observations and optimality theory'. The authors have done a great job in revising the paper and my main concerns from the previous review have been addressed. I still think that it would be better to not use the phrase 'C4/C3 distribution/cover', but rather to clarify 'C4/C3 plants/photosynthesis/grasses distribution'. Just using 'C4 distribution' is similar as e.g. 'vascular distribution' when considering vascular plant distribution, it is missing part of the term.

Some minor suggestions:

Abstract:

Line 41 – need to clarify what the +/- shows, SD, SE, range?

Introduction:

Line 134 – remove 'from'

Results:

Line 149 – use 'we found a strong positive relationship', the fact that it's logistic is statistically determined by the use of percentage data as a response variable and it's not something that is 'found' (logistic relationship is not a characteristic of the relationship between AC4/AC3 and C4 grass coverage specifically, rather it's a characteristic of any relationship where the response variable is fixed between 0 and 1 or 0% and 100%)

Line 194 – use 'as high as'

Line 238 – here and elsewhere maybe better to use 'air' in subscript in 'Tair'

Mapping the global distribution of C₄ vegetation using observations and optimality theory
NCOMMS-23-03867B
Response to Reviewers

=====
Reviewer #4 (Remarks to the Author):

R4C1: I reviewed the revised version of the manuscript 'Mapping the global distribution of C₄ vegetation using observations and optimality theory'. The authors have done a great job in revising the paper and my main concerns from the previous review have been addressed. I still think that it would be better to not use the phrase 'C₄/C₃ distribution/cover', but rather to clarify 'C₄/C₃ plants/photosynthesis/grasses distribution'. Just using 'C₄ distribution' is similar as e.g. 'vascular distribution' when considering vascular plant distribution, it is missing part of the term.

Answer: We thank the reviewer for helping us improve the study. We are very glad to know that their main concerns have been addressed. Following the remaining comment, we have checked the text to ensure that we used the terms "C₄ grass distribution" or "C₄ crop distribution" throughout. When we were referring to the sum of C₄ grass and crop distribution, we used the term "C₄ vegetation distribution" following Still et al. 2003.

Some minor suggestions:

Abstract:

R4C2: Line 41 – need to clarify what the +/- shows, SD, SE, range?

Answer: We added "mean ± one standard deviation" in the statement for clarification.

Introduction:

R4C3: Line 134 – remove 'from'

Answer: Thank you and removed 'from'.

Results:

R4C4: Line 149 – use 'we found a strong positive relationship', the fact that it's logistic is statistically determined by the use of percentage data as a response variable and it's not something that is 'found' (logistic relationship is not a characteristic of the relationship between AC₄/AC₃ and C₄ grass coverage specifically, rather it's a characteristic of any relationship where the response variable is fixed between 0 and 1 or 0% and 100%)

Answer: Thank you, we have changed it to "we found a strong positive relationship".

R4C5: Line 194 – use 'as high as'

Answer: We changed it to "as high as", following the suggestion.

R4C6: Line 238 – here and elsewhere maybe better to use 'air' in subscript in 'T_{air}'

Answer: We have changed T_{air} to T_{air} throughout the text.

Still, C. J., Berry, J. A., Collatz, G. J. & DeFries, R. S. Global distribution of C₃ and C₄ vegetation: Carbon cycle implications. *Global Biogeochemical Cycles* **17**, 6-1-6-14 (2003).